# Sea-Ice Deformation in a Coupled Ocean-Sea Ice Model and in Satellite Remote Sensing Data

Gunnar Spreen[1,2], Ron Kwok[2], Dimitris Menemenlis[2], and An T. Nguyen[2,3]

[1]University of Bremen, Institute of Environmental Physics, Bremen, Germany.
[2]Jet Propulsion Laboratory, California Institute of Technology, Pasadena, CA, USA.
[3]now at The University of Texas at Austin, TX, USA.

*Correspondence to:* Gunnar Spreen (gunnar.spreen@uni-bremen.de)

**Abstract.** A realistic representation of sea-ice deformation in models is important for accurate simulation of the sea ice mass balance. Simulated sea-ice deformation strain rates from model simulations with $4.5$, 9, and 18-km horizontal grid spacing are compared with synthetic aperture radar satellite observations (RGPS). All three model simulations can reproduce the large-scale ice deformation patterns, but small scale sea-ice deformations and linear kinematic features are not adequately reproduced. The mean sea-ice total deformation rate is about 50% lower in all model solutions than in the satellite observations, especially in the seasonal sea ice zone. A decrease in model grid spacing, however, produces a higher density and more localized ice deformation features. The dependence on length scale and probability density functions of absolute divergence for all three model solutions show a power-law behavior similar to the RGPS observations, and contrary to what is found in some other studies. Overall, the $4.5$-km simulation produces the lowest misfits in divergence, vorticity, and shear when compared with RGPS data. Model sensitivity experiments show a strong impact of the ice strength parametrization on the Arctic Basin sea ice volume, which increased by 7% and 35% for a decrease in ice strength of, respectively, 30% and 70%, after 8 years of model integration. This volume increase is caused by a combination of dynamic and thermodynamic processes. On the one hand, a weaker ice cover initially produces more ice due to increased deformation and new ice growth within the Arctic Basin. The thickening of the ice, on the other hand, increases the ice strength and decreases the sea ice volume export out of the Arctic Basin. The balance of these processes leads to a new equilibrium Arctic Basin ice volume. Not addressed in this study is whether the differences between simulated and observed deformation rates are an intrinsic limitation of the viscous-plastic sea ice rheology that was used in the sensitivity experiments, or whether it indicates a lack of adjustment of existing model parameters to better represent these processes. Either way, this study provides new quantitative metrics for existing and new sea ice rheologies to strive for.

## 1 Introduction

The Arctic sea ice in many respects is an important component of the Earth's climate system, e.g., sea ice governs the ocean to atmosphere heat flux, freezing and melting influences the upper ocean salinity and density, and sea ice dynamics act as a latent energy transport (Barry et al., 1993). During recent years substantial changes of the Arctic sea ice cover have been observed (e.g., Comiso et al., 2008; Kwok and Rothrock, 2009; Nghiem et al., 2007). Coupled ocean-sea ice models can reproduce

some aspects of sea ice and its recent changes (e.g., Zhang et al., 2008; Lindsay et al., 2009; Nguyen et al., 2011). In part this can be attributed to the fact that model parameters can be adjusted to produce observed ice concentration (extent) and drift distributions (Nguyen et al., 2011; Fenty et al., 2015). Detailed comparisons between satellite remote sensing data with model results, however, reveal big differences in certain aspects of the sea ice cover, e.g., for fracture zones and for small

scale dynamic processes (Kwok et al., 2008; Girard et al., 2009). It remains unclear whether current model physics are suited to reproduce these observed sea-ice deformation features (Coon et al., 2007) or if new sea-ice rheologies (e.g., Bouillon and Rampal, 2015b; Girard et al., 2011; Sulsky et al., 2007) have to be used. Sea-ice deformation is an important process for (1) sea ice mass balance due to new ice production and ridged ice formation, (2) brine rejection into the ocean due to freezing in open water areas, (3) regulation of ocean-to-air heat and gas fluxes, and (4) altering the air and water drag coefficients. Therefore a

realistic representation of sea-ice deformation in coupled sea ice-ocean models is important.

Here we study sea-ice deformation strain rates in the Arctic obtained from Synthetic Aperture Radar (SAR) satellite measurements using the RADARSAT Geophysical Processor System (RGPS) in comparison to coupled ocean-sea ice simulations carried out with the Massachusetts Institute of Technology general circulation model (MITgcm) as configured for the Estimating the Circulation and Climate of the Ocean, Phase II (ECCO2) project (Menemenlis et al., 2008). Model integrations

with horizontal grid spacing of 18, 9, and $4.5$ km are carried out. The model sensitivity to the model ice strength parameterization is assessed by comparing the model solutions with different ice strength parameters to the RGPS satellite observations spatially and temporally. These comparisons also allow us to study the model uncertainties regarding the sea-ice deformation representation in the current formulation of viscous-plastic sea ice models.

Traditionally sea ice model performance is evaluated by comparing satellite-derived ice area and velocities to model results

(e.g., Nguyen et al., 2011; Zhang et al., 2003). However, it can be shown that the Arctic sea ice velocity field can be divided into mean and fluctuating fields with the fluctuating field not behaving significantly different from a turbulent fluid (Rampal et al., 2009). It is therefore not sufficient to evaluate models on the basis of their first order mean velocity field as these can be correctly predicted even by simple sea ice models (i.e., using a viscous rheology). The second order sea-ice velocity field, represented by the sea ice deformation fields (strain rates), has to be used for comparison to take into account the high frequency

fluctuations of the sea-ice velocity field and to assess the quality of the sea-ice rheology formulation.

Sea ice strain rates do not scale linearly in space and time but follow a power law depending on the length scale $L$ and time interval $\Delta T$ over which the strain rates are integrated. For RGPS total deformation rates $\dot{D}$ in the Arctic, Marsan et al. (2004) and Stern and Lindsay (2009) observe a spatial scale dependence of $\dot{D} \approx dL^{-0.2}$ over a scale range from 10 to 10000 km. The constant $d$ can be interpreted as the mean deformation rate at a given base scale. To make meaningful comparisons between

observations and model simulations both have to be brought to the same reference frame in space and time, i.e., averages have to be calculated for the same area and time interval. Otherwise the scaling nonlinearity will cause nonphysical differences between the datasets.

It can be shown that traditional sea ice models using the Hibler (1979) viscous-plastic (VP) or elastic-visco-plastic (EVP) (Hunke and Dukowicz, 1997) ice rheology have difficulties in correctly representing the sea-ice deformation fields, especially

the distribution of the observed linear kinematic features (LKFs) (Kwok et al., 2008; Lindsay et al., 2003; Wang and Wang,

2009). Girard et al. (2009) also report distinct differences in the statistical scaling behavior of RGPS data and models using a VP and EVP sea ice rheology showing that the modeled deformation distributions can be close to Gaussian while the observed ones follow a power law. Improvements in modeled sea-ice deformation and thickness can be obtained by modifying the form of the yield curve away from an elliptical shape and/or changing the ratio of major to minor axes (Wang and Wang, 2009; Miller et al., 2005). To overcome some of the deficiencies of the viscous-plastic rheology, new ice rheologies with improved ice physics are under development in the hope of better representing the observed sea ice dynamics (e.g., Heil and Hibler, 2002; Sulsky et al., 2007; Girard et al., 2011; Bouillon and Rampal, 2015b). A recent example is the study of Tsamados et al. (2013), which demonstrates how an anisotropic ice rheology changes the sea ice mass balance and ice dynamics compared to the EVP rheology. Current VP and EVP sea ice model implementations, however, are robust and their parameters well tuned to reproduce the broad features of sea ice extent and drift. Therefore, they are widely used in coupled ocean-sea ice and in global climate simulations and thus their evaluation is necessary.

The main purpose of this article is to examine how model grid spacing influences simulated sea-ice deformation representation when compared to satellite observations. In comparison to previous studies we focus on direct comparison between the modeled and observed strain rates. Using the VP model, we construct simulated deformation fields on the same spatial and temporal scales as in the RGPS observations (section 2.3). We then compare the power law scaling properties of the modeled and observed deformation rates (sections 3.2.2and 3.2). To motivate this study we show in section 4 how the model ice strength parameterization influences the sea ice mass balance in the Arctic Ocean. This sensitivity study is similar to the one in Steele et al. (1997) but extends it by also taking changes in sea ice export into account. Itkin et al. (2014) show that the model sea ice strength parametrization can also affect the modeled Atlantic Ocean circulation. Ultimately, we would like to highlight why the sea-ice strength representation and the sea-ice rheology should receive more attention in models.

The remainder of this article is laid out as follows: Section 2 describes the model setup and introduces the RGPS satellite data. Section 3 contains the comparison between modeled sea-ice deformation and RGPS satellite observations. It contains an evaluation of the representation of sea-ice deformation dependencies on horizontal grid spacing both spatially and as time series, and shows the power law scaling behavior of the modeled and observed sea-ice deformation fields. To demonstrate how sensitive the model results depend on sea-ice deformation Section 4 shows how changes in sea-ice strength lead to changes of sea ice volume in the Arctic Basin. Finally, Section 5 concludes and further discusses the results.

## 2  Model Setup and Satellite Data

### 2.1  MITgcm Arctic Model Setup

The model output used for this study is obtained from integrations of a coupled ocean and sea ice configuration of the Massachusetts Institute of Technology general circulation model (MITgcm) (e.g., Losch et al., 2010). The model configuration is similar to that used for global integrations by the Estimating the Circulation and Climate of the Ocean, Phase II (ECCO2) project (Menemenlis et al., 2008), but only a sub-domain covering the Arctic Ocean including the surrounding marginal seas and parts of the North Atlantic and Pacific is used (see Figure 1a).

Briefly, the ECCO2 project uses a cube-sphere grid projection in a volume-conserving C-grid configuration. The ocean model has 50 vertical levels and employs the K-Profile Parameterization (KPP) of Large et al. (1994) for vertical mixing. The cold halocline layer of the Arctic Ocean is realistically reproduced with the use of the subgrid-scale brine rejection parameterization of Nguyen et al. (2009). The sea ice model uses 2-category, zero-layer thermodynamics (Hibler, 1980) and viscous-plastic (VP) dynamics (Zhang and Hibler, 1997; Hibler, 1979). The snow cover is simulated following Zhang et al. (1998). Table 1 summarizes the relevant sea ice parameters used for all model solutions presented herein.

The International Bathymetric Chart of the Arctic Ocean (IBCAO) (Jakobsson et al., 2008) is used as bathymetry, where available. For the remaining part of the model domain, which is not covered by IBCAO, the merged Smith and Sandwell/General Bathymetric Charts of the Oceans (GEBCO) is used and blended with IBCAO along the borders. Sea ice initial conditions (area and thickness) for January 1992 are from the Polar Science Center (Zhang and Rothrock, 2003) and ocean initial conditions (temperature, salinity, velocity) are from the World Ocean Atlas 2005 (Locarnini et al., 2006; Antonov et al., 2006). As lateral boundary conditions the globally optimized simulation from ECCO2 (Menemenlis et al., 2008) are used. Surface boundary conditions are obtained from the Japanese 25-year ReAnalysis (JRA-25; Onogi et al., 2007) with a spatial and temporal resolution of $1.125°$ ($\approx 120$ km) and 6 hours, respectively. These spatial and temporal resolutions do not allow to fully resolve all high frequency atmospheric forcing on the sea ice. Some ice deformation events will be missed, which adds uncertainty to the derived sea-ice deformation rates by the model.

Integrations with three different nominal horizontal grid spacings, 18 km, 9 km and 4.5 km, were performed. An example of the simulated sea ice thickness on 15 November 1999, after about eight years of model integration, is shown in Figure 1b–d for the three different grid spacings. The 4.5-km solutions clearly shows more details, e.g., develops lead patterns. The 18-km model solution was constrained by least squares fit to available satellite and in-situ data (e.g. ice drift, area, thickness) using a Green's function approach (Menemenlis et al., 2005) and is here referred to as the "baseline" simulation. A comprehensive evaluation of the 18-km model simulation and more detailed description of the optimization can be found in Nguyen et al. (2011). They show, by comparison to measurements, that the model using the optimized parameter set can realistically reproduce most important features of the coupled Arctic ocean and sea ice system. For example, sea ice extent and thickness as well as their trends are in good agreement with satellite and in situ measurements. Also the sea ice export through Fram Strait is modeled realistically compared to observations from Kwok et al. (2004). For the higher resolution (9 km and 4.5 km grid spacing) simulations we use the same set of parameters as those derived for the 18-km configuration. As a consequence these higher-resolution simulations exhibit somewhat larger model deviations relative to observations than the 18-km simulation. For example, the mean ice thickness on 15 November 1999 shown in Figure 1b–d shows a modes increase by 24 and 28 cm higher for the 4.5 and 18-km simulations, respectively, compared to the 9-km simulation. They nevertheless have been found of sufficient quality for process studies in the Arctic Ocean and adjacent seas (Nguyen et al., 2012; Rignot et al., 2012).

## 2.2 RGPS Satellite Observations

The RADARSAT Geophysical Processor System (RGPS) produces sea ice data products covering the Arctic Ocean derived from Synthetic Aperture Radar (SAR) imagery acquired by the Canadian RADARSAT satellite. Details of the analysis proce-

dures can be found in the papers of Kwok (1998) and Kwok and Cunningham (2002). In this study the "Lagrangian ice motion" dataset, one of the eight RGPS data products, is used as initial dataset. Sea-ice deformation, i.e., strain rates, are calculated from this ice motion dataset as described below. We start with the "Lagrangian ice motion" dataset to allow highest possible consistency between the observed and modeled deformation rates.

The 460-km wide swath ScanSAR Wide B (SWB) mode of RADARSAT (Raney et al., 1991) is selected to provide routine coverage of the Arctic Ocean for the RGPS system. The western Arctic Ocean is covered by RADARSAT images approximately once every three days. At the beginning of the season (winter or summer) an initial Lagrangian grid with 10 km grid spacing is set up. The movement and deformation of these grid cells are followed throughout the season. Grid cells are removed if they are advected out of the region of interest. Gaps in the ice motion data sets are due to the lack of backscatter contrast

for tracking ice features in the SAR imagery. The actual sea ice tracking is very accurate. Lindsay and Stern (2003) report that the median magnitude of displacement differences between buoy drift (via ARGOS positioning) and RGPS motion estimates is 323 m.

    RGPS observations are available since November 1996 until 2008. In this study we use RGPS data from 20 periods (11 winter and 9 summer) or 97 months between 1996 and 2008 (see Table 2).

## 2.3   Simulating RGPS data Using Model Solutions

    As a prerequisite for a meaningful comparison, the Lagrangian RGPS observations and Eulerian model output have to be brought to a common reference frame. We use the RGPS Lagrangian reference frame. This ensures that both RGPS and model sea ice strain rates are calculated for the same area and time interval. This procedure avoids differences between the datasets caused by the non-linearity of the strain rate scaling (power law dependence, see Sections 1, 3.2, and 3.2.2).

Every RGPS Lagrangian point $k(x_i, t_i)$ has a location, time, and time difference $\Delta t$ until the next observation attached to it. From this $\Delta t = t_{i+1} - t_i$ and the new position $x_{i+1}$ the velocity of point $k$ during the time interval $\Delta t$ can be calculated. We are bilinearly interpolating the Eulerian model velocities to the Lagrangian RGPS positions. The mean RGPS time interval $\Delta t$ is about 3 days, but $\Delta t$ varies from a few hours to about two weeks. We interpolate the mean model sea ice velocity during the individual $\Delta t$'s from the daily model output covering the $\Delta t$ time period.

After this consistent RGPS and model sea ice velocity dataset is established, sea ice strain rates are calculated using Delaunay Triangulation. From the triangle area $A$ and the sea ice velocity components $u$ in $x$ direction and $v$ in $y$ direction at the three triangle corners, the following partial derivatives can be calculated using the Divergence Theorem and the line integral around the triangle boundary:

$$\frac{\partial u}{dx} = \frac{1}{A} \oint u\,dy, \qquad\qquad\qquad \frac{\partial u}{dy} = -\frac{1}{A} \oint u\,dx$$

$$\frac{\partial v}{dx} = \frac{1}{A} \oint v\,dy, \qquad\qquad\qquad \frac{\partial v}{dy} = -\frac{1}{A} \oint v\,dx \qquad\qquad (1)$$

Using Equations 1 the strain-rates invariants divergence $\dot{\nabla}$, shear $\dot{\tau}$, and vorticity $\dot{\zeta}$ can be calculated:

$$\dot{\nabla} = \frac{\partial u}{dx} + \frac{\partial v}{dy}, \tag{2}$$

$$\dot{\tau} = \sqrt{\left(\frac{\partial u}{dx} - \frac{\partial v}{dy}\right)^2 + \left(\frac{\partial u}{dy} + \frac{\partial v}{dx}\right)^2}, \tag{3}$$

$$\dot{\zeta} = \frac{\partial v}{dx} - \frac{\partial u}{dy}. \tag{4}$$

As a measure of the total sea-ice deformation rate $\dot{D}$ we use

$$\dot{D} = \sqrt{\dot{\nabla}^2 + \dot{\tau}^2} \tag{5}$$

, which is used as a measure for the overall sea-ice deformation occurring at a certain point in space (e.g., Stern and Lindsay, 2009).

Erroneous cells, which might, e.g., arise due to errors in the ice tracking or from badly defined triangles from the Delaunay

triangulation, are filtered out using the following constrains: (1) The triangle cell area $A$ has to be between 5 and $400\,\mathrm{km}^2$. For the statistical comparisons and model to RGPS difference calculations, this condition is further restricted to $25 < A < 100\,\mathrm{km}^2$. This condition also assures that the length scale of all observations can be considered to be $\sim 10\,\mathrm{km}$, which is the initial RGPS grid spacing. This is important as sea ice strain rates are scale-dependent (see Section 1). (2) Triangles are not allowed to be overly distorted, i.e., not to be acute. To achieve this condition all angles have to be larger than $10°$. (3) The time interval

$\Delta t$ between two observations must be between 12 hours and 7 days. (4) Cells with a deformation rate $\dot{D}$ (see Equation 5) higher than 1 day$^{-1}$ are considered outliers and are removed. Only filter (4) creates a different number of observations for the RGPS and model dataset (because $\dot{D}$ can differ between model and RGPS). However, to keep the number of observations equal in both datasets, filtered data points from one dataset are also removed from the other one. We do not use a specific smoother as suggested in Bouillon and Rampal (2015a) to remove artificial noise in the sea ice motion fields. This may lead to

an overestimation in the magnitude of the scaling exponent $b$ (Bouillon and Rampal, 2015a) investigated in Sections 3.2 and 3.2.2. We, however, remove acute triangles susceptible to noise and high deformation rates as described above.

## 3  Modeled Sea-Ice Deformation Compared to RGPS Observations

In this section, we compare the simulated sea-ice deformation distribution to satellite observations. Big differences between observed and modeled sea-ice deformation fields have been reported (Lindsay et al., 2003; Kwok et al., 2008; Girard et al.,

2009; Wang and Wang, 2009, , see also section 1). Kwok et al. (2008) evaluated four common sea ice models with horizontal grid spacing ranging from 9 to $40\,\mathrm{km}$. None of these models could produce realistic distributions of small-scale deformation features and linear kinematic features (LKFs), although the large-scale sea-ice deformation pattern was reproduced correctly by some of the models. The model with the smallest grid spacing ($9\,\mathrm{km}$) showed the most confined LKFs. It was speculated that if the model grid spacing would be further decreased, the model could eventually produce more realistic details and have a

better representation of LKF distribution. Girard et al. (2009) compared the statistics of VP and EVP simulations with 12-km

grid spacing to RGPS data and also reported large differences, as did Wang and Wang (2009) and Lindsay and Stern (2003) for different model setups. We use a slightly different approach and reconstruct the RGPS observations from model velocity fields (section 2.3) and explore how the LKF representation changes when the model resolution increases (section 3.1). We also compare the power law scaling between our model simulations and the RGPS data (section 3.2).

## 3.1 Dependence on Model Grid Spacing

### 3.1.1 Spatial Patterns and LKFs: Divergence, Vorticity, and Shear

Figures 2, 3, and, 4 show the monthly November 1999 divergence, vorticity, and shear fields, respectively, obtained from RGPS data and from the three model solutions with $4.5$, $9$, and $18$ km grid spacing. For all maps both the Lagrangian RGPS data and the reconstructed Lagrangian model solutions (see Section 2.3) were interpolated on the same polar stereographic grid with $12.5$-km grid spacing. This means that all differences visible in the model maps (at least for the 9 and 4.5-km ones) are due to changed behavior of the model physics and can not be attributed to the different model grid spacing alone. The $12.5$-km grid spacing are a slight oversampling for the 18-km model output but an undersampling for the 9 and 4.5-km model solutions. Figures 2 to 5 also show a black contour discriminating multiyear ice from first-year sea ice based on QuikSCAT backscatter data.

In general, the large-scale sea-ice deformation patterns are reproduced by the model for all three grid spacings. In November 1999 a pattern of high divergence (Figure 2) can be observed in the Beaufort Sea and a more convergent situation north of the Chukchi and East Siberian Sea (see Figures 1 and 2 for locations). This pattern is also present in all three model solutions, but much weaker. In the RGPS observations the pattern is broader and covering most of the seasonal sea ice in that region. The high divergence in the Beaufort Sea is accompanied by negative vorticity (Figure 3), which can be observed in the RGPS data as well as in the three model solutions. Also the positive vorticity pattern north of Ellesmere Island with strong LKFs is reproduced in all three model integrations. The same is true for the positive vorticity pattern in the East Siberian Sea and the negative vorticity north of the Laptev Sea.

The RGPS data show strong sea ice shear almost everywhere in the marginal sea ice zone (Figure 4). This area of high shear is only partly reproduced by all three model solutions. All three model solutions show almost no large-scale shear patterns. In the Beaufort and East Siberian Seas, only small areas of high shear are present. From the three deformation variables divergence, shear, and vorticity the agreement between the large scale RGPS and model shear is the worst. The agreement of the vorticity patterns between RGPS and models is the best. However, the magnitudes of divergence, shear, and vorticity for all three model solutions are much smaller (less than half, see next section) than the RGPS ones. These statements are true not only for the November 1999 example shown here but also for almost all of the other months with available RGPS data (see Table 2) and will be further discussed in Section 3.1.2.

Next we qualitatively compare the distribution and frequency of occurrence of LKFs followed by more quantitative comparisons in the next sections. The model solutions for all three grid spacings do have significantly less LKFs than the RGPS data. This is true for all three deformation variables: divergence, shear and vorticity. Between the three model solutions there are,

however, significant differences for the LKF distribution. While, e.g., the sea ice shear for the 18-km model solution in Figure 4 shows very little identifiable LKFs, the number of LKFs slightly increase for the 9-km solution and significantly increase for the 4.5-km solution. The same can be observed for the divergence and vorticity fields. The 4.5-km model solution always shows the most LKFs and its deformation distribution is most consistent with RGPS observations based on visual inspection. This conclusion holds for all 97 months with available RGPS data that were analyzed and will be further discussed in Section 3.1.3.

The large-scale difference in sea-ice deformation between RGPS observations and model solutions is not evenly distributed over the Arctic Basin as can already be seen from Figures 2 to 4. Figure 5 shows the deformation rate difference $\Delta\dot{D} = \dot{D}_{\text{RGPS}} - \dot{D}_{\text{MODEL}}$ for the 4.5, 9, and 18-km solutions during November 1999. All three difference maps are smoothed with a 150-km kernel to remove small scale differences (e.g., LKFs) and highlight the large-scale difference patterns. The large-scale difference patterns are very similar for all three model grid spacings. The representation of large-scale sea-ice deformation in the model therefore does depend less on the model grid spacing than the small scale deformation distribution. There is, however, some seasonal dependence as we will see in Section 3.1.2.

The main differences in $\Delta\dot{D}$ are confined to the seasonal ice zone (outside the black contour in Figure 5). In general the seasonal sea ice is thinner and more mobile than the older, thicker perennial ice. For the perennial ice, $\Delta\dot{D}$ is much smaller and mainly stays below $0.02\,\text{day}^{-1}$. This discrepancy between seasonal and perennial ice hints to a shortcoming of the sea ice rheology used in the simulations. To first order the main difference between seasonal and perennial sea ice is the ice thickness. The model sea ice strength $P$, as defined in Equation 8, depends linearly on ice thickness $h$. This is the typical $P$ formulation for a VP or EVP sea ice rheology with two ice classes and might not be the best representation of the $P$ to $h$ relationship. Models with more ice thickness classes often use a $P \propto h^{3/2}$ formulation (Rothrock, 1975; Lipscomb et al., 2007), which can be considered more realistic. There are, however, also other differences between the seasonal and perennial ice zone than the ice thickness. The proximity to open water, for example, will allow more cases of ice divergence at the ice margins than in the ice pack, which might be less well represented by the VP rheology. Anyway, in times of a changing Arctic environment, where seasonal sea ice is becoming the dominant ice type (Comiso, 2012), the problem of large discrepancies in simulated sea-ice deformation of the seasonal ice zone will become more severe.

### 3.1.2 Time Series

For this study RGPS observations for 97 months from 20 RGPS observation periods between November 1996 and May 2008 are used (Table 2). Figure 6 shows (a) the period-averaged sea-ice deformation rate $\dot{D}$ and (b) the monthly-mean seasonal cycle of $\dot{D}$ (both computed with all 20 RGPS periods available). Months September and October are not covered by RGPS data. The time series of $\dot{D}$, $|\dot{\nabla}|$, $|\dot{\tau}|$, and $\dot{\zeta}$ behave very similarly. For simplicity we will therefore concentrate the discussion on the sea-ice deformation rate $\dot{D}$ (Figure 6) but the statistics for all variables are presented in Table 3.

The RGPS deformation rate (black) is consistently higher than the one of the 4.5-km (total mean +51%), 9-km (+55%), and 18-km (+57%) simulations. The same is the case for divergence, shear, and vorticity. The largest difference occurs for absolute divergence, which is 67% to 79% higher for the RGPS data (Table 3). Overall, we conclude that the absolute amount

of sea-ice deformation in our current sea ice model setup is about 50% too low in comparison to RGPS observations and this underrepresentation of deformation is almost independent of model grid spacing during winter months. During summer months, however, the model performance differs depending on horizontal grid spacing and the 4.5-km simulation shows the smallest difference to RGPS observations. This can be seen in the seasonal cycle in Figure 6b where during December to April the three model solutions are indistinguishable and agree within their standard deviation. Only during summer months (June to August) the 4.5-km solution shows a higher deformation rate than the 9-km solution, which again shows a higher deformation rate than the 18-km solution. The RGPS data show a clean, sinusoidal-like seasonal cycle with a clear minimum in March and maximum in August (likely the real maximum would occur during the unobserved month of September). The 9 and 18-km model solutions do not show a sinusoidal behavior. They have a clear maximum during August but no defined minimum. $\dot{D}$ is almost constant during January to May. The 4.5-km solution slightly differs from this general behavior and shows a small but not very pronounced March minimum compared to RGPS data. That is, the 4.5-km solution again shows a better performance than the lower-resolution simulations.

The RGPS and all model deformation time series are highly correlated ($R^2 \approx 0.9$). As is the case for the mean deformation rate, however, the variability of the modeled deformation rate is also much smaller than the observed RGPS variability. The standard deviation $\sigma$ of the monthly $\dot{D}$ time series (not shown) is about 50% smaller for the 18, 9, and 4.5-km solutions ($\sigma = 0.4$ to $0.7 \cdot 10^{-2}$ day$^{-1}$) compared to RGPS data ($\sigma = 1.1 \cdot 10^{-2}$ day$^{-1}$, see Table 3). Again the 4.5-km solution performs best.

### 3.1.3 Localization of Deformation

As seen in Section 3.1.2, the absolute amount of sea-ice deformation in the model is much too low. Nevertheless, in Section 3.1.1 it was shown that the modeled sea-ice deformation distribution gets more similar to the observed one if model grid spacing is decreased. In particular, more and better-confined LKFs appear for smaller grid spacing (e.g., Figure 4). To show this change in the sea-ice deformation distribution more quantitatively we will look at the "localization" of the deformation rate. Following Stern and Lindsay (2009) we calculate the area fraction $Q$, which contains the highest 15% of all sea-ice deformation rates. $Q$ is calculated as:

$$\dot{D}_1 \geq \dot{D}_2 \ldots \dot{D}_{n-1} \geq \dot{D}_n$$

$$\sum_{i=1}^{p} \dot{D}_i = 0.15 \sum_{i=1}^{n} \dot{D}_i$$

$$Q = \sum_{i=1}^{p} A_i \bigg/ \sum_{i=1}^{n} A_i \,,$$

where $\dot{D}_i$ are the individual Lagrangian deformation rate observations sorted by their magnitude starting with the highest. $A_i$ are the accordant triangle areas. The number of observations $n$ is identical for all model simulations and the RGPS data. The smaller the percentage $Q$ gets, the more localized the deformation is distributed. If the deformation rates would be evenly distributed the highest 15% would also occupy 15% of the area. We will see that $Q$ indeed is much smaller than that. As $Q$ is

normalized by the total deformation rate of each complete dataset (either RGPS or model solution) this measure is independent of the absolute amount of deformation rate. Figure 7 shows (a) the time series of $Q$ for all 20 RGPS periods for the three model solutions and the RGPS data and (b) the seasonal cycle of $Q$ (also see Table 3 for statistics).

Sea ice deformation in both the RGPS observations and all three model solutions is very localized. The highest 15% of all deformation rates $Q$ is only occupying between $0.5$ and $1.5\%$ of the total area. $Q$ decreases with decreasing model grid spacing. There is a big difference in $Q$ for the $4.5$-km simulation ($\overline{Q} = 0.5\%$) compared to $Q$ of the 9 and 18-km simulations ($\overline{Q} = 1.2\%$ and $\overline{Q} = 1.5\%$, respectively). The mean $\overline{Q} = 0.5\%$ of the $4.5$-km simulation is even lower than the localization $\overline{Q} = 0.7\%$ of the RGPS observations, which shows that the sea-ice deformation distribution got considerably more confined for the $4.5$-km simulation compared to the other two lower-resolution simulations. This can also be seen in the examples of Figures 2 to 4, which show a strong increase in the number of LKFs when the grid spacing is reduced from 18 and 9 km to $4.5$ km. The strain rate distributions for the 18 and 9-km simulations are much more similar. This is confirmed here by the very strong localization $Q$ for the $4.5$-km solution, which is also closest to the RGPS observations. It is not clear why the change in $Q$ is so big for the $4.5$-km solution compared to the other two solutions. Disregarding the big difference in the mean deformation rate, the $4.5$-km simulation is able to reproduce the fraction of the total area, in which the strong sea-ice deformation events are concentrated very well. Also the seasonal cycle of $Q$ in Figure 7b is more similar for the $4.5$-km solutions compared to RGPS observations than for the other two simulations. The seasonal cycle for the 18 and 9-km simulation is strongly enhanced and shows a drop during summer months July and August, which is not the case for the RGPS observations and the $4.5$-km simulation. The unnatural seasonal cycle for these two simulations also significantly increases the standard deviation of $Q$ for the monthly time series: $1.2\%$ for the 18 and 9-km model solutions and $0.4\%$ for the RGPS data and $4.5$-km simulation. This results in larger differences in $Q$ between 18 and 9-km model simulations and RGPS during summer and hints towards a degraded performance of the 18 and 9-km model simulations to represent sea ice deformation during summer.

In summary, sea-ice deformation in the model solution with the finest grid spacing of $4.5$ km is most confined and localized, as had already been seen in the examples of Section 3.1.1. One has to keep in mind, however, that the absolute model deformation is only about half that of the observations. From the three model solutions, the $4.5$-km simulation can be considered most consistent with the RGPS observations.

## 3.2 Power Law Scaling of Deformation Rates

Sea ice strain rates do not scale linearly in space and time. Instead the scaling follows a power law. Some details about the nature of this scaling dependence are, e.g., given in Weiss (2003, 2013). In Sections 3.2.1 and 3.2.2 we will compare the power law scaling of the model solutions with the RGPS data based on length scale and their respective probability density functions. There is some controversy in current literature how well the VP ice rheology is suitable to reproduce this power law scaling (e.g. Girard et al., 2009). In Section3.2.3 we will use the power laws scaling dependence to compare the sea ice deformation rate for three model solutions with different grid spacing.

### 3.2.1 Dependence on Length Scale

The magnitude of sea ice strain rates depends on the spatial scale over which they are determined. We exemplary use the absolute divergence $|\dot{\nabla}|$ in this section but similar relationships exists for shear $\dot{\tau}$, and vorticity $\dot{\zeta}$. For absolute divergence $|\dot{\nabla}|$ and length scale $L$ over which $|\dot{\nabla}|$ is determined this power law scaling can be expressed as:

$$|\dot{\nabla}| \approx dL^b \tag{6}$$

$b$ is the scaling exponent, and $d$ a constant of proportionality, which can be interpreted as mean deformation rate at a given base scale. We use the consistent Lagrangian strain rate dataset described in Section 2.3 to compare RGPS observations with model solutions. Following the procedure described in Stern and Lindsay (2009) strain rates at different spacial scales are calculated:

– Strain rates for the six nominal length scales $L^* = 10$, 20, 50, 100, 200, 500, and 1000 km are calculated. The initial
length scale of the RGPS data is 10 km and therefore this is the smallest scale that can be obtained.

– All Lagrangian cells within a 5-day window are aggregated for grid cells of size $L^*$.

– A filter is applied: The time interval $\Delta t$ of the Lagrangian cells must be between 2 and 5 days and the area between 25
    and 100 km$^2$. The sum of all cell areas must be greater than $0.75L^{*2}$.

– Averages of $\partial u/dx$ and the other strain rates (see equation 1) are computed by using the cell areas as weight, and the
deformation invariants ($\dot{\nabla}$, $\dot{\tau}$, $\dot{\zeta}$) are computed from the averages.

– The length scale $L$ for each sample is determined by the square root of all cell areas.

Figure 8 shows the absolute divergence $|\dot{\nabla}|$ versus the length scale $L$ for RGPS observations and the three model solutions with 4.5, 9 and 18 km grid spacing on a log-log scale. The dataset was split in the winter and summer RGPS periods (Table 2). The averages $|\dot{\nabla}|$ for the six nominal scales 10, 20, 50, 100, 200, 500, and 1000 km are marked by symbols.

The RGPS observations and as well all three model solutions follow a power-law scaling, both during winter and during summer. Only the mean divergence rate at 1000 km diverges significantly from this relationship. This, however, can be attributed to the low number of samples at the 1000 km scale. Figure 8 shows least square fits for the five mean values between 10 and 500 km as dashed lines. While both the observations and model solutions follow a power-law, the inclinations of the fit, i.e., the scaling exponent $b$ in equation 6 is steeper for RGPS than for the model solutions. $b$ for RGPS is $-0.24$ during
both winter and summer. All model solutions show a $b$ of about $-0.1 \pm 0.02$ during winter. During summer the $b$ values of the model solutions differ with $b = -0.15$ for the 4.5-km solution, closest to RGPS, and $b = -0.09$ and $-0.07$ for the 9 and 18-km solutions, respectively (all $b$ values are summarized in Table 3).

    Our estimates of $b$ for RGPS agree well with previous estimates from, e.g., Marsan et al. (2004) and Stern and Lindsay (2009), who report a $b$ value of $\approx -0.2$ during winter and $\approx -0.3$ during summer in the Arctic. Our split in summer and winter
periods does not seem to resolve the seasonal cycle. Based on the same sea ice drift dataset Bouillon and Rampal (2015a) find the magnitude of the scaling exponent to be about 50% lower (i.e., $b \approx -0.12$ during winter) for the deformation rate. They

attribute the higher scaling exponent in the original RGPS data to artificial noise, which they reduce by a smoother. This could explain the difference in the scaling exponent between our not smoothed RGPS results and the model solutions. The $b$ value of the model solutions during winter agrees within their uncertainty estimate with the $b$ value of $-0.12$ found by Bouillon and Rampal (2015a). We do not observe the strong divergence from power-law scaling for models with VP ice rheology reported by Girard et al. (2009).

### 3.2.2 Probability Density Function

Another way to look at the power-law scaling behavior of sea ice deformation rates is by comparing probability density functions (PDFs) obtained from model solutions and RGPS data. The PDFs for observed sea ice strain rates follow a power law. For example, Girard et al. (2009) report that the PDF of RGPS strain rates during January to March 1997 follows a linear relation in log-log space:

$$p(|\dot{\nabla}|) \propto |\dot{\nabla}|^n \tag{7}$$

For the comparisons, the same 5-day aggregated RGPS and model datasets described in the last Section 3.2.1 were used. We show results obtained for the length scale $L = 20\,\text{km}$. PDFs $p$ for absolute divergence $|\dot{\nabla}|$ from all winter (11 years) and summer (9 years) RGPS periods (see Table 2) were then calculated. Figure 9 shows the PDFs for the three model solutions with $4.5$ (blue), $9$ (green), and $18\,\text{km}$ (red) grid spacing and the RGPS data (black) on a log-log scale.

A linear regression was applied to the PDFs in log-log space for the absolute divergence range $0.1$–$1\,\text{day}^{-1}$, shown as dashed lines in Figure 9. For very small and large deformation rates outside that range, the RGPS PDFs diverge from the power law relationship. The accuracy of the RGPS observations is about $100\,\text{m}$ and noisy at that scale. This noise, which is not removed in this study, can cause artificially higher strain rates (Bouillon and Rampal, 2015a). Low deformation rates therefore could be underrepresented in the RGPS PDF, which potentially could explain the deviation from a straight line for low deformation rates in Figure 9. For very high deformation rates the low number of data points causes artificial variability in the PDFs.

The slope of both the winter and summer RGPS PDF is $n \approx -3$ (winter: $n = -3.0 \pm 0.1$; summer: $n = -2.84 \pm 0.09$). This is consistent with Marsan et al. (2004) and Girard et al. (2009), who report winter RGPS PDF slopes of about $-2.5$ for strain rates at the $\approx 10$-km scale. All slopes $n$ for RGPS and the model simulations are summarized in Table 3. During winter, the slope for the PDFs of all three model solutions is very similar ($4.5\,\text{km}$: $n = -2.74$; $9\,\text{km}$: $n = -2.9$; $18\,\text{km}$: $n = -2.8$) and agrees very well, mostly within the error bars, with the RGPS slope. During summer, the model simulation slopes are smaller and diverge more from the RGPS data ($4.5\,\text{km}$: $n = -2.4$; $9\,\text{km}$: $n = -2.52$; $18\,\text{km}$: $n = -2.5$). Still the agreement can be considered as good.

During winter, the three model solutions show a power-law scaling behavior over an even larger absolute divergence range than the RGPS data (approximately $3 \cdot 10^{-2}$ to $1\,\text{day}^{-1}$). During summer, the model solutions PDFs are more noisy and especially the $4.5\,\text{km}$ solutions diverges from the power-law relationship for large divergence rates. The model solutions show a higher probability for small absolute divergence rates than the RGPS data as can be expected from the about 50% lower deformation rates discussed before.

Overall the PDFs for simulated and observed RGPS deformation rates show good agreement. The observed and simulated power law exponents $n$ agree during both winter and summer months. During winter months both the exponent $n$ and the PDF shapes from the model solutions are very close to the RGPS data with the $4.5\,\mathrm{km}$ solution being closest. During summer there is a larger spread between the model solutions and the $9\,\mathrm{km}$ solutions agrees best with the RGPS data. Again, we do not observe

the strong deviation from power-law scaling reported by Girard et al. (2009) for model simulations using the VP and EVP sea ice rheology. In our model setup the used VP rheology seems to be able to reproduce a realistic distribution of deformation rates, which follow a power-law relationship.

### 3.2.3 Comparing models with different grid size

In this section, we examine whether sea-ice deformation rates in the three model simulation with different horizontal grid
spacing follow a similar power law scaling as found in observations and as discussed in Section 3.2.1. It is a common problem that one wants to compare sea ice deformation rates from different model simulations. These model simulations then, in general, have a different grid resolution and a direct comparison is not possible due to the different length scales involved. We will explore if the power law in equation 6 with a constant exponent $b$ can be used to compare mean absolute deformation rates of model solutions with different grid spacing as it was suggested by, e.g., Stern and Lindsay (2009).

Due to the different averaging length scale $L$ one would not expect $\dot{D}$ to be the same for model solutions with different grid spacing. In Section 3.1 we avoided this problem by interpolating the model solutions to the RGPS Lagrangian locations. At least for the model solutions with higher or similar spatial scale as the RGPS data, i.e., the $4.5$ and $9$-km solutions, this will create comparable datasets. Due to its lower spatial scale, the $18$-km solution cannot, in theory, fully recreate the RGPS data, regardless of the sea ice rheology formulation.

Figure 10a shows the 1992–2008 time series of the mean sea-ice deformation rate $\dot{D}$ in the complete model domain of Figure 1 inset. Different to the previous sections and, e.g., Figure 6, the complete model domain is now considered, not only the areas covered by RGPS data. As expected the deformation rate for the $4.5$-km model solution (blue, mean $\dot{D} = 0.123/\mathrm{day}$) is consistently higher than that of the $9$-km solution (green, mean $\dot{D} = 0.085/\mathrm{day}$, $-31\%$), which itself is higher than that of the $18$-km solution (red, mean $\dot{D} = 0.054/\mathrm{day}$, $-36\%$). The variability from year to year of the mean deformation rate is large,
especially during summer. Some years, e.g., 1997–1999, have clearly reduced summer deformation rates in comparison to, e.g., the beginning of the 1990s or 2007 and 2008. The deformation rate during 2008, both during summer and winter, is the highest of the complete time series (Figure 10a).

We assume that the model deformation rate $\dot{D}$ follows the same power-law as given in equation 6 and apply a least-squares fit in log space to equation 6:

$$\log(\dot{D}_i) = \log(d) + b \log(L_i) \quad (i = 1 \text{ to } 3)$$

with daily mean deformation rates $\dot{D}_i$ from model solutions with grid spacing $L_i$, i.e., in our case $4.5$, $9$ and $18\,\mathrm{km}$. For all sea-ice-covered areas in the model domain and for the complete time series, the power law scaling exponent $b$ is estimated to be $-0.54$. Figure 10b shows the deformation rate time series for the three model solutions normalized to a length scale

of $L = 10\,\mathrm{km}$, using the estimated scaling exponent $b = -0.54$ and equation 6. The length scale of $10\,\mathrm{km}$ was chosen to be comparable to the RGPS data. Using this scaling, the three time series become much more similar than the original ones in Figure 10a. If looked in detail, however, there remain some quite large differences. For example, the mean $\dot{D}$ of the 9-km simulation is now higher than that of the other two simulations; and the standard deviations of all three simulation are still

different (not shown: the standard deviation of the 18-km simulation is $> 0.05/\mathrm{day}$ smaller than that of the 9 and 4.5-km simulations).

These differences imply that a single, constant scaling exponent $b$ is not sufficient to make the strain rates of the three model solutions comparable. $b$ varies seasonally and regionally. Figure 10c and d show, respectively, the dependence of sea-ice deformation rate $\dot{D}$ on sea ice concentration $C$ and sea ice thickness $h$ for the three model solutions during the complete 1992

to 2008 time series. In Figure 10c, the deformation rate decreases with increasing sea ice concentration for all three model runs and $\dot{D}$ approaches zero linearly for 100% ice-covered grid cells. Also for increasing ice thickness in Figure 10d the deformation rate decreases but here the deformation rate decreases exponentially. For sea ice thickness above $2\,\mathrm{m}$, $\dot{D}$ is near zero. It has to be noted that the ice thicknesses $h$ are the effective ice thicknesses of a complete grid cell, which also can contain open water ($C < 100\%$).

From Figures 10c and d, it becomes clear that the scale dependence is much stronger for small ice concentrations and thicknesses than for large ones. The scaling exponent $b$ gets more negative for weaker sea ice and approaches zero for very strong sea ice, i.e., thick ice and 100% ice concentration (see Section 4 and Equation 8 for how the ice strength dependencies are incorporated in the model).

There are additional external factors that influence $b$. For free-drift ice, $b$ gets more negative as can be seen by the strong

dependence on $C$. Therefore, the surrounding geography, i.e., landmasses, influence the scaling exponent with $b$ values closer to zero in channels and near the coast, where the ice cannot drift freely. The estimated power-law scaling factor $b$ represents the balance between all these factors. That is, sea ice concentration, thickness, and geographic location are important contributors to the estimated scaling exponent.

The above factors also explain why the scaling exponent $b = -0.54$ found here for the three simulations is significantly lower

than the values of $b$ of about $-0.2$ found for RGPS data in Section 3.2.1 or by Stern and Lindsay (2009). In the model, the values of $b$ between $-0.3$ to $-0.2$ are typical for ice concentrations $\geq 80\%$. These are typical ice concentrations for the RGPS region, which rarely extends to the marginal ice zones with low ice concentrations. If the calculation of the scaling exponent $b$ in the model is restricted to the region covered by RGPS data, a mean $b$ value of $\approx -0.2$ is found, which is comparable to the $b$ values found for RGPS data. This scaling exponent, however, is not applicable to the complete Arctic. For this reason, it is

difficult to compare sea-ice deformation rates obtained at different spatial scales. For direct comparison, strain rates need to be calculated for identical areas, as was done in Section 2.3. At the very least, for meaningful statistical comparisons, the different scaling behavior for different ice concentrations needs to be considered.

In summary, the three simulations with different horizontal grid spacing, i.e., different resolved spatial scales, follow a similar power law scaling as that estimated using RGPS and buoy observations. We attribute most of the differences between simulated

and observed scaling factor $b$ to the different sea ice concentration and thickness ranges of each dataset. The simulated power

law scaling strongly depends on ice strength, which itself depends on ice concentration and thickness. For strong sea ice, all model solutions converge to comparably small deformation rates.

## 4    Sensitivity of Modeled Sea Ice Mass Balance on Ice Strength Parameterization

In section 3 it was shown that there can be significant differences in the absolute value of simulated and observed sea ice deformation rates. In our case the simulated deformation rates were about 50% lower than the once observed by satellite. In addition, especially for model solutions with grid spacing above 5 km, the spatial distribution of LKFs was very different between model solutions and observations. The question arises if such differences between modeled and observed sea ice deformation rates are important for a realistic simulation of sea ice. In this section we will show in an sensitivity experiment how changed simulated sea ice deformation rates influence the Arctic sea ice mass balance.

Changes in ice strength modifies the mechanical redistribution of sea ice through ridging, rafting and open water production, and thus modifies the ice thickness distribution. Furthermore, changes in the model ice strength alter the sea-ice drift speed and thereby the sea-ice export out of the Arctic Ocean. On a broader scale, these changes can alter the equilibrium sea ice volume in the Arctic. Here we perform a set of experiments to test the sensitivity of our 18-km model to changes in ice strength parameter to highlight the importance of using accurate rheological models and sea-ice deformation fields for the Arctic sea ice mass balance.

The total sea-ice deformation rate $\dot{D}$ (see Equation 5) is used as a measure for the overall sea-ice deformation occurring at a certain point in space. The magnitude of $\dot{D}$ is to some extent controlled by the strength of the sea ice. In our model configuration, we use the typical ice pressure formulation $P$ (or strength) of Hibler (1979):

$$P = P^* h\, e^{[C^*(1-C)]} \tag{8}$$

The ice strength $P$ depends linearly on the ice thickness $h$ and exponentially on the ice concentration $C$. $P^*$ and $C^*$ are scaling constants for the ice strength parameterization. In the optimized Arctic solution with 18-km horizontal grid spacing of Nguyen et al. (2011), $P^* = 23\,\mathrm{kN/m}^2$ and $C^* = -20$ are used. Therefore the value of $P^* = P_0^* = 23\,\mathrm{kN/m}^2$ is used as a baseline and will be compared to two solutions with $P^* = 16\,\mathrm{kN/m}^2$ (70% of baseline value, named "$0.7P_0^*$") and $P^* = 7\,\mathrm{kN/m}^2$ (30% of baseline value, named "$0.3P_0^*$").

The $P^*$ value of $16\,\mathrm{kN/m}^2$ is within the range of values used in current sea ice models (Martin and Gerdes, 2007), while the $P^*$ of $7\,\mathrm{kN/m}^2$ is at the lower end of the used $P^*$ values and can be considered weak ice. We note that the original value of $P^* = 5\,\mathrm{kN/m}^2$ in Hibler (1979), which was even lower than the values we use in this study, is still used and found appropriate for modern setups (e.g., Miller et al., 2005). The differences in the values of $P^*$ that are used in different models come in part because of the need to calibrate the parameters of one's model depending on the forcing used (ocean + atmosphere) and surface drag formulations. There is also the need to recalibrate this $P^*$ parameter depending on the spatial resolution used in the model. All model simulations were carried out for the period 1992 to 2009.

The mean sea-ice deformation rate $\overline{D}$ and sea ice volume $\overline{V}$ within the Arctic Basin, and export of Arctic sea ice volume $\overline{E}$ are analyzed. Figure 11a shows the borders of the Arctic Basin used here. Export $\overline{E}$ is calculated as the sum of sea ice volume fluxes through all red boundaries. All analyses are based on monthly mean values, including the deformation rate calculation.

Figure 11b shows the seasonal cycle of deformation rates $\overline{D}$ for the 1992–2009 period As expected for any given month the deformation rate increases for weaker sea ice. The deformation rate $\overline{D}$ increases by 10% for $0.7P_0^*$ (green) and by 37% for $0.3P_0^*$ (red) compared to the baseline integration (blue). As the deformation rate does not scale linearly, the higher $\overline{D}$ values for $0.3P_0^*$ are expected. All three simulations show a clear seasonal cycle for the deformation rate with a maximum in August and September and a minimum during February to April. The differences in deformation rate "$0.7P_0^*-$baseline" and "$0.3P_0^*-$baseline" in Figure 11c show quite some variability during 1992–2009 and increased deformation rates during the first year of integration, especially for $0.3P_0^*$. During the integration time the difference in deformation rate decreases for both experiments causing negative trends of $-12 \pm 5\%$ ($-1.4 \pm 0.5 \cdot 10^{-3}\,\mathrm{day}^{-1}$) per decade for $0.3P_0^*$ and a statistically insignificant trend of $-5\%$ per decade for $0.7P_0^*$. The sea ice speed (not shown) shows a similar behavior with higher ice speeds for weaker ice. Consequences of changes in the deformation rates and ice velocity on the sea ice mass balance are discussed in the remainder of this section.

## 4.1 Changes in Arctic Basin Sea Ice Volume

In this section, we discuss the differences in sea ice volume between the three solutions. For a discussion of geophysical sea ice volume change over time, see Nguyen et al. (2011). Figure 12a shows the difference of total sea ice volume $\Delta\overline{V}$ in the Arctic Basin from 1992 to 2009 between the baseline model solution and the $0.7P_0^*$ (green) and $0.3P_0^*$ (red) solutions. The ice volume for the two "weak" solutions quickly diverges from the baseline solution. Hence, after 8 years of integration, the sea ice volume has increased by 7% and 35% for $0.7P_0^*$ and $0.3P_0^*$, respectively, compared to the baseline. In 2009, at the end of the integration, the differences are 6% ($870\,\mathrm{km}^3$) and 45% ($6700\,\mathrm{km}^3$), respectively. The 1992–2009 mean sea ice volume in the Arctic Basin is $17607\,\mathrm{km}^3$ for the baseline integration. The sea ice volume of the $0.3P_0^*$ solution diverges from the baseline at a much faster rate than for the solution with $0.7P_0^*$. The rate of increase of the ice volume gets smaller after 1999, but the volume keeps increasing until 2005. Thereafter the difference to the baseline slightly decreases again. The $0.7P_0^*$ solution also diverges until 1999 but by a much smaller rate. Thereafter, the difference stays almost constant with a small negative trend. A similar sensitivity of ice thickness to the choice of the ice strength parameter $P^*$ was reported by Steele et al. (1997) and Itkin et al. (2014). A reduction of $P^*$ initially leads to a speed up of the Beaufort Gyre, which can cause enhanced dynamic ice growth by ice compression along the Greenland and Canadian costs as reported by Steele et al. (1997). We will add to these observations by also taking the balance between ice production and ice export into account.

Reasons for the observed ice volume increase will be discussed in the following. The changes in Arctic Basin sea ice volume can either be caused by changes in ice production/melting (thermodynamic and dynamic; Section 4.2) or by changes of the sea ice volume export out of the Arctic Basin (Section 4.3).

## 4.2 Sea Ice Production and Melt

The net Arctic Basin sea ice production (winter) or melting (summer) $\overline{B}$ is calculated as ice volume changes from one month, $(m-1)$, to the next month, $m$, after adding the sea ice export $\overline{E}$ during month $m$:

$$\overline{B}(m) = \overline{V}(m) - \overline{V}(m-1) + \overline{E}(m) \tag{9}$$

5     The sea ice production includes both thermodynamic ice growth as well as ice thickening caused by ice dynamics, i.e., ridging and rafting. Figure 12c shows the difference in sea ice production/melting $\Delta\overline{B}$ between the "weak" experiments and the baseline simulation. The difference of the monthly time series has high variability and is very noisy and we therefore apply a 5-year running mean to evaluate the long-term changes. A positive $\Delta\overline{B}$ means that more ice is produced (thermodynamically and dynamically) for the "weak" experiments, a negative $\Delta\overline{B}$ the opposite. The smoothed $\Delta\overline{B}$ here represents the net ice 10   production difference including both ice growth and melting and both processes can change $\Delta\overline{B}$.

    For the first 6 years of the integration, the sea ice production $\overline{B}$ for both "weak" experiments increases compared to the baseline but this increase is much stronger for the $0.3P_0^*$ experiment (red). For both "weak" integrations $\Delta\overline{B}$ stays positive until 2001. For the rest of the model integration period, the mean ice production for the "weak" experiments are lower than the baseline. The mean difference in sea ice production for the complete time series is small (identical for $0.7P_0^*$ and $-5\%$ for 15   $0.3P_0^*$ compared to the baseline).

    At the beginning of the integration, when all experiments have the same ice thickness distribution, more thick sea ice can be produced by ice deformation in the "weak" experiments. This causes the ice production $\overline{B}$ to increase compared to the baseline. $\overline{B}$ is corrected for the influence of ice export $\overline{E}$ (eq. 9), which, however, does not change much from the baseline integration during the first two years (not shown). After the strong increase in ice thickness for the "weak" experiments during 20   the first about 8 years (Figure 12a), sea ice formation by ice deformation gets reduced as well as the thermodynamic growth of sea ice. Therefore the difference in ice production decreases again and even gets negative during the second half of the model integration period (red shaded area in Figure 12c).

    Steele et al. (1997) find an increase in ice volume for "weak" $P^*$ experiments simulating 7 years and attribute the thickenning mainly to dynamic ice thickening due to circulation changes. This is in agreement with our findings where the Arctic Basin ice 25   production $overlineB$ is larger for the "weak" experiments for the first 10 years and contributes to the ice volume increase. Only after 10 years the ice production difference gets negative, which is outside the Steele et al. (1997) simulation period of 7 years.

    At the end of the model run, after 18-years, the cumulative sea ice production

$$\Sigma\overline{B} = \sum_{t=1}^{m} \overline{B}(t)$$

30   for the $0.3P_0^*$ experiment is 5% $(-2600\,\mathrm{km}^3)$ lower than the baseline. This reduction in ice production counteracts the increase in ice volume caused by the reduced ice volume export for the $0.3P_0^*$ experiment discussed in the next Section 4.3. For the $0.7P_0^*$ experiment, $\Sigma\overline{B}$ is about equal to that of the baseline simulation. These differences in $\Sigma\overline{B}$ are small compared to the

total differences in Arctic Basin sea ice volume $\overline{V}$ between the three experiments (Figure 12a). Therefore, changes in sea ice export also have to contribute to the sea ice volume difference, which is discussed in the next section.

## 4.3 Sea Ice Export Out of the Arctic Basin

The sea ice export out of the Arctic Basin is smaller for the "weak" experiments than for the baseline. Similar to sea ice volume,
the yearly mean sea ice export $\overline{E}$ for the "weak" experiments also diverges from the baseline and for the $0.3P_0^*$ experiment this difference increases over time (not shown). Even more pronounced is the change in seasonal cycle of the volume export. The monthly mean sea ice volume export during 1992–2009 for the baseline integration is $260\,\mathrm{km}^3$/month. Figure 12b shows the difference in the seasonal cycle of the sea ice export between the two "weak" experiments and the baseline. The seasonal cycle is enhanced for weaker ice: the sea ice export $\overline{E}$ is larger than the baseline during summer months for both the $0.7P_0^*$ and
$0.3P_0^*$ experiments (blue shaded area), and during winter, $\overline{E}$ is lower than the baseline for both experiments (red shaded area). The much lower ice export during winter for the $0.3P_0^*$ experiment causes a 17% decrease in the net annual sea ice export. For the $0.7P_0^*$ experiment the summer increase and winter decrease in $\overline{E}$ nearly balance in the course of one year and this results in a net annual decrease in $\overline{E}$ by only $1.5\%$.

These above results are one example of the non-linear behavior between (a) ice strength and (b) sea ice volume and export.
Intuitively one might expect an increase of ice export for weaker ice even during winter since the ice speed increases. However, during both summer and winter, the ice area export (not shown) is smaller for both "weak" experiments. Along the Arctic Basin boundaries the reduction in $P^*$ alone therefore does not favor ice export. The increase in ice thickness will partly compensate for the reduction in $P^*$ in the weak experiments (see Equation 8). However, it also shows the nonuniform behavior in ice strength $P$, which, e.g., can cause ice arching north of the ice exit gateways or lead to a change of the ice circulation pattern
(see also Steele et al., 1997). Parts or a combination of these effects can cause the lower ice export during winter for the "weak" experiments.

During summer the difference in sea ice area export between the "weak" experiments and the baseline is minimal, which suggests that already in the baseline simulation the ice along the basin boundaries was close to a free drift state and a further reduction in $P^*$ has little effect on the drift. Together with the increased ice thickness for the "weak" experiments, however,
this leads to an increased ice volume export during summer as shown in Figure 12b.

Overall, the decrease in ice export $\overline{E}$ for both "weak ice" experiments explains most of the sea ice volume increase in the Arctic Basin shown in Section 4.1. The cumulative sea ice export from the beginning of the model integration until month $m$ is defined as:

$$\Sigma\overline{E} = \sum_{t=1}^{m} \overline{E}(t)\,.$$

The difference in $\Sigma\overline{E}$ between the baseline and "weak" experiments at the end of the model integration in 2009 ($m = 216$) is $-790\,\mathrm{km}^3$ for the $0.7P_0^*$ experiment and $-9300\,\mathrm{km}^3$ for the $0.3P_0^*$ experiment. This cumulative reduction in ice export will increase the ice volume in the Arctic Basin for the "weak" experiments as it was found in Section 4.1. For the $0.3P_0^*$ experiment, the decrease in $\Sigma\overline{E}$ is even larger than the increase in Arctic Basin sea ice volume of $6700\,\mathrm{km}^3$ at the end of the

integration. For the $0.7P_0^*$ experiment, it accounts for 90% of the $870\,\mathrm{km}^3$ ice volume increase. However, we also have to take changes in sea ice production/melting into account as was discussed in the last section 4.2. Together they explain the observed sea ice volume difference for the "weak" experiments. Be reminded again that sea ice production here does not only refer to thermodynamic ice growth but also to thickening caused by changed ice dynamics, i.e., circulation changes.

## 4.4 Conclusion of Model Ice Strength Sensitivity

The cumulative differences of ice production $\Sigma\overline{B}$ and ice export $\Sigma\overline{E}$ at the end of the time series together with the mean and maximum differences are summarized in Table 4. The results suggest that the largest part of the difference in sea ice volume $\overline{V}$ in Figure 12a is caused by the changed sea ice export $\overline{E}$, especially for the $0.3P_0^*$ experiment. It also, however, becomes evident that changes in ice strength can cause large changes in ice production (Figure 12c). The increased ice speeds and changed sea ice circulation for the "weak" experiments initially cause an ice thickening mainly due to enhanced ice dynamics. After the ice got thicker the thermodynamic ice growth gets reduced, which reduces the net ice production again. In the particular case of the $0.3P_0^*$ experiment, the increase in sea ice production during the first 10 years of simulation is followed by a decrease, which happens to approximately cancel out volume change over the complete 18-year simulation period.

The examples presented in this section highlight the importance of sea-ice deformation processes in a coupled ocean-sea ice model. By changing the sea ice strength parameterization and thus the overall sea-ice deformation in the Arctic Basin, large changes in the sea ice volume, ice export, and ice production can be observed in the model simulations. These changes can be attributed to enhanced sea ice dynamics, which cause a stronger seasonal cycle in sea ice production and ice export. Thus sea ice dynamics, including ice deformation processes, should be adequately represented in a sea ice model if the overall sea ice mass balance is to be simulated realistically. We changed the ice strength parameter $P^*$ within the range used by current model setups. As there is not any true real-world equivalent of $P^*$ the best value for $P^*$ is hard to determine and highly uncertain. This uncertainty could well dominate the uncertainties in thermodynamic parameters, e.g., ice albedo (Steele et al., 1997). The $P^*$ parameter used for our baseline experiment was obtained from an optimized model solution constrained by observations (Nguyen et al., 2011).

## 5 Summary and Concluding Remarks

A realistic representation of sea-ice deformation in models is important for accurate simulation of the sea ice mass balance. Sensitivity experiments show a strong dependence of simulated sea ice volume in the Arctic Basin on the sea ice strength parameterization of the model. A weakening of the simulated sea ice increases the ice deformation rate and drift speed. For the same atmospheric surface boundary conditions, a weakened, more deformable sea ice cover produces a different, in our case increased, Arctic Basin sea ice volume state (Section 4.1). This volume increase is caused by a combination of dynamic and thermodynamic processes. In our sensitivity experiments, a weaker ice cover produces more ice volume due to increased deformation and new ice growth in leads. The thickening of the ice, however, increases ice strength and decreases sea ice volume export out of the Arctic Basin compared to the baseline experiment (Section 4.3). Thermodynamic growth is also

reduced for thicker ice. The balance of these processes leads to a new equilibrium Arctic Basin ice volume after about eight years of simulation.

Multiple equilibrium flow states (i.e., when ice growth equals ice export) can exist for the Arctic Basin, and their characteristics are influenced by sea ice strength and ice rheology (Hibler et al., 2006). For the last decade of simulation, all sensitivity experiments show a decrease in ice volume, which is consistent with the observed sea ice volume loss (Kwok and Rothrock, 2009; Nguyen et al., 2011). The changing Arctic sea ice volume in the simulations is caused both by changes in sea ice export (Section 4.3) as well as in sea ice production and melting (Section 4.2). The reduced sea ice export, however, makes up the largest part of the volume change. These model sensitivity experiments therefore illustrate the importance of properly representing sea-ice deformation in sea ice models in order to accurately simulate the overall sea ice mass balance.

Deformations in Arctic ocean and sea ice simulations with horizontal grid spacing of 18, 9, and 4.5 km were compared to RGPS satellite observations during the 1992–2008 period (Section 3). Lagrangian sea ice drift was reconstructed from the three model solutions for a direct comparison with the RGPS data. Sea ice strain rate divergence, vorticity, and shear were calculated in the same way for the three simulations and for satellite observations from the Lagrangian ice drift datasets. Even though the viscous-plastic dynamic sea ice model with elliptical yield curve is able to produce what appears to be linear kinematic features (LKFs), the orientation and spatial density of these LKFs are very different from what is observed in the RGPS data. For the 4.5 km simulation, however, many more and more confined LKFs are visible compared to the two lower resolution simulations. A small model grid spacing seems to be essential to represent LKFs using a VP sea ice rheology. The mean sea-ice deformation rate, however, is between 51% to 57% lower in all simulations than in the RGPS data. The largest difference occurs for the magnitude of divergence, which is 67% to 79% too low (Table 3). Also the large-scale shear pattern is not well reproduced in the model solutions (Figure 4). In addition the LKFs occur less frequently in the simulations. Of the three model solutions, the one with the smallest grid spacing of 4.5 km has characteristics closest to RGPS observations.

While RGPS sea-ice deformation data show a clear discrimination between the thinner seasonal sea ice with more deformation and the thicker perennial sea ice, the model deformation zones are mainly confined to a few LKFs at the ice margins. Differences are largest for seasonal sea ice, where the model strongly underestimates sea-ice deformation. This suggests a shortcoming of the ice rheology, for example, the linear dependence between ice strength and ice thickness. Model solutions with smaller grid spacing, however, result in more small-scale deformation features. In particular, the 4.5-km simulation has more LKF-like features in the Central Arctic than the coarser-resolution simulations and, visually, the spatial distribution of these LKF-like features agrees better with RGPS observations. This improved realism is evaluated by computing the percentage $Q$ of sea ice area containing the highest 15% of sea-ice deformation rates, which is a measure of how confined the deformation processes are. For this metric, the 4.5-km model solution performs closest to the RGPS data, with a $Q$ value of 0.5% compared to $Q = 0.7\%$ for RGPS, while the 9 and 18-km simulations have higher $Q$ values of 1.2 and 1.5%, receptively, i.e., the deformation features are much less confined. These differences in small-scale deformation features can be important because ocean-to-atmosphere heat transfer tends to occurs on small scales. For example, the heat flux from narrow leads can be twice as high as that from larger leads (Marcq and Weiss, 2012) and ocean upwelling events caused by sea ice shear motion happen on small scales (McPhee et al., 2005).

In Section 3.2 we compare the power-law scaling behavior of the three model solutions with the RGPS observations. Both the RGPS data and all model solutions show a clear power-law dependence of the absolute divergence $|\dot{\nabla}|$ to the length scale $L$. The scaling exponent $b$ for the RGPS data, however, is with $-0.2$ about twice as negative as the $b \approx -0.1$ for the three model solutions. The power-law scaling of the probability density functions (PDF) of the absolute divergence for the three model solutions is very similar to the RGPS data, especially during winter where the models and RGPS exhibit the same power law exponent $n \approx -3$. During summer the PDFs of the three model solutions get more different but, however, still follow a power law. Neither for the spatial scale dependence nor for the PDFs we do observe the strong divergence from power-law scaling for the VP sea ice rheology reported by Girard et al. (2009).

We tested if the power-law dependence can be used to compare deformation rates obtained with model simulations using different grid spacings. The scaling of the deformation rate in our three model solutions with different grid spacing, i.e., different length scales follows a similar power law as is observed for the RGPS observations (Section 3.2.3). The power law exponent $b$ for the complete model domain is $-0.54$. If the domain is restricted to the area covered by RGPS observations, consisting of compact, thick ice, the exponent becomes with $b = -0.2$ very similar to what is observed for the RGPS data. The power law scaling exponent strongly depends on ice concentration and thickness, i.e., the internal ice stress. In most cases it therefore will not be possible to compare absolute numbers of strain rates obtained from models with different grid spacing.

On larger scales the sea-ice deformation rate of all three model solutions is very similar, with only small improvements for the 4.5-km simulation (Figure 5). Almost independent of grid spacing, the modeled sea-ice deformation is much lower than the RGPS observations ($\sim 50\%$). Bouillon and Rampal (2015a) suggest that RGPS deformation rates are too high due to artificial noise in the motion fields, which could explain part of this difference. Nevertheless, the large scale pattern of divergence(Figure 2) and vorticity (Figure 3), but not shear (Figure 4), are reproduced by all model simulations. Even if the differences are small for the large scale deformation patterns, the 4.5-km simulation, the one with the smallest horizontal grid spacing, always performs best out of the three solutions. This difference becomes more pronounced if small scale deformation features are considered. The 4.5-km simulation is the only one that reproduces a reasonable number of LKFs in the Central Arctic, even on length scales ($2\times$ the grid spacing) where the lower resolution models theoretically are capable of reproducing these features. We conclude that increasing the spatial model resolution can improve the sea-ice deformation representation for a viscous-plastic sea ice rheology. However, big differences to the observed sea-ice deformation strain rates still remain.

An interesting future study would be to attempt to adjust sea ice and ocean model parameters in order to reproduce the metrics discussed in this paper. For example, in a separate sensitivity experiment, not discussed in this manuscript, we changed the sea ice strength dependence on sea ice thickness (Equation 8) from linear to cubic, which considerably increased deformation rate in both perennial and seasonal ice zones. Of course, adjusting a single parameter can improve a certain set of model features but is likely to make others, e.g., sea ice velocity, worse. What is needed is the simultaneous adjustment of several key model parameters, in the manner discussed in Menemenlis et al. (2005) and Nguyen et al. (2011). Other possible approaches for improving the representation of sea ice strain rates include the introduction of multiple categories for different ice thicknesses and deformed and undeformed ice, since multicategory models allow weaker resistance, more leads, and enhanced ice growth

(Mårtensson et al., 2012); and experimentation with new ice rheologies that do not rely on the viscous-plastic assumptions (Sulsky et al., 2007; Girard et al., 2011; Tsamados et al., 2013; Bouillon and Rampal, 2015b).

*Acknowledgements.* The constant development and support for the MITgcm from the people involved with mitgcm.org is acknowledged. We thank Martin Losch and Pierre Rampal for constructive discussions about the model and sea-ice deformation analysis. This work was performed at the Jet Propulsion Laboratory, California Institute of Technology under contract with the National Aeronautics and Space Administration (NASA). Parts of this research has been supported by the Institutional Strategy of the University of Bremen, funded by the German Excellence Initiative. High End Computing resources were provided by the NASA Advanced Supercomputing Division.

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

**Figures**

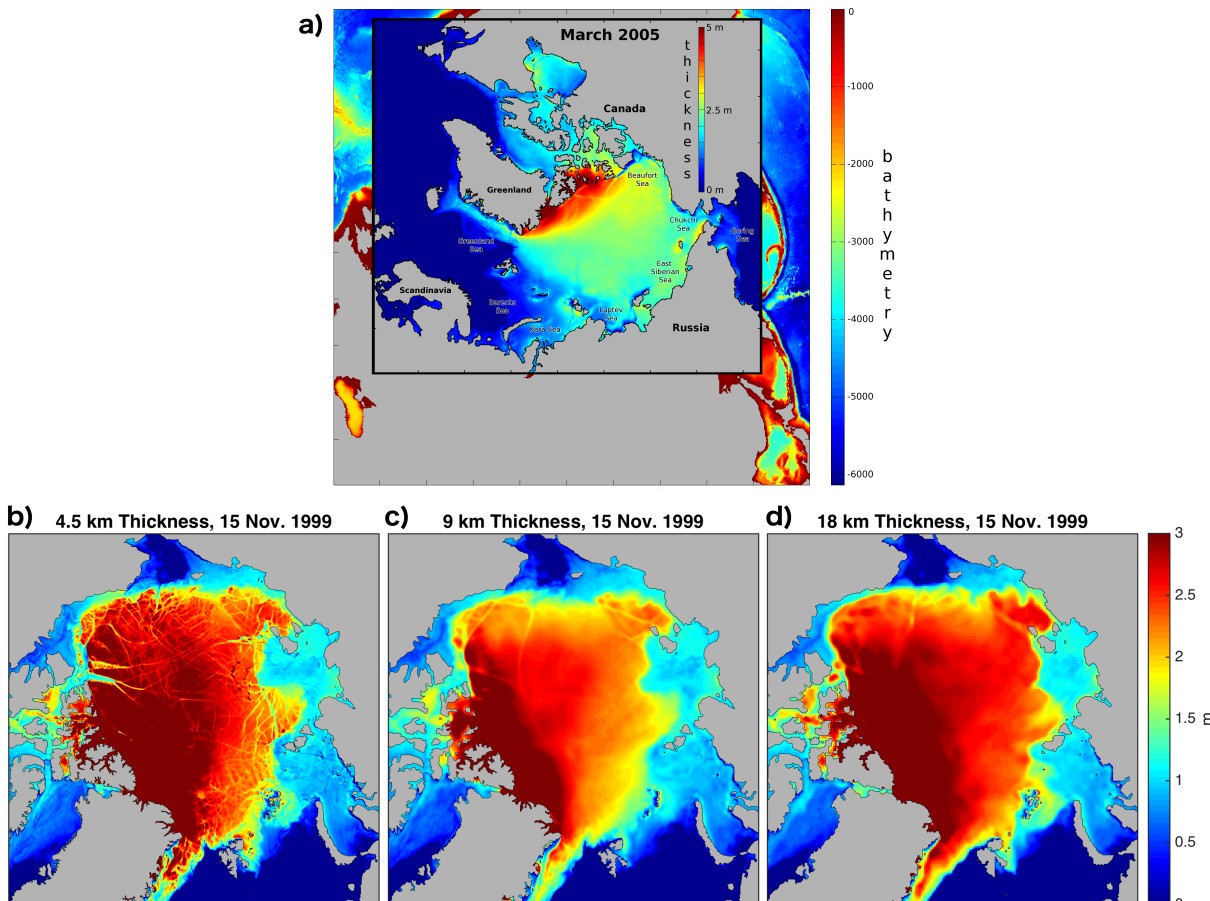

**Figure 1.** a) The Arctic face of the cube sphere grid used by the ECCO2 project. The March 2005 ice thickness inset shows the regional grid used in this study. Note that North Pacific coastline in the regional grid is modified relative to the global set-up in order to remove unconnected seas. Boundary conditions are obtained from the ECCO2 18-km cube sphere solution. b) – d) Sea ice thickness on 15 November 1999 after about 8 years of model integration for the 4.5, 9, and 18 km simulation, respectively. Sea ice thickness is shown on the respective original model grid.

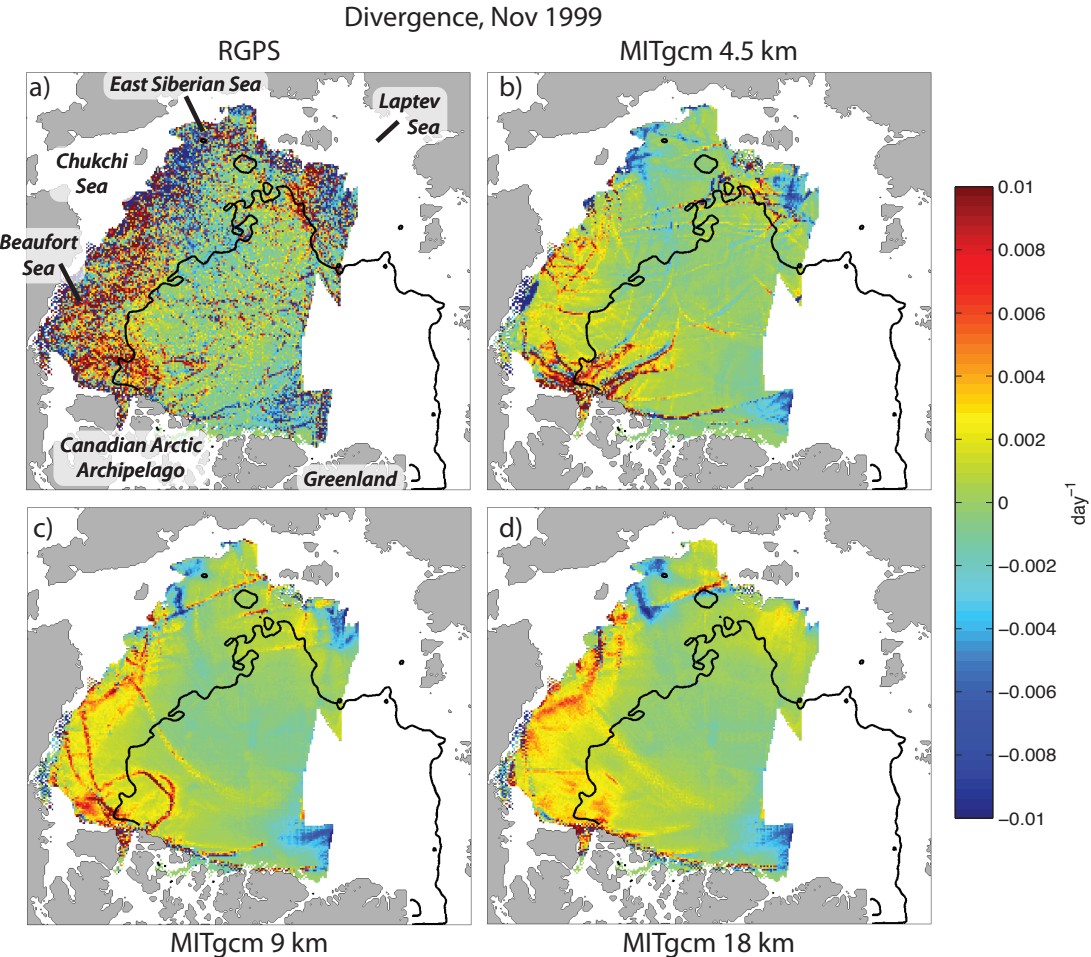

**Figure 2.** Examples of monthly mean November 1999 sea ice divergence. The divergence from (a) RGPS and the model runs with (b) 4.5-km, (c) 9-km, and (d) 18-km grid spacing are shown. The number of LKFs increases with decreasing model grid spacing. All maps are shown on the same 12.5 km grid and are constructed from the same number of observations (see Section 2.3). The black line discriminates seasonal and perennial sea ice. White areas are not covered by RGPS observations.

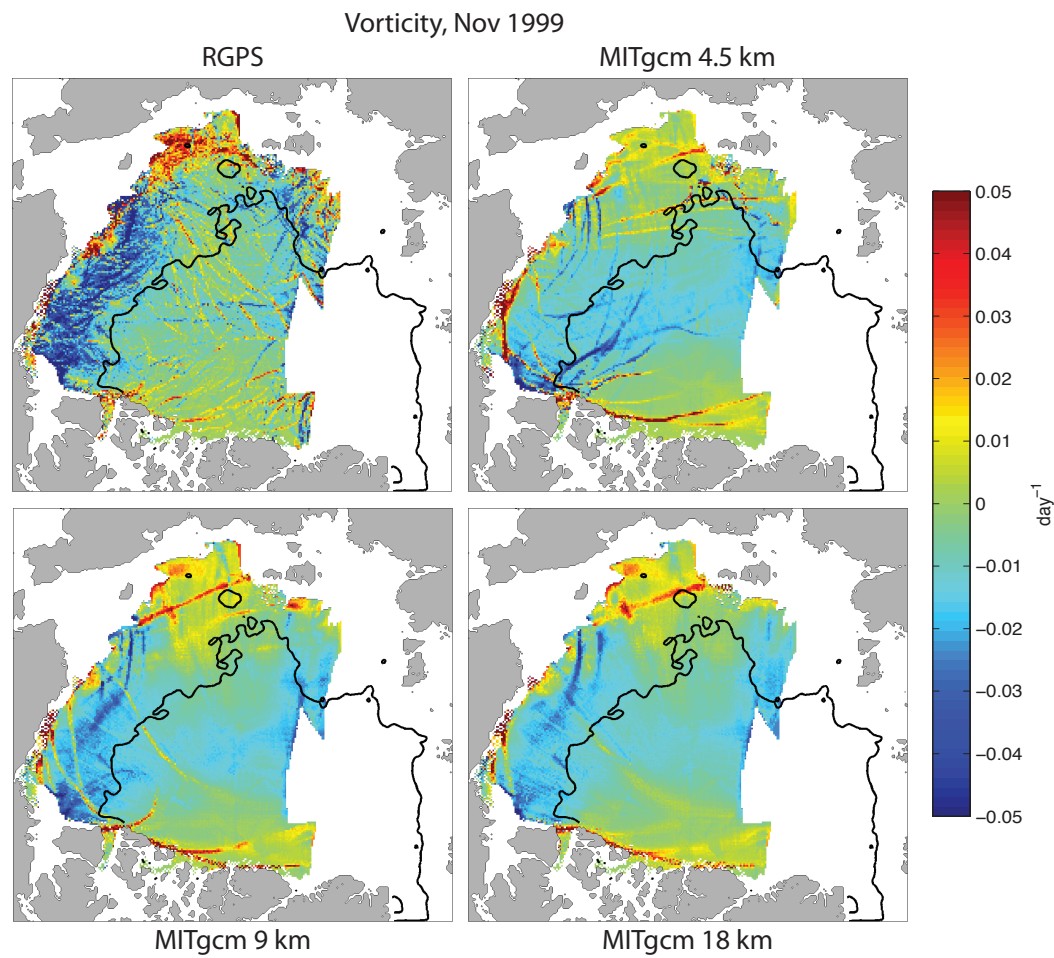

**Figure 3.** As Figure 2 but for vorticity.

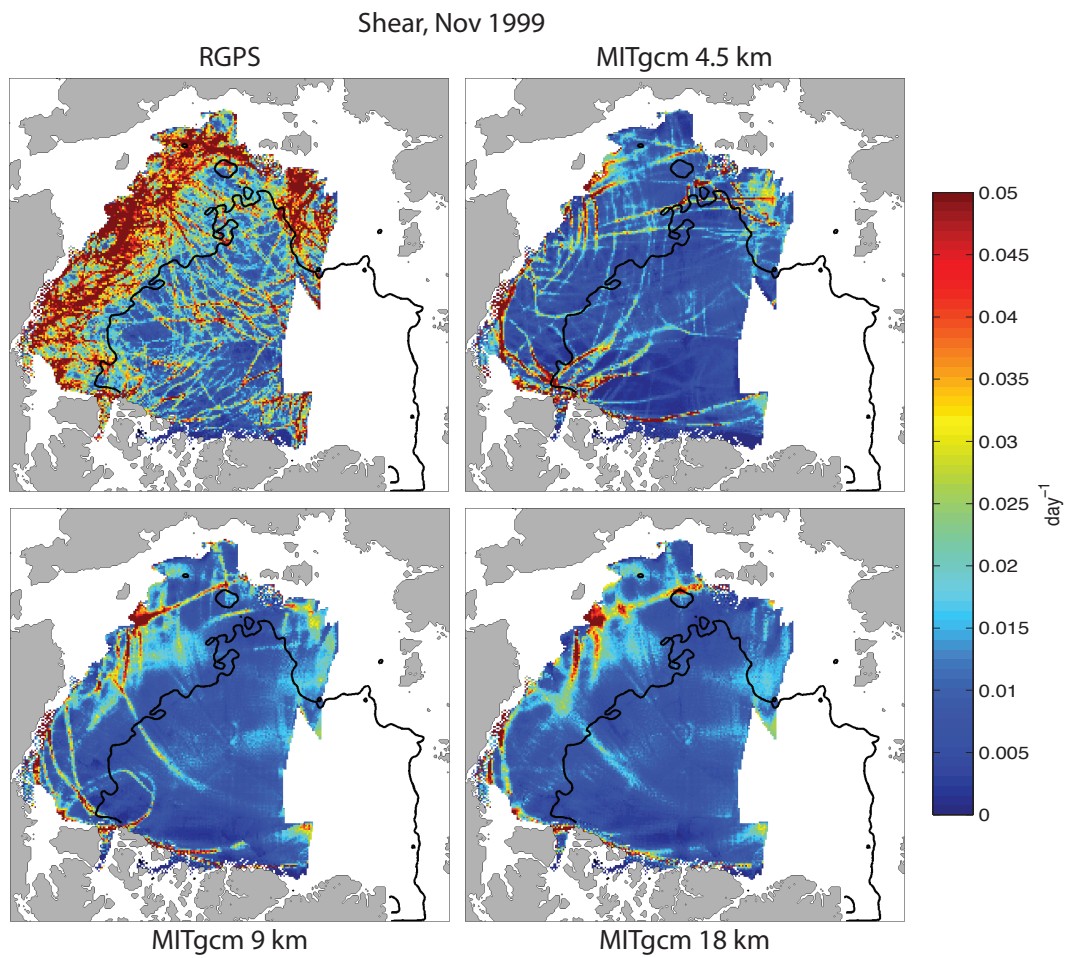

**Figure 4.** As Figure 2 but for shear.

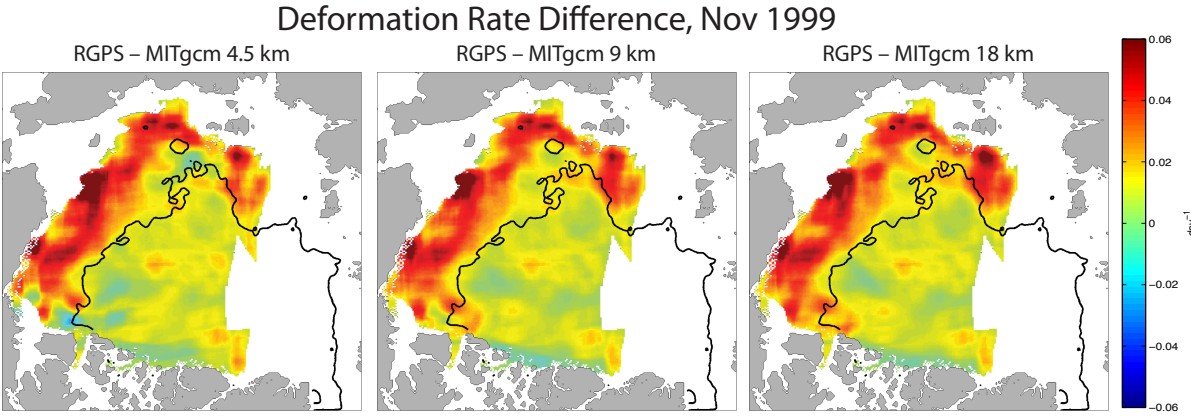

**Figure 5.** Smoothed (150 km) difference in deformation rate $\dot{D}$ between RGPS and model solutions with $4.5\,$km (left), $9\,$km (middle), and $18\,$km (right) grid spacing. Largest differences occur in the seasonal ice zone outside the black contour.

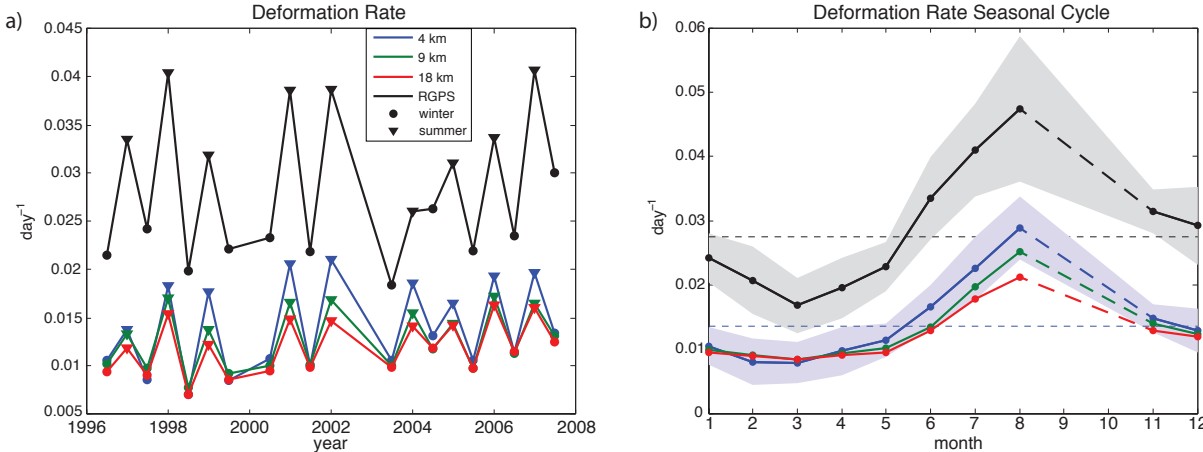

**Figure 6.** a) Mean deformation rate $\dot{D}$ for all 20 RGPS periods and the corresponding modeled values. Circles mark winter periods and triangles summer periods; note that periods have different length (see Table 2). b) Seasonal cycle of $\dot{D}$; shaded areas show standard deviations for RGPS and the $4.5$-km solution (9 and 18-km solutions are similar); horizontal dashed lines show the mean calculated from the monthly time series; note that no data is available for September and October.

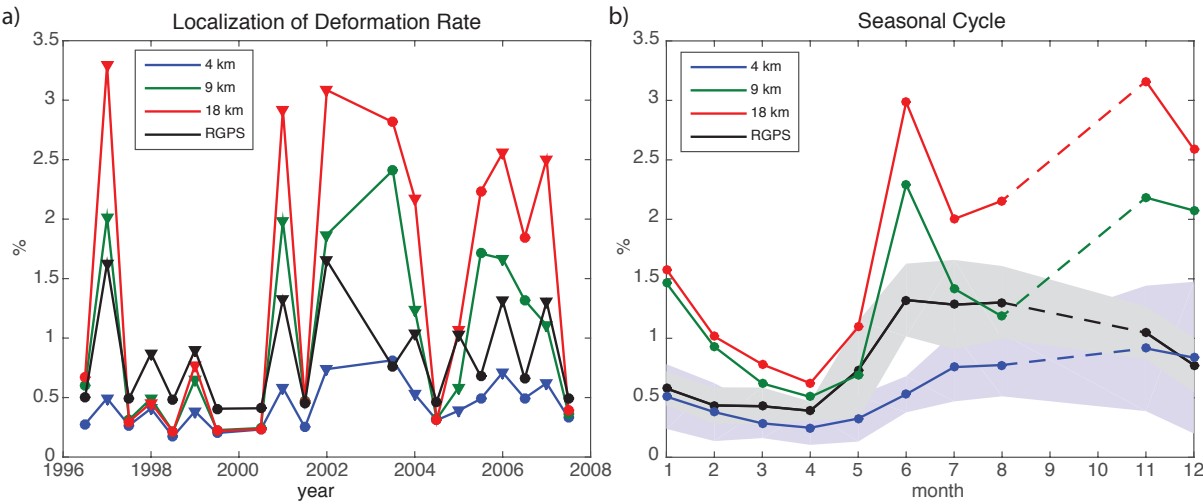

**Figure 7.** The percentage $Q$ of area containing the highest 15% of all sea-ice deformation rates shows the localization of deformation. a) Time series showing the absolute percentage $Q$ for RGPS data (black) and model solutions with 4.5 (blue), 9 (green), and 18 km (red) grid spacing for all 20 RGPS periods. Circles mark winter periods and triangles summer periods; note that periods have different length (see Table 2). b) Seasonal cycle of $Q$; shaded areas show standard deviations for RGPS and the 4.5-km solution (9 and 18-km solutions are similar); note that no data is available for September and October.

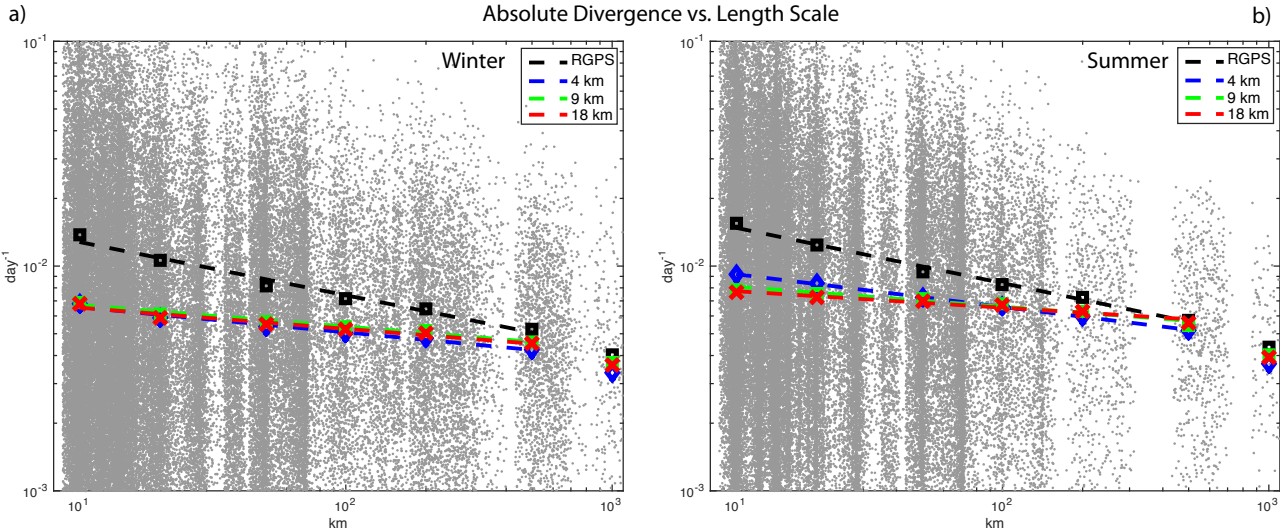

**Figure 8.** Scaling properties of absolute sea-ice divergence $|\dot{\nabla}|$ for RGPS and model solutions for all a) winter (Nov–Apr) and b) summer (May–Jun) periods. For length scales of 10, 20, 50, 100, 200, 500, and 1000 km the ice divergence from the Lagrangian cells were aggregated over 5-day periods. Individual data points for the RGPS dataset are shown in grey. Mean values for the six different length scales are marked with symbols. Dashed lines are least square fits to the five mean values from 10 to 500 km. Note the logarithmic axes scaling.

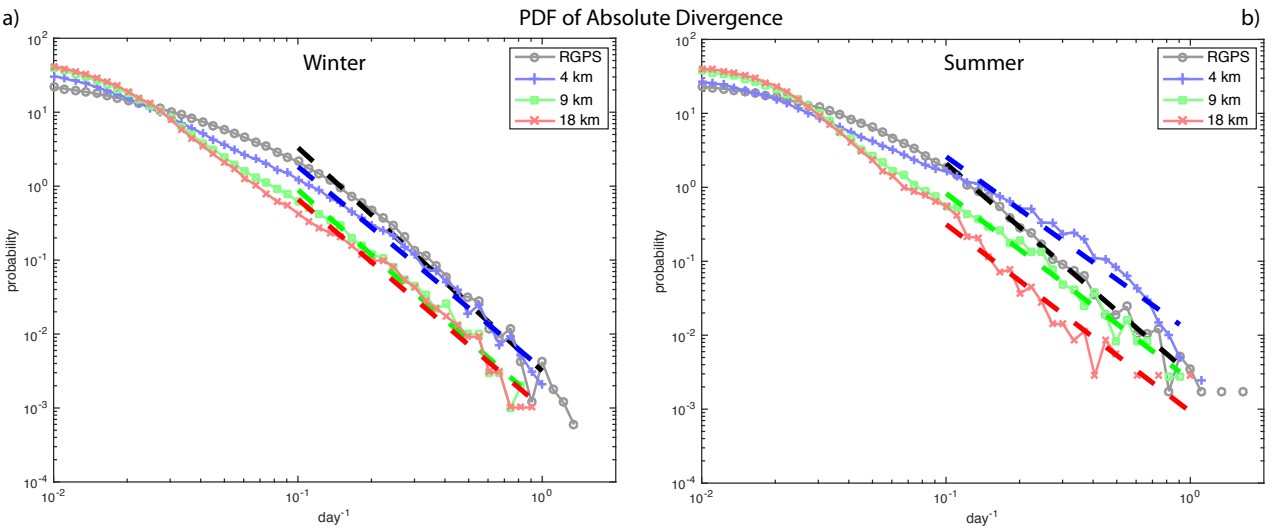

**Figure 9.** Probability density function of absolute sea-ice divergence $|\dot{\nabla}|$ for a length scale of 20 km based on 5-daily aggregated Lagrangian cells for all a) winter (Nov–Apr) and b) summer (May–Jun) periods for RGPS and model solutions. Dashed lines are least square fits to the approximately linear part of the PDFs between 0.1 and $1.0\,\mathrm{day}^{-1}$. Note the logarithmic axes scaling.

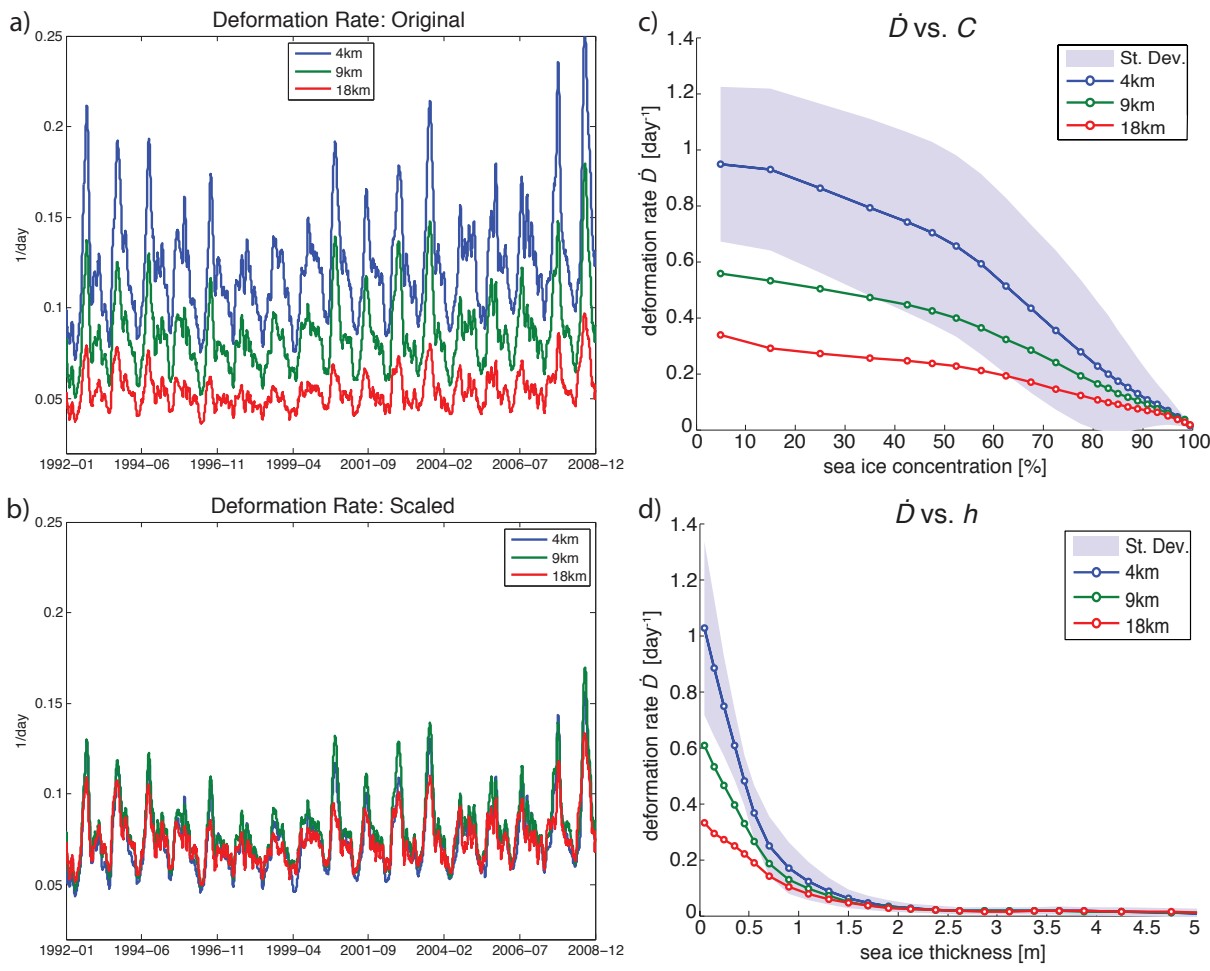

**Figure 10.** a) Time series 1992–2008 of mean deformation rate $\dot{D}$ in the complete model domain (see Fig. 1) for model runs with 4.5 (blue), 9 (green), and 18 km (red) grid spacing. b) as a) but for deformations normalized to a 10 km scale using equation 6 with $b = -0.54$. All curves are one month running means. c) and d) show, respectively, the dependence of sea-ice deformation rate $\dot{D}$ on sea ice concentration $C$ and sea ice thickness $h$ for the three model integrations. Blue shaded color areas mark $\pm$ one standard deviation for the 4.5 km solution.

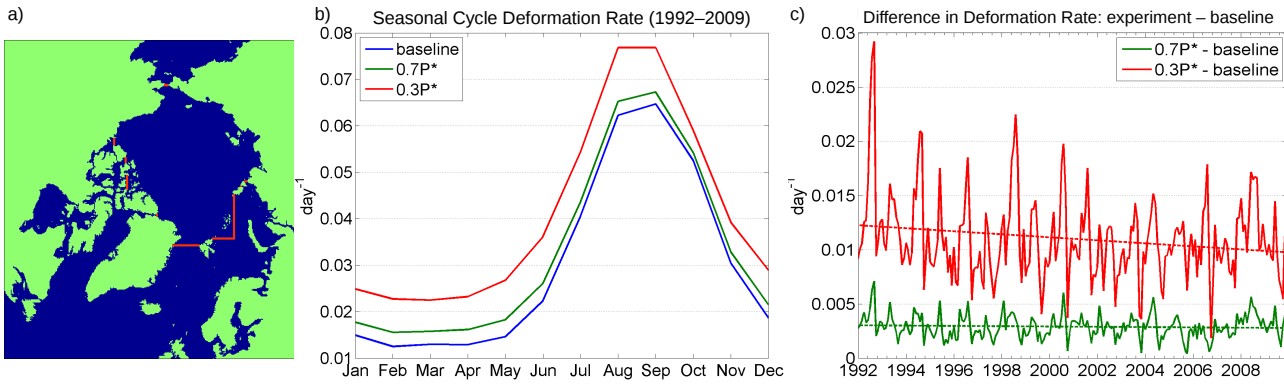

**Figure 11.** a) Red lines mark the Arctic Basin boundaries and flux gates used. b) Seasonal cycle (1992–2009) of mean monthly deformation rate $\overline{D}$ within the Arctic Basin for the three experiments baseline in blue, $0.7P_0^*$ in green, and $0.3P_0^*$ in red. c) Difference in $\overline{D}$ for $0.7P_0^* -$ baseline in green and $0.3P_0^* -$ baseline in red.

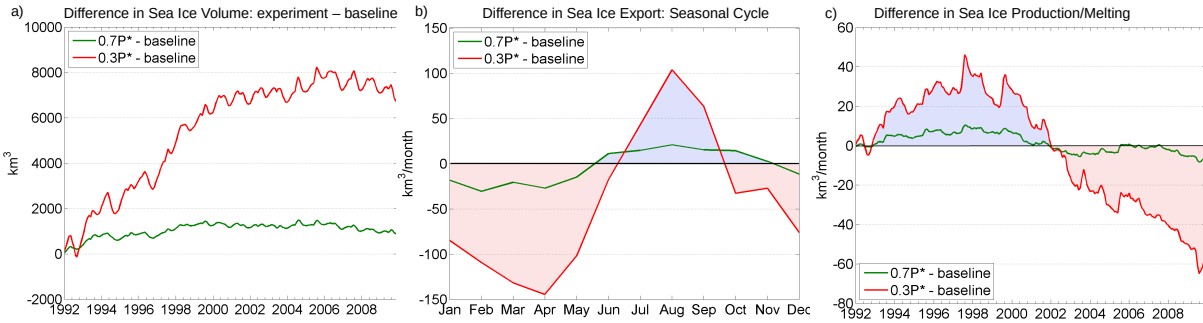

**Figure 12.** Differences between simulations for "$0.7P_0^* -$ baseline" in green and "$0.3P_0^* -$ baseline" in red. a) Difference in Arctic Basin sea ice volume, b) difference in seasonal cycle of sea ice export, and c) difference in sea ice production/melting (5-year running mean).

# Tables

**Table 1.** Selected sea ice model parameters

| | | |
|---|---|---|
| Atmospheric forcing | JRA-25 | |
| Sea ice dry albedo | 0.7 | |
| Sea ice wet albedo | 0.71 | |
| Snow dry albedo | 0.87 | |
| Snow wet albedo | 0.81 | |
| Ocean albedo | 0.16 | |
| Air/sea ice drag coefficient | 0.0011 | |
| Ocean/sea ice drag coefficient | 0.0054 | |
| Ice strength parameter $P^*$ | 23 | kN/m$^2$ |
| Lead closing parameter $H_o$ | 0.6 | |
| elliptical yield curve major to minor axis ratio $e$ | 2 | |

**Table 2.** RGPS periods used in this study. Column 3 gives the number of monthly mean values used.

| start date | end date | no. months | season |
|---|---|---|---|
| 1996-11-07 | 1997-06-01 | 7 | winter |
| 1997-05-18 | 1997-08-01 | 2 | summer |
| 1997-11-02 | 1998-06-01 | 7 | winter |
| 1998-05-10 | 1998-09-01 | 2 | summer |
| 1998-10-28 | 1999-05-17 | 6 | winter |
| 1999-05-08 | 1999-09-01 | 4 | summer |
| 1999-11-01 | 2000-05-14 | 7 | winter |
| 2000-11-04 | 2001-06-01 | 7 | winter |
| 2001-05-15 | 2001-09-01 | 3 | summer |
| 2001-11-05 | 2002-06-01 | 7 | winter |
| 2002-05-16 | 2002-08-01 | 2 | summer |
| 2003-12-04 | 2004-06-01 | 6 | winter |
| 2004-05-11 | 2004-09-01 | 3 | summer |
| 2004-11-10 | 2005-06-01 | 7 | winter |
| 2005-05-15 | 2005-09-01 | 3 | summer |
| 2005-11-29 | 2006-06-01 | 6 | winter |
| 2006-05-19 | 2006-09-01 | 3 | summer |
| 2006-12-03 | 2007-06-01 | 6 | winter |
| 2007-05-14 | 2007-09-01 | 3 | summer |
| 2007-12-01 | 2008-06-01 | 6 | winter |
| 20 periods (11 winter/9 summer) | | 97 | |

**Table 3.** Overview of some statistical parameters for the complete 97-month time series of RGPS and model sea ice strain rates. All units are $10^{-2}\,\mathrm{day}^{-1}$ if not otherwise indicated; $\pm$ values denote the standard deviation of the time series; 'difference' is the difference between model and RGPS in %; and 'correlation' is the correlation coefficient between the model and RGPS time series. The last four rows summarize the power-law scaling exponents discussed in Section 3.2 (no units, see also equations 6 and 7).

| $\cdot 10^{-2}$ | | RGPS | 4.5 km | 9 km | 18 km |
|---|---|---|---|---|---|
| deformation rate $\dot{D}$ | mean | $2.8 \pm 1.1$ | $1.4 \pm 0.7$ | $1.3 \pm 0.5$ | $1.2 \pm 0.4$ |
| | difference | | $-51\%$ | $-55\%$ | $-57\%$ |
| | correlation | | 0.89 | 0.90 | 0.90 |
| absolute divergence $|\dot{\nabla}|$ | mean | $1.1 \pm 0.5$ | $0.4 \pm 0.2$ | $0.3 \pm 0.2$ | $0.2 \pm 0.1$ |
| | difference | | $-67\%$ | $-77\%$ | $-79\%$ |
| | correlation | | 0.85 | 0.87 | 0.86 |
| absolute vorticity $|\dot{\tau}|$ | mean | $2.3 \pm 0.7$ | $1.4 \pm 0.6$ | $1.3 \pm 0.4$ | $1.2 \pm 0.4$ |
| | difference | | $-40\%$ | $-43\%$ | $-47\%$ |
| | correlation | | 0.82 | 0.84 | 0.82 |
| shear $\dot{\zeta}$ | mean | $2.4 \pm 0.9$ | $1.3 \pm 0.6$ | $1.2 \pm 0.5$ | $1.1 \pm 0.4$ |
| | difference | | $-47\%$ | $-50\%$ | $-53\%$ |
| | correlation | | 0.89 | 0.90 | 0.90 |
| percentage $Q$ of area | mean | $0.7 \pm 0.4\%$ | $0.5 \pm 0.4\%$ | $1.2 \pm 1.1\%$ | $1.5 \pm 1.4\%$ |
| containing highest 15% | difference | | $-36\%$ | $65\%$ | $122\%$ |
| of deformation rates | correlation | | 0.59 | 0.60 | 0.75 |
| spatial scaling exponent $b$ for | winter | $-0.24 \pm 0.05$ | $-0.11 \pm 0.02$ | $-0.10 \pm 0.02$ | $-0.09 \pm 0.02$ |
| absolute divergence $|\dot{\nabla}|$ | summer | $-0.24 \pm 0.03$ | $-0.146 \pm 0.004$ | $-0.09 \pm 0.03$ | $-0.07 \pm 0.02$ |
| PDF scaling exponent $n$ for | winter | $-3.0 \pm 0.1$ | $-2.74 \pm 0.08$ | $-2.9 \pm 0.1$ | $-2.8 \pm 0.1$ |
| absolute divergence $|\dot{\nabla}|$ | summer | $-2.84 \pm 0.09$ | $-2.4 \pm 0.1$ | $-2.52 \pm 0.09$ | $-2.5 \pm 0.2$ |

**Table 4.** Difference in Arctic Basin sea ice volume $\Delta \overline{V}$, cumulative ice export $\Delta \Sigma \overline{E}$, and cumulative sea ice production $\Delta \Sigma \overline{B}$ between the two experiments and the baseline.

| in km$^3$ | End (Dec 2009) | | Mean | | Max. | |
| | $0.7P_0^*$ | $0.3P_0^*$ | $0.7P_0^*$ | $0.3P_0^*$ | $0.7P_0^*$ | $0.3P_0^*$ |
|---|---|---|---|---|---|---|
| $\Delta \overline{V}$ | 870 | 6700 | 1050 | 5480 | 1490 | 8230 |
| $\Delta \Sigma \overline{E}$ | -790 | -9300 | -660 | -4340 | -940 | -9460 |
| $\Delta \Sigma \overline{B}$ | 80 | -2600 | 390 | 1140 | 810 | 3130 |