# Peer review of "Sea-Ice Deformation in a Coupled Ocean-Sea Ice Model and in Satellite Remote Sensing Data"

_The Cryosphere, 2016_

## Editor Comment (EC1) · D. Notz (Editor) · 1 Apr 2016

A comment on this paper has been removed today because it contained inappropriate language.

---

## Referee Comment (RC2) · Anonymous Referee #1 · 4 Apr 2016

**1   General comments**

This paper is essentially split into two parts. The first part discusses how modifications of the model parameter $P^*$ affect modelled sea-ice volume, export, and production and melt. The second part analyses the modelled sea-ice deformation in differently resolved model runs, comparing it to observations using the RGPS set of observations. The two parts are poorly linked, even though the authors do point out that such a link is possible. The splitting of the paper into two parts like this is cause for concern. An immediately obvious way to link the two would be to analyse the deformation patterns of the two low $P^*$ runs used in part one in part two as well. With the current set-up I would recommend splitting the paper in two and expanding on each part. As it stands I will review the two parts of the paper independently, since this makes the most sense

to me.

**2  Specific comments**

**2.1  Part one**

In part one the authors consider the effects of decreasing $P^*$ on modelled sea-ice volume, export, and production and melt "to motivate the importance of sea ice deformation for the Arctic sea ice mass balance". I'm not sure the second part really needs this motivation. To me, seeing if we're modelling the deformation correctly is motivation enough. The question of to what extent modelled deformation affects the sea-ice mass balance is also interesting enough on its own. I am, however, not convinced by the approach taken by the authors. There is no comparison to observations or estimates so I don't know whether the normal $P^*$ is even giving a reasonable deformation rate or mass balance, or how changing $P^*$ affects the deformation, other than the deformation rate. We can't compare figures 2b and 8b either, since the d eformation is calculated over different areas for the two. So while we can see that changing $P^*$ does affect the deformation rate we don't know how it affects various other properties of the deformation. It is therefore also not clear (to me at least) that $P^*$ is an appropriate tuning knob to get the deformation rate right. It is a possible one, but more work is needed to show that it is an appropriate one. Also, what happens to the deformation rate once the model has spun up properly after changing $P^*$? This is not clear, since figure 2b shows the deformation rate for 1992–2009, which is arguably a period of transient response as discussed below.

This leaves us with nothing much to judge the results of this experiment. It doesn't help that what we're looking at is essentially the model's transient response to a large change in its internal mechanics. Normally one would spin the model up to see the
effect of a lower $P^*$ on a model in equilibrium, but this is not done here. The authors claim that the model has reached a new equilibrium after about 8 years, but the difference in "sea-ice production/melting" is still changing rapidly at the end of the model run (figure 3c). If we knew how the deformation rate changes from 1992 to 2009 and that the model does not capture that, and that tuning $P^*$ correctly would give the right deformation rate, then we could say something about how simulating the wrong deformation rate gives the wrong mass balance, but the manuscript gives none of those building blocks.

In terms of analysis of the low $P^*$ runs the authors also miss what must be in my opinion the most obvious cause for increase in volume, and that is thickness increase due to excessive convergence. This is also pointed out by Steele et al . (1997), who performed a similar experiment. When the ice is artificially weakened (which is what we should consider is happening when using 30% of $P^*$) it can be expected to ridge excessively and pile up at the north-Greenland and Canadian coasts. This effect is completely ignored by the authors, even though Steele et al. (1997) discuss it quite nicely and the authors cite that paper. In particular, the authors state that "[o]verall, the decrease in ice export $\bar{E}$ for both "weak ice" experiments explains most of the sea ice volume increase in the Arctic Basin shown in Section 3.1" —- a statement which seems to contradict the results of Steele et al. (1997) without giving due consideration to the piling up of ice. The pile-up of ice is, in my opinion clearly what causes the increased "sea-ice production" that the authors note in section 3.3. From the text it seems clear that the authors consider the sea-ice production(/melt) to be thermodynamic production, but there is no reason to assume that this is the case. Without considering the ice pile-up the analysis of the difference in "sea-ice production/melting" is deeply flawed.

**2.2 Part two**

Part two of the paper has, in my opinion much more potential than part one. It is really what I was hoping to see when I read the title and agreed to review the paper. In my opinion the title belongs to part two and part one should be relegated to a different paper. In part two the authors compare the results of differently resolved model runs to the RGPS observations. This is a worthy goal and I would be very interested in a more detailed and thorough analysis of the high resolution MITgcm model. This could function as a continuation of the work done by Girard et al. (2009,2011), and a contrast to that done by Bouillon and Rampal (2015b) and Rampal et al. (2015). I know there are a number of people within the sea-ice modelling community who hope and believe that running an (E)VP model at a higher resolution than Girard et al (2009) did will give better results than what they got. It is, therefore particularly interesting to know whether the results of Girard et al. (2009) hold for the 4.5 km resolution and to get an independent verification, or contradiction of the results for lower resolutions, as well as an indication of the resolution dependence.

Unfortunately the current analysis is inferior to that performed by Girard et al. (2009,2011), Bouillon and Rampal (2015b), and Rampal et al. (2015). The authors of the current work mainly base their conclusions on monthly averaged deformation, which is inappropriate, and on visual and qualitative inspection of the simulated and observed deformation fields. They should instead use the quantitative statistical tools and metrics previous authors have used. This would have made for much more solid conclusions and results that are quantitatively comparable to observations (e.g. Marsan et al., 2004 or Stern and Lindsey, 2009) and the model analysis mentioned above.

I want to stress, in particular that using monthly averages when studying deformation is inappropriate, since nearly all of the deformation happens at a much shorter time scale. This is a major problem with section 4.3.1. If the authors want to consider long-term differences in deformation then figure 7 is a more appropriate approach than figures 4,

5, and 6. I would even recommend taking a multi-month or seasonal average instead of only one month, in that case. It is interesting how large the difference in deformation rate is between the seasonal and multi-year ice is.

I'm also left wondering if the deformation rates used in section 4.5 are monthly averages or not. Using monthly averages there would be inappropriate for the same reason as before, although it is not immediately clear how large an error we get using monthly averages in this case. Should the results in section 4.5 hold then they are a very interesting contradiction of the results of Girard et al. (2009). It does seem strange though, that the authors choose not to remove the noise of the RGPS data as prescribed by Bouillon and Rampal (2015a). They need to either remove the noise or justify not removing it.

It is also inappropriate to consider the percentage of area containing 80% of the deformation as a measure of localisation (section 4.3.3). It should be the largest 15% of the deformation, like Stern and Lindsay (2009) use. Using 80% of the deformation you essentially include all the deformation so this is no longer a measure of the localisation of deformation. The way it stands the metric is essentially meaningless.

The authors also do the power law scaling of deformation rate incorrectly (section 4.4). They use different model realisations (i.e. 4.5, 9, and 18 km resolutions) to determine the scaling, but the correct thing to do is to use a coarse graining method (like the authors named above) and calculate the scaling based on it. The authors of this manuscript argue that the high resolution model gives better results than the low resolution ones, but they then combine all three to calculate the scaling. This makes no sense.

**3  Conclusions**

I am sorry to say that I will be recommending that this paper be rejected publication in The Cryosphere. The reasons for this decision are the poor structure of the paper, it being split into two unrelated parts, and the substantial shortcomings of both parts. This is quite disappointing since I believe that the comparison of the MITgcm results with RGPS data could be very interesting indeed. My recommendation to the authors is to thoroughly review Girard et al (2009) and the related literature, and then to revisit part two of the manuscript with the aim to refute or support the conclusions of Girard et al (2009) in the case of the 4.5 km resolution simulation, give an indication of the resolution dependence, and to provide contrast with the results of Bouillon and Rampal (2015b) and Rampal et al (2015). If this is properly done then that would make for an interesting paper and one that would be important for further evaluation and development of dynamical sea-ice models.

**4  References**

All the papers I refer to here are already cited in the paper, with the exception of Rampal et al (2015), which is still under review at The Cryosphere Discussions: http://www.the-cryosphere-discuss.net/tc-2015-127/

---

## Referee Comment (RC3) · Anonymous Referee #1 · 4 Apr 2016

Thank you for the clarification, its very good to have.

W.r.t. your point a) then I'm happy to see that we essentially agree on what is happening in the model. When reading the paper I assumed you meant that increased deformation would result in increased cracking and growth in leads - but what is really happening (I assume!) is that the ice becomes thicker at the Greenland and Canadian coast with more production on the Siberian side. It was really a statement along those lines that I missed.

On point b) - that's good! This makes the results of section 4.3.2 and particularly that of 4.5 very interesting.

---

## Short Comment (SC1) · 4 Apr 2016

We thank the reviewer for raising some points which seem to be unclear in our text. At this point we are not answering to the critics of the review directly but like to clarify two misunderstandings the reviewer, in our view, had and which therefore might be of relevance to other readers.

a) We do not consider the change in sea ice production/melt of the "weak" ice experiment in section 3.3 to be a thermodynamic process (and also do not write that in our opinion). We agree with the reviewer that this a combination of dynamic and thermodynamic effects. We agree with the reviewer and Steele et al. (1997) that dynamic ice thickening due to increased convergence for the weaker ice is causing the increased ice production, especially at the beginning of the experiment when the ice thicknesses

is similar for all three experiments. As written in the introduction we want to add (not contradict) to the analysis of Steele et al. (1997) by also taking changes in ice export into account, which in our opinion was not done before. If the ice just would get dynamically thicker but the circulation, i.e., ice speed at the ice export gates would stay the same one would observe an increase in ice export. This is not the case. See also the discussion of possible different sea ice flow states in Hibler et al. (2006). The winter reduction in sea ice export as shown here is a positive feedback, which increases the sea ice volume for the weak experiment (in addition to initial dynamic sea ice thickening).

b) The ice deformation analysis in section 4 are not based on monthly statistics. All analysis use the simulated RGPS dataset described in section 4.2, which has an about 3-daily time resolution. We then aggregate all deformations over one month (e.g. in Figs. 4-6) to not show a single day or show a noisy time series (e.g. Fig 8) (could be changed to other time ranges if important but would not change the results).

---

## Referee Comment (RC4) · B. Tremblay (Referee) · 2 May 2016

Title:  Sea Ice Deformation in a Coupled Ocean-Sea Ice Model and in Satellite Remote Sensing Data

Authors: G. Spreen, R. Kwok, D. Menemenlis,  A. Nguyen

In this paper, the authors present: 1- a sensitivity study of the simulated sea ice mass balance on the sea ice strength parameterization and 2- a sensitivity study of the simulated sea ice deformation (divergence, shear, vorticity) on the spatial resolution of the model. The model is the coupled ice-ocean MITgcm with a two-category ice thickness model and a viscous plastic sea-ice rheology. The pressure term in this model is the standard parameterization of Hibler (1979) with a linear dependence on h and exponential dependence on sea ice concentration.

The authors show that a lower ice strength parameter leads to a reduced net annual ice export through Fram Strait and an overall reduced ice production in the simulations after 8 years of integration. They show that the reduced ice export is the dominant mechanism explaining an increase in ice volume in their runs with reduced ice strength. They conclude that the ice mass balance in coupled ice-ocean models is very sensitive to the value used for the ice strength parameter.

In the second part of the paper, they compare their simulated deformation fields (divergence, shear and vorticity) at different spatial resolutions with the Radarsat Geophysical Processor System (RGPS) satellite observations on the basis of their spatial patterns, power law scaling and probability density functions (PDFs). They find that the simulated deformations with the highest spatial resolution (4.5 km) agree best with observations on all metrics tested. However, they show that the model does not capture the enhanced deformations (magnitude and spatial density) in the seasonal ice zone at any spatial resolution and that it has a mean total deformation rate that is about 50% lower than observations. The authors attribute this shortcoming to the ice strength formulation being linearly proportional to the ice thickness. On the other hand, they are able to reproduce the power law scaling of the total deformation rate with the spatial resolution as observed in RGPS observations and the PDFs also agrees with those of RGPS – but are in contradiction with results from Girard et al 2009.

The paper is generally well written – despite some awkward sentence structures and typos (see specific comments below). It presents a long-awaited (re) analysis of the scaling law for sea ice deformations simulated by viscous plastic sea ice models – with results that are contrary to what was published in Girard et al. but that are in accord with several other modeling groups that have done similar analysis. This paper constitutes a welcomed clarification. The results on the effect of the sea ice strength parameterization on the sea ice mass balance are also insightful. Given that the Arctic is transitioning to a seasonal ice cover, and that current rheological

models do not simulate the correct deformation characteristics of the seasonal pack ice (as reported here) is interesting.

The tone of the paper should be less a little less defensive and/or more assertive. The paper presents very interesting results. Those new results need to be prominent. For instance, negative results are presented first followed by positive results. The particular is presented before the general. The results that cannot be compared with observations are presented first followed by the results that can be compared with observations. All of this makes the key findings of the paper more difficult to find and appreciate. More specifically, a key finding of the paper (one that is buried deep in the paper) is that the simulated sea-ice deformation simulated by a viscous-plastic model follows a power law - contrary to what was presented in Girard et al 2009. The results presented in Girard et al. 2009 cannot be reproduced by the authors nor by any other modeling group in the community, yet it has become common (accepted) knowledge that VP rheologies do not follow a power law. This must really be stated early on and clearly. More suggestions regarding this issue are listed below.

We recommend that the paper be accepted for publication after having addressed the comments below carefully.

Amelie Bouchat, PhD candidate
Bruno Tremblay

**Major Points**:

1.    Page 6: general comment: Since the ice export depends on ice thickness in the central Arctic. I would discuss the change in ice thickness in the Arctic with changing P* first. Then I would discuss the change in ice export. I understand that it is a chicken and egg situation, but still ice will thicken in the Arctic irrespective of lower export because of weaker ice. The lower export is a positive feedback of the increase in ice thickness – i.e. the increase in ice thickness does not compensate for the reduction in sea ice velocity. Now we are reading the paper about the export changes without knowing all a-priori knowledge.

2.    A discussion of the ice thickness distribution should be included in the manuscript. The fact that the deformations in the model are generally too low in magnitude and too sparse maybe due to the fact that the ice is too thick. This may also explain why the deformations in the seasonal ice zone are too weak.

3.    We disagree with the interpretation from the authors that the discrepancy between RGPS and the simulated deformation in the seasonal ice zone is

necessarily due to the linear relationship between P and h. A map of the simulated ice thickness for March and September for different ice strength would be useful to better understand this issue.

4. Page 11, line 23: I am not sure we can blame all of this on the linear h dependence of P*. The ellipse results in equally large viscous coefficients (eta and zeta) for the same divergence (in absolute value) and for a given shear. In reality, sea ice would interact little with other ice floes when we have divergent sea ice motion. I would think that in the seasonal ice zone, where there is more space for the pack ice to expand (in regions of coastal polynya, etc), an elliptical yield curve and normal flow rule that gives unrealistically large viscous coefficient in divergence, would lead to reduced deformation as you see here. This is jus another possibility. The point is that I do not think that this can simply be related to the linear dependence of P on h as discussed here.

5. Page 15, line 20: Start your discussion here where you analyze the results for the same geographical region as that of the RGPS. Then you discuss the caveat associated with including points close to coastlines. I.e. you go from General to specific. The way it is presented is a little defensive (i.e. you show the problems first and then show what works well). These are very nice results, one that is in conflict with that of Girard et al. but in accord with results from all other sea ice modeling groups. The authors need to make this point more prominent. I would say this point is one of the highlight of your paper and finally clarifies this situation.

6. In section 4.4, I would discuss the case where you compute the scaling exponent with same domain as RGPS first, since this is what you are interested in to compare with observations. Then when you know you are doing fine, you can go and discuss the fact that this scaling exponent depends on ice concentration and thickness. Also, 3-day means should be used instead of daily means of deformation to have data as similar to RGPS as possible for the comparison.

**Minor Points - A**:

Page 5, line 2: define shear and divergence. They are defined but only much later in section 4.2.

Page 6, line 18: It is not clear what the authors are referring to by "anisotropic behavior of sea ice". The authors are using the standard Hibler rheology which is isotropic. This should be clarified.

Page 6, Line 19: Type-O. "the the"

Page 6, line 19: These are important sentences. They must be expanded. Describe the ice arching. Show example in a figure? "Leads to change in the sea ice circulation". This is vague. What kind of changes? How are they link with ice export? The paper is about P* and ice export. These must be documented.

Page 6, line 20: Again vague statement. What fraction is due to arching, and what fraction is due to changes in the sea-ice circulation. This must be quantified.

Page 6, line 30: Add space before 0.3 P*.

Page 6, line 30: Is it really interesting to quote the total (sum over years) difference in ice export? I would prefer to see the new equilibrium numbers in km/yr.

Page 7, line 2. No it should be discussed first. The fact that the change in export cant totally be discussed at this stage suggest that the order should be changed.

Page 7, line 5: "...sea ice export (E^bar)..."

Page 7, line 15: I am guessing the export must increase since the ice strength is lower and that the ridging more than compensate for this in the first 5 years. You need to discuss the ice export variation in this part of the paper.

Page 7, Line 23: This is counter-intuitive. I would have expected an increase in the ice volume export. Again, two opposing effects are at play: increase ice thickness and reduced ice velocity. A few additional words should be included to clarify this.

Page 8, line 5: Give many examples or kill "e.g."

Page 8, line 9: The best value for P* is traditionally found minimizing the error between the simulated drift and the observed drift using models where the wind forcing is specified as observed. Of course biases in the thickness field will impact the optimal P*. But in principle, a model that assimilates sea ice concentrations, and ice thickness from satellite and forced with reanalysis data could be used to find an optimal value for P*.

Page 8, line 12: give references.

Page 8, line 28: This should read "from the simulated ice motion dataset..."?

Page 9, line 6:"...since November 1996 until 2008..."

Page 10, line 5: Why are they removed? Please clarify.

Page 11, line 29: Define the periods here as well (not just in the Table)

Page 11, line 33: "… on the sea-ice deformation rate"

Page 12, line 13: "…slightly differs from this general behavior…"

Page 12, line 12: This sentence is not English. "… shows a weak minimum in March in contrast with the RGPS data…"

Page 12, line 23: Is the model iterated to convergence? We see much better defined LKFs in a model that was iterated to convergence compared with one that was not, see for instance Lemieux Tremblay (JGR). I am curious if this has an impact on your simulation results.

Page 12, line 24: "… is calculated as:,… where Di are …"

Page 13, line 14: say which summer months.

Page 14, line 2: missing word or one word too many. "…find an in magnitude…"

Page 14, line 18: When we do best linear fit in log-log scale the error for large D will be underestimated. I.e. you best fit will preferentially minimize the error for the small D. Can you comment on the impact of doing this?

Page 14, line 22: typo. Missing dot in -0.54.

Page 14, line 18: You have already said above that there is a constant b value in the winter and a higher b value in the summer. I.e. we cannot just use a constant value. Why test the constant b case if this is so? Eliminate this part? Or say why you still want to look at it.

Page 14, line 20: "…approaches zero linearly…" instead? "…for 100% ice-covered … the deformation rate decreases exponentially". The part of the sentence "but in a more exponential way" is colloquial English.

Page 15, line 8: It is not clear why A=1 would prevent the power law to exist. The exponential dependence of P on A is a continuous function. Why are we loosing it only for A=1?

Page 15, line 12: "geographic location" is not a physical parameter. I think you mean, that the power law exponent depends on the "mean internal ice stress" which is higher when we are in the proximity of continents.

Page 17, paragraph starting at line 24: The authors need to discuss what works first and then discuss what does not work. It is the same content, just the order that needs to be changed.

Page 18, line 5: Again the order should be reversed. The authors need to discuss the results using the same domain as the RGPS and then the one where they include the regions close to the coastlines.

**Minor points - B**

Suggestion: "sea ice deformation" should read "sea-ice deformations" in most places in the text. "Sea ice" takes a hyphen when used as a compound adjective.

-- PAGE 1 --

Line 8-9 : Replace "All three model simulations can reproduce the large-scale ice deformation patterns but ..." with: "All three model simulations can reproduce the large-scale ice deformation patterns, but small scale sea-ice deformations and linear kinematic features are not adequately reproduced." Then go with "The overall sea ice..." followed by "A decrease in ...".

Line 10: Replace "The overall sea ice deformation" with "The mean sea-ice total deformation rate"

Line 16-17: "Either way, this study..." Delete sentence.

-- PAGE 2 --
Line 4-5: Suggestion: Change "or if new sea ice rheologies like the one..." for "or if new sea-ice rheologies (Girard et al. 2011, Sulsky et al. 2007, etc.) have to be used."

Line 6: "(2) brine rejection into the ocean, (3)..." Add "(2) brine rejection in the ocean due to freezing in open water areas, (3)..."

Line 13: "were" should be "are"

Line 13-15: Suggestion: Change to "The model sensitivity to the model ice strength parameterization is assessed by comparing the model solutions with different ice strength parameters to the RGPS satellite observations spatially and temporally. These comparisons also allow us to study the model uncertainties regarding the sea-ice deformation representation in the current formulation of VP models."

Line 18: "into a mean and fluctuating field" change to "into mean and fluctuating fields"

Line 19: "to evaluate models with first order..." change to "to evaluate models on the basis of their first order mean velocity field and it can be correctly predicted even by simple sea ice models..."

Line 20: "Second order sea ice deformation fields..." change to "The second order sea-ice velocity field, represented by the sea ice deformation fields (strain rates), has to be used for comparison to take into account the high frequency fluctuations of the sea-ice velocity field and to assess the quality of the sea-ice rheology formulation."

Line 24: "For RGPS deformation rates" should be "For RGPS total deformation rates"

Line 25: "a scale dependence" should be "a spatial scale dependence"

Line 34: Replace "for example they show" with "showing"

Line 35: Replace "Some improvement in modeling sea ice deformation" with "Improvements in the modeled sea-ice deformation"

-- PAGE 3 --
Line 4-6: "A recent example...." Delete sentence.

Line 11: Replace "We reconstruct the observed sea ice deformation..." with "Using the VP model, we construct simulated deformation fields on the same spatial and temporal scales as in the RGPS observations."

Line 12: Replace "In addition we also compare..." with "We then compare the power law scaling properties of the modeled and observed deformation rates (section 4.4) and we perform a sensitivity study of the deformation fields properties to the model ice strength parameter (section ??)"

Line 13-14: Delete "sea" and "and thereby ice deformation"

Line 16: Delete "as a consequence also" and replace "can effect the Atlantic Ocean circulation" with "can also affect the modeled Atlantic Ocean circulation"

Line 16-18: "Ultimately, we would like..." Reformulate. Maybe write: "Ultimately, we would like to highlight why the sea-ice strength representation and the sea-ice rheology should receive more attention in models."

-- PAGE 4 --
Line 15: "fit to available satellite and in-situ data..." Data of what? Ice velocity? Ice thickness? Please specify.

Line 22: "As a consequence these higher-resolution simulations exhibit somewhat larger model drifts relative to observations than the 18-km simulation." Does that mean that therefore you would need to increase P* with increasing resolution to slow down the pack? Please state so if it is the case.

Line 27: Replace "thus the local ice thickness distribution" with "thus modifies the ice thickness distribution" and change "Furthermore, changes in the model ice strength alter the sea-ice drift speed..."

Line 28: Replace "changes in sea ice deformation therefore..." with "these changes can alter the equilibrium sea ice volume in the Arctic."

Line 29: Replace "a set of sensitivity experiments" with "a set of experiments" and replace "changes in sea ice deformation to motivate the importance of sea ice deformation" with "changes in ice strength parameter to highlight the importance of using accurate rheological models and sea-ice deformation fields"

Line 31: Replace "start" with "are done"

Line 32: Replace "The sea ice deformation rate" with "The total sea-ice deformation rate"

-- PAGE 5 --
Line 1-3: Rewrite as: ", where nabla_dot is the divergence rate and tau_dot is the shear rate, is used as a measure for the overall sea-ice deformation occurring at a certain point in space (e.g. Stern and Lindsay 2009). The magnitude of both the divergence and shear rates are to some extent controlled by the strength of the sea ice. In our model configuration, we use the typical ice pressure formulation P (or strength) of Hibler 1979:"

Line 13: Maybe it would be worth noting that the differences in the values of P* that are used in different models come in part because of the need to calibrate the parameters of one's model depending on the forcing used (ocean + atm.) and drag formulations. There is also the need to recalibrate this P* parameter depending on the spatial resolution used in the model.

Line 13: What is the time step used for simulations?

Line 18: Add "For any given month, the monthly deformation rate D_bar increases..."

Line 20: Replace "deformation rates" with "simulations"

Line 22: Replace "of these sea ice deformation" with "of changes in the deformation rates and ice velocity on..."

Line 25: Delete "will" and "for a discussion of geophysical sea ice volume change over time, see Nguyen et al. (2011)."

Line 28: Replace "starts immediately to" with "rapidly" and delete sentence "A similar sensitivity...". Instead, add "Hence, after 8 years of integration, the sea ice volume has increased by 7%..." and continue with sentence from line 30-31.

Line 29: Maybe add a sentence here to clearly state that you do have thicker ice in agreement with Steele et al, but what controls the ice volume change in your simulations are the changes in ice export and ice production and melt.

Line 33: Replace "quickly diverges from the baseline. The divergence gets..." with "diverges from the baseline at a much faster rate than for the solution with 0.7P*_0. The rate of increase of the ice volume gets smaller after 1999, but the volume keeps increasing until 2005."

-- PAGE 6 --
Line 1: Why does the volume start decreasing after 2005 in both runs? And there seems to be much more variability in the case P*=0.3P*0. than with P* = 0.7P*0. Can you comment?

Line 4-5: Put this sentence in previous section, and maybe add something like "both these mechanisms are explored in the following sections".

Line 5: Delete "also" and add it on line 6 between "experiments" and "diverges"

Line 8: Add "Even more pronounced is the change" Delete "however".

Line 11: Rewrite: "...(blue shaded area), and during winter, E_bar is lower than..."

Line 12: Delete "however" and "large" and replace "overall" with "the net annual"

Line 13: Add "nearly balance in the course of one year and this results in a net annual decrease in... "

Line 13: Can the very enhanced seasonal cycle of run with P*=0.30P*0 explain the high variability seen in Fig. 1a of sea ice export compared to run with P* = 0.7P*0?

(See comment for p.6 line 1 above.) If it is the case, then I would suggest moving this section before section 3.1 for clarity.

Line 15 : "Intuitively one might expect an increase of ice export for weaker ice since the ice speed increases." Add "Intuitively one might expect an increase of ice export for weaker ice even during winter since the ice speed increases."

Line 15-16: Change "The ice area export (not shown), however, is smaller for both "weak" experiments during the complete year." for "However, during both summer and winter, the ice area export (not shown) is smaller for both "weak" experiments."

Line 17: "The increase in ice thickness..." This isn't shown in the paper. It would benefit the reader to see maps of mean thickness for your runs and could help you explain better the differences in ice volume, export and even later for your deformation fields.

Line 18-20: I am confused here. You are using an isotropic VP model, yet you are talking about the anisotropic behavior of P. It is also not very clear why the export is less during the winter when the ice strength is weaker. Please expand this paragraph with further explanations.

-- PAGE 7 --
Line 11: Please specify in text what a positive/negative delta_B means. Does a positive delta_B means that there is more ice production and negative delta_B means that there is more ice melting?

Line 25-26: Delete "and also small compared to the volume differences caused by the reduced sea ice export (Figure 3b)." In the run with P*=0.3P*0, it is approximately a third of the changes in the ice volume. It is not small.

Line 27-28: "The results suggest that..." Maybe state that up front in section 3.1 when talking about the sea ice volume changes and say that you explain this in the next sections. Or again, move this section before section 3.1

Line 29: Replace "deformation" with "strength"

-- Page 8 --
Line 28: Why not use the "Lagrangian ice deformation" product directly? Or even the Eulerian ice deformation product?

-- Page 9 --
Line 19: Why using triangles and not a square grid? If I am not mistaken, RGPS uses a square grid to calculate these integrals. Also, the error associated with the estimates of deformation are greater when using triangles than with squares. See Thorndike, Kinematics of Sea Ice, Chapter 7 in The Geophysics of Sea Ice, NATO ASI Series, vol 146, 1986. In particular: section 5.4.5 - Errors in Estimating the Large Scale Deformation.

Equations (3) : Do you compute these integrals assuming u/v vary linearly between each corner? Please specify.
* * *
Line 4: In what sense do you associate a total deformation of 1 day^-1 to a deformation of 100%? What ratio are you taking to find a percentage?

Line 17-18: Put this sentence before the last one? It is really referring to the fact that you are putting everything on the same grid, not that some runs are under-sampled or oversampled.

Last paragraph: Maybe differences in ice thickness could explain this? If the ice is too thick in the model, it will be stronger and you will have less deformations. It would be nice to see the thickness fields.

Line 30: Replace "...and model shear is worst." with "...and model shear is the worst."

Line 31: Replace "...and model is best." with "...and model is the best."
* * *
Line 3: Delete sentence "The picture changes when..."

Line 5: Delete ": divergence, shear and vorticity."

Line 9: "...its deformation distribution is most consistent with RGPS observations." On what basis? PDFs? Spatial Patterns?

Line 16-17: Delete sentence: "The representation of large-scale sea ice deformation..."

Line 18: What is the black contour? How do you define seasonal ice? Please mention in your text.

Line 22-23: "The model sea ice strength P, as defined in Equation 2, depends linearly on ice thickness h. Clearly the linear relationship between P and h is not suitable to realistically model sea ice deformation." As mentioned earlier, the problem here could be instead that the model has too thick ice in the seasonal ice zone....

Line 24: "Models with more ice thickness classes often use a $P \sim h^{(3/2)}$ formulation (Rothrock, 1975; Lipscomb et al., 2007)" Doesn't this mean that you make ice more stiff? This will not fix the problem that you do not have enough deformations in the seasonal ice zone... it will in fact make it deform even less.

What I see is that the problem here is that your seasonal ice (supposed to be thinner) may be too and not deforming enough... Can you show a map of sea ice thickness? Increasing the dependence of P on h will not help this problem, since stronger ice deforms less and leads overall to an ice pack that is thinner (see Steele et al. 1997 for example).

Line 31: "for visual clarity the period means... " Not clear... Does this apply to figure 8a only? If so, then maybe write something like :
"Figure 8 shows (a) the period-averaged sea-ice deformation rate D_dot, and (b) the monthly-mean seasonal cycle of D_dot (both computed with all 20 RGPS periods available)."

-- PAGE 12 --
Line 1: Are these numbers the total mean? Please specify.

Line 5: Again, I would check the differences in the thickness field to see if it can explain the differences between your runs. Also, the fact that your model seems 50% too low in deformation could again be linked to the fact that the ice in your model is generally too thick, too strong...

Line 11: March instead of May?

Line 12: Replace "and shows a small but, compared to RGPS data, not very pronounced minimum during March." with "and shows a small but not very pronounced March minimum compared to RGPS data."

Line 13: Delete sentence "That is, the 4.5km solution..."

Line 17: Delete sentence "Again the 4.5km solution..."

-- PAGE 13 --

The discussion on Q could maybe be combined with section 4.3.1?

Line 12: Can you give more details about the implications of having an enhanced seasonal cycle of Q in the model?

Line 27: Here do you compute the deformation rates D_dot from the triangulation of the RGPS positions? Or do you use the Eulerian grid of the model? Please clarify.

-- PAGE 14 --

Line 3: Replace "find an in magnitude about 50% lower scaling exponent (i.e. b ∼ −0.12 during winter) for the deformation rate." with "find the magnitude of the scaling exponent to be about 50% lower (ie, b approx -0.12 during winter) for the deformation rate."

Line 8: "...the mean sea ice deformation rate" Monthly means?

Line 10-12: As you can see here with your mean deformation rates, you have much higher values than in figure 8 because you are considering regions of very high strain rates (probably near the coast and in the region of the transpolar drift)... If you are to compare those number with RGPS, you have to bring everything on the same domain covered by RGPS only.

Line 13-14: "Some years, e.g., 1997–1999, have clearly reduced summer deformation rates in comparison to, e.g., the beginning of the 1990s or 2007 and 2008." This is not very clear to see on the figure... Maybe plot winter average and summer average on Fig 10 (a) and (b) instead of monthly means?

Line 14-15: Delete sentence "The deformation rate during 2008..."

Line 19: "daily mean", Maybe use a 3-day period to be as close as possible to RGPS?

Line 20: "the power law scaling exponent b is estimated to be −0.54." Maybe you should show the graph with all the daily mean deformation rates as a function of L and plot the regression line you find. It would make it more clear as to where that number comes from.

Line 20-21: "Figure 10b shows the deformation rate time series for the three model solutions normalized to a length scale of L = 10 km, using the estimated scaling exponent b = −0.54" How do you do this normalization to a different length scale?

Line 23-24: "If looked in detail, however, there remain some quite large differences." This is really not clear on figure. Maybe, as suggested earlier, if you present season means in the graph it would be more clear and we could see better the differences.

-- PAGE 15 --
Line 2-3: "The scaling exponent b gets more negative for weaker sea ice and approaches zero for very strong sea ice, i.e., thick ice and 100% ice concentration" Maybe you need to explain clearly what is the relation between b and Fig.10 b and c. It is the spacing between the curves, ie the larger the space, the larger the slope?

Line 6: Replace "even at 100% ice-cover a cell should show power-law scaling behavior." with "a cell should show power law scaling behavior even with a 100% concentration."

Line 7-8: Why is that? So then, can we really expect to find a power-law scaling in winter, when concentration is almost 1 everywhere?

Line 9: Replace "free ice drift" with "free-drift ice"

Line 15: Replace "the b values of" with "the values of b of" and replace "b values between" with "the values of b between"

Line 17-18: Why not start the section with this? And then say that the value of b is dependent on the ice concentration and thickness, so that if you consider different regions in the Arctic you end up with different b's. And then present your results when considering the whole Arctic domain.

Line 30-31: "model output was bin-averaged to the same spatial scale, L = 12.5 km," What does that mean that the data is bin-averaged? Please explain method.

-- PAGE 16 --
Line 5: "A linear regression was applied to the PDFs in log-log space 5 for the deformation rate range 0.03–0.8 day−1, shown as dashed lines in Figure 11." Not very visible on the graph. Could be removed or offset.

Line 25: Girard et al. 2009
* * *
Line 5: Replace "(ice growth equals ice export)" with "(ie, when ice growth equals ice export)"

Line 10: Ocean sensitivity was never really mentioned in the paper... Delete this sentence?

Line 11: Replace "more deformation" with "more deformations"

Line 11: "the ocean mixed layer depth increases during winter time." This was not shown.

Line 14: Add "Deformations in Arctic ocean and sea ice simulations..."

Line 20-21: "The largest difference occurs for the magnitude of divergence, which is 67% to 79% too low (Table 4)." I do not recall seeing this clearly stated in the discussion. Please add.

Line 26-27: "This suggests a shortcoming of the ice rheology, for example, the linear dependence between ice strength and ice thickness." Not necessarily... Again, you have to check the ice thickness first. It could be due to the fact that your seasonal ice is too thick.

---

## Author Comment (AC2) · 6 Jun 2016

Answer to RC4 (Bruno Tremblay and Amelie Bouchat) for the manuscript "Sea Ice Deformation in a Coupled Ocean-Sea Ice Model and in Satellite Remote Sensing Data" by G. Spreen, R. Kwok, D. Menemenlis, and A.T. Nguyen

Dear Bruno, dear Amelie,

Thank you very much for your detailed and very helpful review of our manuscript. Find below your comments in blue and our answers to them in black. We will follow them closely and think that we can address almost all of them in a revised version.

In this paper, the authors present: 1- a sensitivity study of the simulated sea ice mass balance on the sea ice strength parameterization and 2- a sensitivity study of the simulated sea ice deformation (divergence, shear, vorticity) on the spatial resolution of the model. The model is the coupled ice-ocean MITgcm with a two-category ice thickness model and a viscous plastic sea-ice rheology. The pressure term in this model is the standard parameterization of Hibler (1979) with a linear dependence on h and exponential dependence on sea ice concentration.

The authors show that a lower ice strength parameter leads to a reduced net annual ice export through Fram Strait and an overall reduced ice production in the simulations after 8 years of integration. They show that the reduced ice export is the dominant mechanism explaining an increase in ice volume in their runs with reduced ice strength. They conclude that the ice mass balance in coupled ice-ocean models is very sensitive to the value used for the ice strength parameter.

In the second part of the paper, they compare their simulated deformation fields (divergence, shear and vorticity) at different spatial resolutions with the Radarsat Geophysical Processor System (RGPS) satellite observations on the basis of their spatial patterns, power law scaling and probability density functions (PDFs). They find that the simulated deformations with the highest spatial resolution (4.5 km) agree best with observations on all metrics tested. However, they show that the model does not capture the enhanced deformations (magnitude and spatial density) in the seasonal ice zone at any spatial resolution and that it has a mean total deformation rate that is about 50% lower than observations. The authors attribute this shortcoming to the ice strength formulation being linearly proportional to the ice thickness. On the other hand, they are able to reproduce the power law scaling of the total deformation rate with the spatial resolution as observed in RGPS observations and the PDFs also agrees with those of RGPS – but are in contradiction with results from Girard et al 2009.

The paper is generally well written – despite some awkward sentence structures and typos (see specific comments below). It presents a long-awaited (re) analysis of the scaling law for sea ice deformations simulated by viscous plastic sea ice models – with results that are contrary to what was published in Girard et al. but that are in accord with several other modeling groups that have done similar analysis. This paper constitutes a welcomed clarification. The results on the effect of the sea ice strength parameterization on the sea ice mass balance are also insightful. Given that the Arctic is transitioning to a seasonal ice cover, and that current rheological

models do not simulate the correct deformation characteristics of the seasonal pack ice (as reported here) is interesting.

The tone of the paper should be less a little less defensive and/or more assertive. The paper presents very interesting results. Those new results need to be prominent. For instance, negative results are presented first followed by positive results. The particular is presented before the general. The results that cannot be compared with observations are presented first followed by the results that can be compared with observations. All of this makes the key findings of the paper more difficult to find and appreciate. More specifically, a key finding of the paper (one that is buried deep in the paper) is that the simulated sea-ice deformation simulated by a viscous-plastic model follows a power law - contrary to what was presented in Girard et al 2009. The results presented in Girard et al. 2009 cannot be reproduced by the authors nor by any other modeling group in the community, yet it has become common (accepted) knowledge that VP rheologies do not follow a power law. This must really be stated early on and clearly. More suggestions regarding this issue are listed below.

Thank you for this comment. We will restructure the paper following these lines. We will change the order of sections 3 and 4 and start with the model to data comparison first, as both reviews agree that this is the most important part of the paper. Also the order of four sub-sections will be changed to allow a better flow of the results. The power law result will already be mentioned in the abstract.

We recommend that the paper be accepted for publication after having addressed the comments below carefully.

Amelie Bouchat, PhD candidate
Bruno Tremblay

**Major Points**:

1. Page 6: general comment: Since the ice export depends on ice thickness in the central Arctic. I would discuss the change in ice thickness in the Arctic with changing P* first. Then I would discuss the change in ice export. I understand that it is a chicken and egg situation, but still ice will thicken in the Arctic irrespective of lower export because of weaker ice. The lower export is a positive feedback of the increase in ice thickness – i.e. the increase in ice thickness does not compensate for the reduction in sea ice velocity. Now we are reading the paper about the export changes without knowing all a-priori knowledge.

We will change the order of sections 3.2 export and 3.3 ice production/melting and now first discuss ice thickness changes.

2. A discussion of the ice thickness distribution should be included in the manuscript. The fact that the deformations in the model are generally too low in magnitude and too sparse maybe due to the fact that the ice is too

thick. This may also explain why the deformations in the seasonal ice zone
are too weak.

Figure 1 shows the spatial distribution of ice thickness for the 9 km model solution. A sub-figure showing the ice thickness distribution for the three model solutions will be added. For a detailed comparison of the model ice thickness to measurements see Nguyen et al. (2011), who use the same 9 km model solution as presented here.

3. We disagree with the interpretation from the authors that the discrepancy between RGPS and the simulated deformation in the seasonal ice zone is necessarily due to the linear relationship between P and h. A map of the simulated ice thickness for March and September for different ice strength would be useful to better understand this issue.

We reformulate this statement to become more a hypothesis. We did some test with changed relationships, which support the hypothesis but this would be a different study and don't think it is necessary to discuss this in detail here.

4. Page 11, line 23: I am not sure we can blame all of this on the linear h dependence of P*. The ellipse results in equally large viscous coefficients (eta and zeta) for the same divergence (in absolute value) and for a given shear. In reality, sea ice would interact little with other ice floes when we have divergent sea ice motion. I would think that in the seasonal ice zone, where there is more space for the pack ice to expand (in regions of coastal polynya, etc), an elliptical yield curve and normal flow rule that gives unrealistically large viscous coefficient in divergence, would lead to reduced deformation as you see here. This is jus another possibility. The point is that I do not think that this can simply be related to the linear dependence of P on h as discussed here.

We will change these sentences to:
"This discrepancy between seasonal and perennial ice hints to a shortcoming of the sea ice rheology used in the simulations. To first order the main difference between seasonal and perennial sea ice is the ice thickness. The model sea ice strength P , as defined in Equation 2, depends linearly on ice thickness h. This is the typical P formulation for a VP or EVP sea ice rheology with two ice classes and might not be the best representation of the P to h relationship. Models with more ice thickness classes often use a $P \propto h^{3/2}$ formulation (Rothrock, 1975; Lipscomb et al., 2007), which can be considered more realistic. There are, however, also other differences between the seasonal and perenial ice zone than the ice thickness. The proximity to open water, for example, will allow more cases of ice divergence at the ice margins than in the ice pack, which might be less well represented by the VP rheology."

5. Page 15, line 20: Start your discussion here where you analyze the results for the same geographical region as that of the RGPS. Then you discuss the caveat associated with including points close to coastlines. I.e. you go from

General to specific. The way it is presented is a little defensive (i.e. you show the problems first and then show what works well). These are very nice results, one that is in conflict with that of Girard et al. but in accord with results from all other sea ice modeling groups. The authors need to make this point more prominent. I would say this point is one of the highlight of your paper and finally clarifies this situation.

We changed the order of Sections 4.4 and 4.5 and first show that the PDFs of the model solutions in general follow a power law comparable to the RGPS observations. This is then followed by the section where we use a power law to make the deformation rates of the model solutions with different grid spacing comparable. Parts of these sections were reformulated to make the findings more prominent.

6.    In section 4.4, I would discuss the case where you compute the scaling exponent with same domain as RGPS first, since this is what you are interested in to compare with observations. Then when you know you are doing fine, you can go and discuss the fact that this scaling exponent depends on ice concentration and thickness. Also, 3-day means should be used instead of daily means of deformation to have data as similar to RGPS as possible for the comparison.

In section 4.4 we use the power law dependence of deformation rates to make deformation rates of model solutions with different grid spacing comparable. This approach cannot directly be compared to the RGPS data as also other factors than the model grid spacing will influence the deformation rate between the three model solutions.
We therefore changed the order of sections 4.4. and 4.5 (see last comment) and now start with the power law behavior of the probability density functions, which can be directly compared with the RGPS data.

**Minor Points - A**:

Page 5, line 2: define shear and divergence. They are defined but only much later in section 4.2.

Added reference to strain rate definitions

Page 6, line 18: It is not clear what the authors are referring to by "anisotropic behavior of sea ice". The authors are using the standard Hibler rheology which is isotropic. This should be clarified.

Yes, there is no sub-grid scale anisotropy and "anisotropic" probably was not the best word here. We were referring to the irregular distribution of ice stress causing e.g. LKFs and ice arches. Changed in manuscript.

Page 6, Line 19: Type-O. "the the"

corrected

Page 6, line 19: These are important sentences. They must be expanded. Describe the ice arching. Show example in a figure? "Leads to change in the sea ice circulation". This is vague. What kind of changes? How are they link with ice export? The paper is about P* and ice export. These must be documented.

We did not explore changes in ice circulation within the Arctic Basin in detail. This was already done by Steele et al. (1997) who find an acceleration of the Beaufort Gyre and a stronger piling up of ice at the coast of North America for reduced P*. We agree that this manuscript should focus on ice export. We will expand the discussion on this. However, the main part of the paper is about the model to RGPS data comparison. For a more detailed discussion about changes in the force balance, Arctic Basin thickness, and circulation we can only refer to Steele et al. (1997). They, however, do not consider changes in ice export, which we show to be one of the main contributors to the observed changes within the Arctic Basin (which we will not explore in detail here).

Page 6, line 20: Again vague statement. What fraction is due to arching, and what fraction is due to changes in the sea-ice circulation. This must be quantified.

See last answer.

Page 6, line 30: Add space before 0.3 P*.

corrected

Page 6, line 30: Is it really interesting to quote the total (sum over years) difference in ice export? I would prefer to see the new equilibrium numbers in km/yr.

We assume by this you mean the mean difference in export from e.g. 2000 onwards. We will add these numbers.

Page 7, line 2. No it should be discussed first. The fact that the change in export cant totally be discussed at this stage suggest that the order should be changed.

Order of sections 3.2 and 3.3 will be changed

Page 7, line 5: "...sea ice export (E^bar)..."

added

Page 7, line 15: I am guessing the export must increase since the ice strength is lower and that the ridging more than compensate for this in the first 5 years. You need to discuss the ice export variation in this part of the paper.

The ice export is not changing significantly between the three experiments during the first two years. Note that the export E is removed for the calculation of the ice production B. We will add the sentence: "This causes the ice production B to increase compared to the

baseline. B is corrected for the influence of ice export E, which, however, does not change much from the baseline integration during the first two years (not shown).

Page 7, Line 23: This is counter-intuitive. I would have expected an increase in the ice volume export. Again, two opposing effects are at play: increase ice thickness and reduced ice velocity. A few additional words should be included to clarify this.

Yes, the reduction in ice export is not directly intuitive. Therefore we describe the different effects leading to it in a separate section 3.2 and only reference to it here.

Page 8, line 5: Give many examples or kill "e.g."

Removed "e.g."

Page 8, line 9: The best value for P* is traditionally found minimizing the error between the simulated drift and the observed drift using models where the wind forcing is specified as observed. Of course biases in the thickness field will impact the optimal P*. But in principle, a model that assimilates sea ice concentrations, and ice thickness from satellite and forced with reanalysis data could be used to find an optimal value for P*.

We used a method similar what you describe to find the optimal P* value for our baseline integration. Will add that information to the text.

Page 8, line 12: give references.

added

Page 8, line 28: This should read "from the simulated ice motion dataset…"?

No, we are still talking about the observed RGPS SAR ice motion here.
Clarified in text.

Page 9, line 6:"…since November 1996 until 2008…"

done

Page 10, line 5: Why are they removed? Please clarify.

Deformation rates higher than 1 are considered outliers. Clarified in text.

Page 11, line 29: Define the periods here as well (not just in the Table)

Maybe we misunderstand what you mean. Table 3 lists 20 periods. To include them all in the text would be hard to read.

Page 11, line 33: "… on the sea-ice deformation rate"

done

Page 12, line 13: "…slightly differs from this general behavior…"

done

Page 12, line 12: This sentence is not English. "… shows a weak minimum in March in contrast with the RGPS data…"

done

Page 12, line 23: Is the model iterated to convergence? We see much better defined LKFs in a model that was iterated to convergence compared with one that was not, see for instance Lemieux Tremblay (JGR). I am curious if this has an impact on your simulation results.

We did not perform explicit tests regarding the convergence of the model. We, however, discussed the Lemieux Tremblay (JGR) paper and concluded that our iterations should be sufficient.

Page 12, line 24: "… is calculated as:,… where Di are …"

done

Page 13, line 14: say which summer months.

Information added.

Page 14, line 2: missing word or one word too many. "…find an in magnitude…"

corrected

Page 14, line 18: When we do best linear fit in log-log scale the error for large D will be underestimated. I.e. you best fit will preferentially minimize the error for the small D. Can you comment on the impact of doing this?

We are not completely sure we understand the question. For large D the number of observations gets very small and therefore the scatter large. We therefore stop our fit at 0.8. We do not aim to minimize the error for larger D
If you are talking about doing the fit in log space and therefore having a non-linear distribution of D than you are right, the fit will preferentially minimize the error for small D. We cannot comment on the impact on that because that would depend on the question. One has to keep in mind that the probability is also scaled logarithmic and therefore there are many more observations with small deformation rates, which one could argue therefore should have a higher impact on the fit.

Page 14, line 22: typo. Missing dot in -0.54.

corrected

Page 14, line 18: You have already said above that there is a constant b value in the winter and a higher b value in the summer. I.e. we cannot just use a constant value.

Why test the constant b case if this is so? Eliminate this part? Or say why you still want to look at it.

Yes, we agree one cannot use a single scaling exponent b to make detailed comparisons between strain rates from models with different grid spacing. As Fig 10 a and b demonstrates using a power law with constant b is still useful to compare mean (complete domain, yearly) strain rates of models with different resolution. While the reproduced details in sea ice deformation are very different between the three solutions, Fig 10b demonstrates that the mean deformation rate of all solutions is quite comparable if one takes the different grid scales into account. Will add a sentence about that to the manuscript.

Page 14, line 20: "...approaches zero linearly..." instead? "...for 100% ice-covered ... the deformation rate decreases exponentially". The part of the sentence "but in a more exponential way" is colloquial English.

Reformulated

Page 15, line 8: It is not clear why A=1 would prevent the power law to exist. The exponential dependence of P on A is a continuous function. Why are we loosing it only for A=1?

As said in the manuscript we do not have a clear answer to that. From theory one should expect the power law scaling to also exist for 100% ice cover. We are not saying that we are loosing the power law scaling just for 100% ice the bower law scaling exponent is converging to zero for high ice concentrations. We can only speculate that this has to do with the exponential dependence on the ice concentration in the model implementation. We remove this discussion from the manuscript as it is not conclusive at the moment.

Page 15, line 12: "geographic location" is not a physical parameter. I think you mean, that the power law exponent depends on the "mean internal ice stress" which is higher when we are in the proximity of continents.

Yes, will be reformulated

Page 17, paragraph starting at line 24: The authors need to discuss what works first and then discuss what does not work. It is the same content, just the order that needs to be changed.

Yes, agreed. The order will be changed.

Page 18, line 5: Again the order should be reversed. The authors need to discuss the results using the same domain as the RGPS and then the one where they include the regions close to the coastlines.

Yes, agreed. The order will be changed.

**Minor points - B**

Suggestion: "sea ice deformation" should read "sea-ice deformations" in most places in the text. "Sea ice" takes a hyphen when used as a compound adjective.

done

-- PAGE 1 --

Line 8-9 : Replace "All three model simulations can reproduce the large-scale ice deformation patterns but ..." with: "All three model simulations can reproduce the large-scale ice deformation patterns, but small scale sea-ice deformations and linear kinematic features are not adequately reproduced." Then go with "The overall sea ice..." followed by "A decrease in ...".

done

Line 10: Replace "The overall sea ice deformation" with "The mean sea-ice total deformation rate"

done

Line 16-17: "Either way, this study..." Delete sentence.

We prefer to keep this sentence.

-- PAGE 2 --
Line 4-5: Suggestion: Change "or if new sea ice rheologies like the one..." for "or if new sea-ice rheologies (Girard et al. 2011, Sulsky et al. 2007, etc.) have to be used."

Followed your suggestion

Line 6: "(2) brine rejection into the ocean, (3)..." Add "(2) brine rejection in the ocean due to freezing in open water areas, (3)..."

done

Line 13: "were" should be "are"

done

Line 13-15: Suggestion: Change to "The model sensitivity to the model ice strength parameterization is assessed by comparing the model solutions with different ice strength parameters to the RGPS satellite observations spatially and temporally. These comparisons also allow us to study the model uncertainties regarding the sea-ice deformation representation in the current formulation of VP models."

Will follow your suggestion

Line 18: "into a mean and fluctuating field" change to "into mean and fluctuating fields"

done

Line 19: "to evaluate models with first order..." change to "to evaluate models on the basis of their first order mean velocity field and it can be correctly predicted even by simple sea ice models..."

Reformulated along the lines of your suggestion.

Line 20: "Second order sea ice deformation fields..." change to "The second order sea-ice velocity field, represented by the sea ice deformation fields (strain rates), has to be used for comparison to take into account the high frequency fluctuations of the sea-ice velocity field and to assess the quality of the sea-ice rheology formulation."

done

Line 24: "For RGPS deformation rates" should be "For RGPS total deformation rates"

done

Line 25: "a scale dependence" should be "a spatial scale dependence"

done

Line 34: Replace "for example they show" with "showing"

done

Line 35: Replace "Some improvement in modeling sea ice deformation" with "Improvements in the modeled sea-ice deformation"

done

-- PAGE 3 --
Line 4-6: "A recent example...." Delete sentence.

Why? The Tsamados et al. (2013) study should be mentioned. We kept the sentence.

Line 11: Replace "We reconstruct the observed sea ice deformation..." with "Using the VP model, we construct simulated deformation fields on the same spatial and temporal scales as in the RGPS observations."

done

Line 12: Replace "In addition we also compare..." with "We then compare the power law scaling properties of the modeled and observed deformation rates (section 4.4)

and we perform a sensitivity study of the deformation fields properties to the model ice strength parameter (section ??)"

Reformulated sentence

Line 13-14: Delete "sea" and "and thereby ice deformation"

done

Line 16: Delete "as a consequence also" and replace "can effect the Atlantic Ocean circulation" with "can also affect the modeled Atlantic Ocean circulation"

done

Line 16-18: "Ultimately, we would like..." Reformulate. Maybe write: "Ultimately, we would like to highlight why the sea-ice strength representation and the sea-ice rheology should receive more attention in models."

done

We looked through the comments in the following and think that we can address almost all of them in a revised version.

-- PAGE 4 --
Line 15: "fit to available satellite and in-situ data..." Data of what? Ice velocity? Ice thickness? Please specify.

Line 22: "As a consequence these higher-resolution simulations exhibit somewhat larger model drifts relative to observations than the 18-km simulation." Does that mean that therefore you would need to increase P* with increasing resolution to slow down the pack? Please state so if it is the case.

Line 27: Replace "thus the local ice thickness distribution" with "thus modifies the ice thickness distribution" and change "Furthermore, changes in the model ice strength alter the sea-ice drift speed..."

Line 28: Replace "changes in sea ice deformation therefore..." with "these changes can alter the equilibrium sea ice volume in the Arctic."

Line 29: Replace "a set of sensitivity experiments" with "a set of experiments" and replace "changes in sea ice deformation to motivate the importance of sea ice deformation" with "changes in ice strength parameter to highlight the importance of using accurate rheological models and sea-ice deformation fields"

Line 31: Replace "start" with "are done"

Line 32: Replace "The sea ice deformation rate" with "The total sea-ice deformation rate"

-- PAGE 5 --

Line 1-3: Rewrite as: ", where nabla_dot is the divergence rate and tau_dot is the shear rate, is used as a measure for the overall sea-ice deformation occurring at a certain point in space (e.g. Stern and Lindsay 2009). The magnitude of both the divergence and shear rates are to some extent controlled by the strength of the sea ice. In our model configuration, we use the typical ice pressure formulation P (or strength) of Hibler 1979:"

Line 13: Maybe it would be worth noting that the differences in the values of P* that are used in different models come in part because of the need to calibrate the parameters of one's model depending on the forcing used (ocean + atm.) and drag formulations. There is also the need to recalibrate this P* parameter depending on the spatial resolution used in the model.

Line 13: What is the time step used for simulations?

Line 18: Add "For any given month, the monthly deformation rate D_bar increases..."

Line 20: Replace "deformation rates" with "simulations"

Line 22: Replace "of these sea ice deformation" with "of changes in the deformation rates and ice velocity on..."

Line 25: Delete "will" and "for a discussion of geophysical sea ice volume change over time, see Nguyen et al. (2011)."

Line 28: Replace "starts immediately to" with "rapidly" and delete sentence "A similar sensitivity...". Instead, add "Hence, after 8 years of integration, the sea ice volume has increased by 7%..." and continue with sentence from line 30-31.

Line 29: Maybe add a sentence here to clearly state that you do have thicker ice in agreement with Steele et al, but what controls the ice volume change in your simulations are the changes in ice export and ice production and melt.

Line 33: Replace "quickly diverges from the baseline. The divergence gets..." with "diverges from the baseline at a much faster rate than for the solution with $0.7P^*\_0$. The rate of increase of the ice volume gets smaller after 1999, but the volume keeps increasing until 2005."

-- PAGE 6 --
Line 1: Why does the volume start decreasing after 2005 in both runs? And there seems to be much more variability in the case $P^*=0.3P^*0$. than with $P^* = 0.7P^*0$. Can you comment?

Line 4-5: Put this sentence in previous section, and maybe add something like "both these mechanisms are explored in the following sections".

Line 5: Delete "also" and add it on line 6 between "experiments" and "diverges"

Line 8: Add "Even more pronounced is the change" Delete "however".

Line 11: Rewrite: "...(blue shaded area), and during winter, $E\_bar$ is lower than..."

Line 12: Delete "however" and "large" and replace "overall" with "the net annual"

Line 13: Add "nearly balance in the course of one year and this results in a net annual decrease in... "

Line 13: Can the very enhanced seasonal cycle of run with $P^*=0.30P^*0$ explain the high variability seen in Fig. 1a of sea ice export compared to run with $P^* = 0.7P^*0$?

(See comment for p.6 line 1 above.) If it is the case, then I would suggest moving this section before section 3.1 for clarity.

Line 15 : "Intuitively one might expect an increase of ice export for weaker ice since the ice speed increases." Add "Intuitively one might expect an increase of ice export for weaker ice even during winter since the ice speed increases."

Line 15-16: Change "The ice area export (not shown), however, is smaller for both "weak" experiments during the complete year." for "However, during both summer and winter, the ice area export (not shown) is smaller for both "weak" experiments."

Line 17: "The increase in ice thickness..." This isn't shown in the paper. It would benefit the reader to see maps of mean thickness for your runs and could help you explain better the differences in ice volume, export and even later for your deformation fields.

Line 18-20: I am confused here. You are using an isotropic VP model, yet you are talking about the anisotropic behavior of P. It is also not very clear why the export is less during the winter when the ice strength is weaker. Please expand this paragraph with further explanations.

-- PAGE 7 --
Line 11: Please specify in text what a positive/negative delta_B means. Does a positive delta_B means that there is more ice production and negative delta_B means that there is more ice melting?

Line 25-26: Delete "and also small compared to the volume differences caused by the reduced sea ice export (Figure 3b)." In the run with P*=0.3P*0, it is approximately a third of the changes in the ice volume. It is not small.

Line 27-28: "The results suggest that..." Maybe state that up front in section 3.1 when talking about the sea ice volume changes and say that you explain this in the next sections. Or again, move this section before section 3.1

Line 29: Replace "deformation" with "strength"

-- Page 8 --
Line 28: Why not use the "Lagrangian ice deformation" product directly? Or even the Eulerian ice deformation product?

-- Page 9 --
Line 19: Why using triangles and not a square grid? If I am not mistaken, RGPS uses a square grid to calculate these integrals. Also, the error associated with the estimates of deformation are greater when using triangles than with squares. See Thorndike, Kinematics of Sea Ice, Chapter 7 in The Geophysics of Sea Ice, NATO ASI Series, vol 146, 1986. In particular: section 5.4.5 - Errors in Estimating the Large Scale Deformation.

Equations (3) : Do you compute these integrals assuming u/v vary linearly between each corner? Please specify.
* * *
Line 4: In what sense do you associate a total deformation of 1 day^-1 to a deformation of 100%? What ratio are you taking to find a percentage?

Line 17-18: Put this sentence before the last one? It is really referring to the fact that you are putting everything on the same grid, not that some runs are under-sampled or oversampled.

Last paragraph: Maybe differences in ice thickness could explain this? If the ice is too thick in the model, it will be stronger and you will have less deformations. It would be nice to see the thickness fields.

Line 30: Replace "...and model shear is worst." with "...and model shear is the worst."

Line 31: Replace "...and model is best." with "...and model is the best."
* * *
Line 3: Delete sentence "The picture changes when..."

Line 5: Delete ": divergence, shear and vorticity."

Line 9: "...its deformation distribution is most consistent with RGPS observations." On what basis? PDFs? Spatial Patterns?

Line 16-17: Delete sentence: "The representation of large-scale sea ice deformation..."

Line 18: What is the black contour? How do you define seasonal ice? Please mention in your text.

Line 22-23: "The model sea ice strength P, as defined in Equation 2, depends linearly on ice thickness h. Clearly the linear relationship between P and h is not suitable to realistically model sea ice deformation." As mentioned earlier, the problem here could be instead that the model has too thick ice in the seasonal ice zone....

Line 24: "Models with more ice thickness classes often use a P ~ h^(3/2) formulation (Rothrock, 1975; Lipscomb et al., 2007)" Doesn't this mean that you make ice more stiff? This will not fix the problem that you do not have enough deformations in the seasonal ice zone... it will in fact make it deform even less.

What I see is that the problem here is that your seasonal ice (supposed to be thinner) may be too and not deforming enough... Can you show a map of sea ice thickness? Increasing the dependence of P on h will not help this problem, since stronger ice deforms less and leads overall to an ice pack that is thinner (see Steele et al. 1997 for example).

Line 31: "for visual clarity the period means... " Not clear... Does this apply to figure 8a only? If so, then maybe write something like :
"Figure 8 shows (a) the period-averaged sea-ice deformation rate D_dot, and (b) the monthly-mean seasonal cycle of D_dot (both computed with all 20 RGPS periods available)."

 -- PAGE 12 --
Line 1: Are these numbers the total mean? Please specify.

Line 5: Again, I would check the differences in the thickness field to see if it can explain the differences between your runs. Also, the fact that your model seems 50% too low in deformation could again be linked to the fact that the ice in your model is generally too thick, too strong...

Line 11: March instead of May?

Line 12: Replace "and shows a small but, compared to RGPS data, not very pronounced minimum during March." with "and shows a small but not very pronounced March minimum compared to RGPS data."

Line 13: Delete sentence "That is, the 4.5km solution..."

Line 17: Delete sentence "Again the 4.5km solution..."

-- PAGE 13 --
The discussion on Q could maybe be combined with section 4.3.1?

Line 12: Can you give more details about the implications of having an enhanced seasonal cycle of Q in the model?

Line 27: Here do you compute the deformation rates D_dot from the triangulation of the RGPS positions? Or do you use the Eulerian grid of the model? Please clarify.

-- PAGE 14 --
Line 3: Replace "find an in magnitude about 50% lower scaling exponent (i.e. b ~ −0.12 during winter) for the deformation rate." with "find the magnitude of the scaling exponent to be about 50% lower (ie, b approx -0.12 during winter) for the deformation rate."

Line 8: "...the mean sea ice deformation rate" Monthly means?

Line 10-12: As you can see here with your mean deformation rates, you have much higher values than in figure 8 because you are considering regions of very high strain rates (probably near the coast and in the region of the transpolar drift)... If you are to compare those number with RGPS, you have to bring everything on the same domain covered by RGPS only.

Line 13-14: "Some years, e.g., 1997–1999, have clearly reduced summer deformation rates in comparison to, e.g., the beginning of the 1990s or 2007 and 2008." This is not very clear to see on the figure... Maybe plot winter average and summer average on Fig 10 (a) and (b) instead of monthly means?

Line 14-15: Delete sentence "The deformation rate during 2008..."

Line 19: "daily mean", Maybe use a 3-day period to be as close as possible to RGPS?

Line 20: "the power law scaling exponent b is estimated to be −0.54." Maybe you should show the graph with all the daily mean deformation rates as a function of L and plot the regression line you find. It would make it more clear as to where that number comes from.

Line 20-21: "Figure 10b shows the deformation rate time series for the three model solutions normalized to a length scale of L = 10 km, using the estimated scaling exponent b = −0.54" How do you do this normalization to a different length scale?

Line 23-24: "If looked in detail, however, there remain some quite large differences." This is really not clear on figure. Maybe, as suggested earlier, if you present season means in the graph it would be more clear and we could see better the differences.

-- PAGE 15 --
Line 2-3: "The scaling exponent b gets more negative for weaker sea ice and approaches zero for very strong sea ice, i.e., thick ice and 100% ice concentration" Maybe you need to explain clearly what is the relation between b and Fig.10 b and c. It is the spacing between the curves, ie the larger the space, the larger the slope?

Line 6: Replace "even at 100% ice-cover a cell should show power-law scaling behavior." with "a cell should show power law scaling behavior even with a 100% concentration."

Line 7-8: Why is that? So then, can we really expect to find a power-law scaling in winter, when concentration is almost 1 everywhere?

Line 9: Replace "free ice drift" with "free-drift ice"

Line 15: Replace "the b values of" with "the values of b of" and replace "b values between" with "the values of b between"

Line 17-18: Why not start the section with this? And then say that the value of b is dependent on the ice concentration and thickness, so that if you consider different regions in the Arctic you end up with different b's. And then present your results when considering the whole Arctic domain.

Line 30-31: "model output was bin-averaged to the same spatial scale, L = 12.5 km," What does that mean that the data is bin-averaged? Please explain method.

-- PAGE 16 --
Line 5: "A linear regression was applied to the PDFs in log-log space 5 for the deformation rate range 0.03–0.8 day−1, shown as dashed lines in Figure 11." Not very visible on the graph. Could be removed or offset.

Line 25: Girard et al. 2009
* * *
Line 5: Replace "(ice growth equals ice export)" with "(ie, when ice growth equals ice export)"

Line 10: Ocean sensitivity was never really mentioned in the paper... Delete this sentence?

Line 11: Replace "more deformation" with "more deformations"

Line 11: "the ocean mixed layer depth increases during winter time." This was not shown.

Line 14: Add "Deformations in Arctic ocean and sea ice simulations..."

Line 20-21: "The largest difference occurs for the magnitude of divergence, which is 67% to 79% too low (Table 4)." I do not recall seeing this clearly stated in the discussion. Please add.

Line 26-27: "This suggests a shortcoming of the ice rheology, for example, the linear dependence between ice strength and ice thickness." Not necessarily... Again, you have to check the ice thickness first. It could be due to the fact that your seasonal ice is too thick.

---

## Author Response (AR1)

Answer to review RC2 and RC3 for the manuscript "Sea Ice Deformation in a Coupled Ocean-Sea Ice Model and in Satellite Remote Sensing Data" by G. Spreen, R. Kwok, D. Menemenlis, and A.T. Nguyen

Dear Anonymous Referee #1,

Thank you for raising some concerns about our manuscript. It is our understanding that some of your concerns already could be reduced by the clarifications we made in SC1. These clarifications will be included in the revised version of the manuscript.

Find below your comments in blue and our answers to them in black. We addressed as many of them as possible. We do not agree with all your comments and some of them are in contradiction to recommendations by reviewer #2, which, if in doubt, we will follow. We have, however, incorporated as much of your criticism in a revised manuscript as possible. Especially, the power-law scaling part of the manuscript was rewritten and extended.

**1 General comments**

This paper is essentially split into two parts. The first part discusses how modifications of the model parameter $P*$ affect modelled sea-ice volume, export, and production and melt. The second part analyses the modelled sea-ice deformation in differently resolved model runs, comparing it to observations using the RGPS set of observations. The two parts are poorly linked, even though the authors do point out that such a link is possible. The splitting of the paper into two parts like this is cause for concern. An immediately obvious way to link the two would be to analyse the deformation patterns of the two low $P*$ runs used in part one in part two as well. With the current set-up I would recommend spliting the paper in two and expanding on each part. As it stands I will review the two parts of the paper independently, since this makes the most sense to me.

The main part of the manuscript is the model to satellite data comparison, which you and the other reviewers seem to agree have the strongest impact. We therefore changed the order of the manuscript and start with the model to data comparison. The influence of the ice strength parametrization on the model results will follow that.
The same 18 km simulation is used in the RGPS comparison and P* sensitivity study part. As expected the simulations with reduced P* show higher deformation rates. We do not want to focus on that but rather write about the maybe not so obvious effects on the Arctic ice export and ice mass balance.

**2 Specific comments**

**2.1 Part one**

In part one the authors consider the effects of decreasing $P*$ on modelled sea-ice volume, export, and production and melt "to motivate the importance of sea ice deformation for the Arctic sea ice mass balance". I'm not sure the second part really needs this motivation. To me, seeing if we're modelling the deformation correctly is motivation enough. The question

of to what extent modelled deformation affects the sea-ice mass balance is also interesting enough on its own. I am, however, not convinced by the approach taken by the authors. There is no comparison to observations or estimates so I don't know whether the normal P∗ is even giving a reasonable deformation rate or mass balance, or how changing P∗ affects the deformation, other than the deformation rate.

The performance and realism of the 18 km solution has been assessed in detail in Nguyen et al. (2011). The mass balance is well reproduced in comparison to observations.

We can't compare figures 2b and 8b either, since the deformation is calculated over different areas for the two. So while we can see that changing P ∗ does affect the deformation rate we don't know how it affects various other properties of the deformation. It is therefore also not clear (to me at least) that P ∗ is an appropriate tuning knob to get the deformation rate right. It is a possible one, but more work is needed to show that it is an appropriate one.

P* is commonly used as tuning knob in current VP/EVP models not only to get the deformation but also circulation in better agreement with observations. We do a sensitivity study by varying P* within the range of published values and look at the consequences. We do not claim that P* is the only knob to improve the modeled ice deformation. Actually the opposite. Fig. 8 (now Fig. 5) shows that there is a stronger contrast in ice deformation between observations and model for seasonal, i.e., thinner ice. This contrast cannot be removed by changing P* alone.

Also, what happens to the deformation rate once the model has spun up properly after changing P ∗? This is not clear, since figure 2b shows the deformation rate for 1992–2009, which is arguably a period of transient response as discussed below.

The difference in the seasonal cycle of the deformation rate is similar for the second half 2000-2009 (see figure below) as for the complete simulation shown in Fig 2b (now Fig. 11b). There is only a small change in the difference of D during the simulation period (see below). We added the deformation rate difference time series below to Fig. 11 and discussed it briefly in the text. Anyway, for the short sensitivity experiment section we actually like to focus on the effects on the sea ice export and ice production/melting as this, in our opinion, was not looked at yet. That D is increasing when P* is reduced is not really exciting or surprising.

[Figure]

[Figure]

This leaves us with nothing much to judge the results of this experiment. It doesn't help that what we're looking at is essentially the model's transient response to a large change in its internal mechanics. Normally one would spin the model up to see the effect of a lower P∗ on a model in equilibrium, but this is not done here.

We agree that it would be better to look at the solutions after some spin up phase. If we, however, would look only at the time series after a spin-up phase, e.g. after 2000, the three model solutions would already have a very different ice thickness distribution, which is part of the explanation for the found differences between the simulations. All three simulations start with a similar "initial shock". Also the baseline integration was not spun up before but all start from climatology.

The authors claim that the model has reached a new equilibrium after about 8 years, but the difference in "sea-ice production/melting" is still changing rapidly at the end of the model run (figure 3c).

We agree. We were talking about a new equilibrium in ice volume after about 8 years (see Fig. 3a (now Fig .12a). It is true that not all variables reached equilibrium after the complete simulation (as also the real world Arctic probably is not in equilibrium at the moment).

If we knew how the deformation rate changes from 1992 to 2009 and that the model does not capture that, and that tuning P∗ correctly would give the right deformation rate, then we could say something about how simulating the wrong deformation rate gives the wrong mass balance, but the manuscript gives none of those building blocks.

We again only can refer to Nguyen et al. (2011) who show that the 18 km baseline simulation is capturing many aspects of the coupled Arctic ocean-sea ice system quite well. We therefore consider the strong deviations of the mass balance of the "weak" P experiments from the baseline a degradation. Anyway, we are more interested in the sensitivity of the modeled mass balance on P* here not in finding the "best" P* value.

In terms of analysis of the low P∗ runs the authors also miss what must be in my opinion the most obvious cause for increase in volume, and that is thickness increase due to

excessive convergence. This is also pointed out by Steele et al . (1997), who performed a similar experiment.  When the ice is artificially weakened (which is what we  should consider is happening when using 30% of P ∗) it can be expected to ridge excessively and pile up at the north-Greenland and Canadian coasts. This effect is completely ignored by the authors, even though Steele et al. (1997) discuss it quite nicely and the authors cite that paper. In particular, the authors state that "[o]verall, the decrease in ice export E⁻ for both "weak ice" experiments explains most of the sea ice volume increase in the Arctic Basin shown in Section 3.1" —- a statement which seems to contradict the results of Steele et al. (1997) without giving due consideration to the piling up of ice. The pile-up of ice is, in my opinion clearly what causes the increased "sea-ice production" that the authors note in section 3.3. From the text it seems clear that the authors consider the sea-ice production(/melt) to be thermodynamic production, but there is no reason to assume that this is the case. Without considering the ice pile-up the analysis of the difference in "sea-ice production/melting" is deeply flawed.

We hope that this criticism was in large parts already resolved by our clarification in S1: "We do not consider the change in sea ice production/melt of the "weak" ice experiment in section 3.3 to be a thermodynamic process (and also do not write that in our opinion). We agree with the reviewer that this a combination of dynamic and thermodynamic effects. We agree with the reviewer and Steele et al. (1997) that dynamic ice thickening due to increased convergence for the weaker ice is causing the increased ice production, especially at the beginning of the experiment when the ice thicknesses is similar for all three experiments. As written in the introduction we want to add (not contradict) to the analysis of Steele et al. (1997) by also taking changes in ice export into account, which in our opinion was not done before. If the ice just would get dynamically thicker but the circulation, i.e., ice speed at the ice export gates would stay the same one would observe an increase in ice export. This is not the case. See also the discussion of possible different sea ice flow states in Hibler et al. (2006). The winter reduction in sea ice export as shown here is a positive feedback, which increases the sea ice volume for the weak experiment (in addition to initial dynamic sea ice thickening)."
We added this discussion to the revised manuscript at the end of section 4.3 and in 4.4.

**2.2    Part two**

Part two of the paper has, in my opinion much more potential than part one. It is really what I was hoping to see when I read the title and agreed to review the paper. In my opinion the title belongs to part two and part one should be relegated to a different paper.

We focused the revised manuscript more on this part in agreement with also the other review and changed the order of the sections and start with the model to data comparison. We also improved and extended the power-law scaling section.

In part two the authors compare the results of differently resolved model runs to the RGPS observations. This is a worthy goal and I would be very interested in a more detailed and thorough analysis of the high resolution MITgcm model. This could function as a continuation of the work done by Girard et al. (2009,2011), and a contrast to that done by

Bouillon and Rampal (2015b) and Rampal et al. (2015). I know there are a number of people within the sea-ice modelling community who hope and believe that running an (E)VP model at a higher resolution than Girard et al (2009) did will give better results than what they got. It is, therefore particularly interesting to know whether the results of Girard et al. (2009) hold for the 4.5 km resolution and to get an independent verification, or contradiction of the results for lower resolutions, as well as an indication of the resolution dependence. Unfortunately the current analysis is inferior to that performed by Girard et al. (2009,2011), Bouillon and Rampal (2015b), and Rampal et al. (2015). The authors of the current work mainly base their conclusions on monthly averaged deformation, which is inappropriate, and on visual and qualitative inspection of the simulated and observed deformation fields.

Also this part hopefully should be clarified by our comment S1. We did not use monthly values for the analysis: "The ice deformation analysis in section 4 are not based on monthly statistics. All analysis use the simulated RGPS dataset described in section 4.2, which has an about 3-daily time resolution. We then aggregate all deformations over one month (e.g. in Figs. 4-6 (now Figs. 2-4)) to not show a single day or show a noisy time series (e.g. Fig 8) (could be changed to other time ranges if important but would not change the results)." We made that more clear in the revised version by adding "monthly averages based on 3-daily deformation rates" were appropriate.

They should instead use the quantitative statistical tools and metrics previous authors have used. This would have made for much more solid conclusions and results that are quantitatively comparable to observations (e.g. Marsan et al., 2004 or Stern and Lindsey, 2009) and the model analysis mentioned above.

As explained above our analysis was comparable to the ones performed in the references you cite here. Especially, our PDF analysis is similar to the one in Girard et al. However, to also look into the spatial power-law scaling behavior (new section 3.2.1) and not only the PDFs we followed the procedure described in Stern and Lindsay (2009) for RGPS data and applied it to the model and RGPS data. For the spatial power law scaling between different model solutions (section 3.2.3) we are referring to a suggestion from Stern and Lindsay (2009) to use the power-law relationship to compare datasets with different resolution. We clarified that at the beginning of section 3.2.3. For the revised manuscript we extended the power law scaling analysis as both you and the second review found this part particular important.

I want to stress, in particular that using monthly averages when studying deformation is inappropriate, since nearly all of the deformation happens at a much shorter time scale. This is a major problem with section 4.3.1. If the authors want to consider long-term differences in deformation then figure 7 is a more appropriate approach than figures 4, 5, and 6. I would even recommend taking a multi-month or seasonal average instead of only one month, in that case.

Clarified now. We do not use monthly means.

It is interesting how large the difference in deformation rate is between the seasonal and multi-year ice is.

I'm also left wondering if the deformation rates used in section 4.5 are monthly averages or not. Using monthly averages there would be inappropriate for the same reason as before, although it is not immediately clear how large an error we get using monthly averages in this case. Should the results in section 4.5 hold then they are a very interesting contradiction of the results of Girard et al. (2009).

Also in section 4.5 (now section 3.2.2) we did not use monthly averages.

It does seem strange though, that the authors choose not to remove the noise of the RGPS data as prescribed by Bouillon and Rampal (2015a). They need to either remove the noise or justify not removing it.

Our analysis actually was done before the Bouillon and Rampal paper was published. Also currently the RGPS data is still available in its current form and still used in many studies. We prefer keep doing our analysis with the original RGPS data even if this means that they are a bit noisy. Qualitatively we would not expect different results by removing the noise. We mention the Bouillon and Rampal (2015) paper several times and that the artificial noise in the RGPS data could explain some of the differences in the absolute amount of deformation rates between RGPS and the model solutions.

It is also inappropriate to consider the percentage of area containing 80% of the deformation as a measure of localisation (section 4.3.3). It should be the largest 15% of the deformation, like Stern and Lindsay (2009) use. Using 80% of the deformation you essentially include all the deformation so this is no longer a measure of the localisation of deformation. The way it stands the metric is essentially meaningless.

We followed this suggestion and calculated the percentage of area containing the highest 15% of deformation rates. The new Section 3.1.3 was rewritten and Figure 7 exchanged with a new one.

The authors also do the power law scaling of deformation rate incorrectly (section 4.4). They use different model realisations (i.e. 4.5, 9, and 18 km resolutions) to determine the scaling, but the correct thing to do is to use a coarse graining method (like the authors named above) and calculate the scaling based on it. The authors of this manuscript argue that the high resolution model gives better results than the low resolution ones, but they then combine all three to calculate the scaling. This makes no sense.

The purpose of this exercise was to study how one can compare deformation rates originating from models with different grid resolutions, which we consider a common problem. Applying a power law one can bring the deformation rates closer together. Comparisons then might be possible if large scale (model domain) and long-term (yearly) averages are compared.  We, however, also clearly state that this is by far not ideal due to the strong seasonal dependence and dependence on ice concentration and thickness and that statistical comparisons might be more appropriate. We stressed that even stronger in the revised version of the manuscript and added some more explanation add the beginning of section 4.4 (now section 3.2.3).

**3    Conclusions**

I am sorry to say that I will be recommending that this paper be rejected publication  in The Cryosphere. The reasons for this decision are the poor structure of the paper, it being split into two unrelated parts, and the substantial shortcomings of both parts. This is quite disappointing since I believe that the comparison of the MITgcm results with RGPS data could be very interesting indeed. My recommendation to the authors is to thoroughly review Girard et al (2009) and the related literature, and then to revisit part two of the manuscript with the aim to refute or support the conclusions of Girard et al (2009) in the case of the 4.5 km resolution simulation, give an indication of the resolution dependence, and to provide contrast with the results of Bouillon and Ram- pal (2015b) and Rampal et al (2015).  If this is properly done then that would make  for an interesting paper and one that would be important for further evaluation and development of dynamical sea-ice models.

We hope that we could dispel and clarify some of the concerns the reviewer had. We believe that many of them were based on misunderstandings and we worked hard on making the revised manuscript clearer and easier to follow. We restructured the manuscript as described above. As the second reviewer recommends to keep the P* sensitivity study we did not remove it as suggested here but rather move it more to the end of the manuscript. We extended and recalculated the power-law scaling analysis. The PDF results stayed the same but the power-law dependence on spatial scale was added to the analysis. Together with the changes proposed in the answer to the second review we hope that a revised version will receive the reviewer's approval.

**4    References**

All the papers I refer to here are already cited in the paper, with the exception of Rampal et al (2015), which is still under review at The Cryosphere Discussions: http://www. the-cryosphere-discuss.net/tc-2015-127/

Answer to RC4 (Bruno Tremblay and Amelie Bouchat) for the manuscript "Sea Ice Deformation in a Coupled Ocean-Sea Ice Model and in Satellite Remote Sensing Data" by G. Spreen, R. Kwok, D. Menemenlis, and A.T. Nguyen

Dear Bruno, dear Amelie,

Thank you very much for your detailed and very helpful review of our manuscript. Find below your comments in blue and our answers to them in black. We will follow them closely and think that we can address almost all of them in a revised version.

In this paper, the authors present: 1- a sensitivity study of the simulated sea ice mass balance on the sea ice strength parameterization and 2- a sensitivity study of the simulated sea ice deformation (divergence, shear, vorticity) on the spatial resolution of the model. The model is the coupled ice-ocean MITgcm with a two-category ice thickness model and a viscous plastic sea-ice rheology. The pressure term in this model is the standard parameterization of Hibler (1979) with a linear dependence on h and exponential dependence on sea ice concentration.

The authors show that a lower ice strength parameter leads to a reduced net annual ice export through Fram Strait and an overall reduced ice production in the simulations after 8 years of integration. They show that the reduced ice export is the dominant mechanism explaining an increase in ice volume in their runs with reduced ice strength. They conclude that the ice mass balance in coupled ice-ocean models is very sensitive to the value used for the ice strength parameter.

In the second part of the paper, they compare their simulated deformation fields (divergence, shear and vorticity) at different spatial resolutions with the Radarsat Geophysical Processor System (RGPS) satellite observations on the basis of their spatial patterns, power law scaling and probability density functions (PDFs). They find that the simulated deformations with the highest spatial resolution (4.5 km) agree best with observations on all metrics tested. However, they show that the model does not capture the enhanced deformations (magnitude and spatial density) in the seasonal ice zone at any spatial resolution and that it has a mean total deformation rate that is about 50% lower than observations. The authors attribute this shortcoming to the ice strength formulation being linearly proportional to the ice thickness. On the other hand, they are able to reproduce the power law scaling of the total deformation rate with the spatial resolution as observed in RGPS observations and the PDFs also agrees with those of RGPS – but are in contradiction with results from Girard et al 2009.

The paper is generally well written – despite some awkward sentence structures and typos (see specific comments below). It presents a long-awaited (re) analysis of the scaling law for sea ice deformations simulated by viscous plastic sea ice models – with results that are contrary to what was published in Girard et al. but that are in accord with several other modeling groups that have done similar analysis. This paper constitutes a welcomed clarification. The results on the effect of the sea ice strength parameterization on the sea ice mass balance are also insightful. Given that the Arctic is transitioning to a seasonal ice cover, and that current rheological

models do not simulate the correct deformation characteristics of the seasonal pack ice (as reported here) is interesting.

The tone of the paper should be less a little less defensive and/or more assertive. The paper presents very interesting results. Those new results need to be prominent. For instance, negative results are presented first followed by positive results. The particular is presented before the general. The results that cannot be compared with observations are presented first followed by the results that can be compared with observations. All of this makes the key findings of the paper more difficult to find and appreciate. More specifically, a key finding of the paper (one that is buried deep in the paper) is that the simulated sea-ice deformation simulated by a viscous-plastic model follows a power law - contrary to what was presented in Girard et al 2009. The results presented in Girard et al. 2009 cannot be reproduced by the authors nor by any other modeling group in the community, yet it has become common (accepted) knowledge that VP rheologies do not follow a power law. This must really be stated early on and clearly. More suggestions regarding this issue are listed below.

Thank you for this comment. We restructured the paper following these lines. We changed the order of sections 3 and 4 and start with the model to data comparison first, as both reviews agree that this is the most important part of the paper. Also the order of four sub-sections was changed to allow a better flow of the results. The power law result will already be mentioned in the abstract. The new power-law section 3.2 was expanded to also look at the spatial scaling of absolute divergence following the procedure by Stern & Lindsay (2009).
We added a figure showing the ice thickness and related discussion. The figure and discussion in section 3.1.3 was changed and now addresses the "Localization of LKFs" using the highest 15% deformation criterion.

We recommend that the paper be accepted for publication after having addressed the comments below carefully.

Amelie Bouchat, PhD candidate
Bruno Tremblay

**Major Points**:

1.   Page 6: general comment: Since the ice export depends on ice thickness in the central Arctic. I would discuss the change in ice thickness in the Arctic with changing P* first. Then I would discuss the change in ice export. I understand that it is a chicken and egg situation, but still ice will thicken in the Arctic irrespective of lower export because of weaker ice. The lower export is a positive feedback of the increase in ice thickness – i.e. the increase in ice thickness does not compensate for the reduction in sea ice velocity. Now we are reading the paper about the export changes without knowing all a-priori knowledge.

We changed the order of sections 3.2 export and 3.3 ice production/melting and

now first discuss ice thickness changes.

2. A discussion of the ice thickness distribution should be included in the manuscript. The fact that the deformations in the model are generally too low in magnitude and too sparse maybe due to the fact that the ice is too thick. This may also explain why the deformations in the seasonal ice zone are too weak.

Figure 1 shows the spatial distribution of ice thickness for the 9 km model solution. We added a sub-figure showing the ice thickness distribution for the three model solutions to Figure 1. This subfigure is referenced when the too low deformation rates are discussed. For a detailed comparison of the model ice thickness to measurements see Nguyen et al. (2011), who use the same 9 km model solution as presented here and find a good agreement with observations.

3. We disagree with the interpretation from the authors that the discrepancy between RGPS and the simulated deformation in the seasonal ice zone is necessarily due to the linear relationship between P and h. A map of the simulated ice thickness for March and September for different ice strength would be useful to better understand this issue.

We reformulate this statement to become more a hypothesis. We did some test with changed relationships, which support the hypothesis but this would be a different study and we don't think it is necessary to discuss this in detail here.

4. Page 11, line 23: I am not sure we can blame all of this on the linear h dependence of P*. The ellipse results in equally large viscous coefficients (eta and zeta) for the same divergence (in absolute value) and for a given shear. In reality, sea ice would interact little with other ice floes when we have divergent sea ice motion. I would think that in the seasonal ice zone, where there is more space for the pack ice to expand (in regions of coastal polynya, etc), an elliptical yield curve and normal flow rule that gives unrealistically large viscous coefficient in divergence, would lead to reduced deformation as you see here. This is jus another possibility. The point is that I do not think that this can simply be related to the linear dependence of P on h as discussed here.

We changed these sentences to:
"This discrepancy between seasonal and perennial ice hints to a shortcoming of the sea ice rheology used in the simulations. To first order the main difference between seasonal and perennial sea ice is the ice thickness. The model sea ice strength P, as defined in Equation 2, depends linearly on ice thickness h. This is the typical P formulation for a VP or EVP sea ice rheology with two ice classes and might not be the best representation of the P to h relationship. Models with more ice thickness classes often use a $P \propto h^{3/2}$ formulation (Rothrock, 1975; Lipscomb et al., 2007), which can be considered more realistic. There are, however, also other differences between the seasonal and perenial ice zone than the ice thickness. The proximity to open water, for example, will allow more cases of ice divergence at the ice margins

than in the ice pack, which might be less well represented by the VP rheology."

5.  Page 15, line 20: Start your discussion here where you analyze the results for the same geographical region as that of the RGPS. Then you discuss the caveat associated with including points close to coastlines. I.e. you go from General to specific. The way it is presented is a little defensive (i.e. you show the problems first and then show what works well). These are very nice results, one that is in conflict with that of Girard et al. but in accord with results from all other sea ice modeling groups. The authors need to make this point more prominent. I would say this point is one of the highlight of your paper and finally clarifies this situation.

We changed the order of Sections 4.4 and 4.5 (now sections 3.2.2 and 3.2.3) and first show that the PDFs of the model solutions in general follow a power law comparable to the RGPS observations. This is then followed by the section where we use a power law to make the deformation rates of the model solutions with different grid spacing comparable. We added a new power-law scaling analysis as section 3.2.1 looking at the spatial scale dependence of the modeled and observed divergence (the same dataset is then also used for the PDF discussion in 3.2.2 to be consistent). Large parts of the power-law section 3.2 were reformulated to make the findings more prominent.

6.  In section 4.4, I would discuss the case where you compute the scaling exponent with same domain as RGPS first, since this is what you are interested in to compare with observations. Then when you know you are doing fine, you can go and discuss the fact that this scaling exponent depends on ice concentration and thickness. Also, 3-day means should be used instead of daily means of deformation to have data as similar to RGPS as possible for the comparison.

In section 4.4 (now section 3.2.3) we use the power law dependence of deformation rates to make deformation rates of model solutions with different grid spacing comparable. This approach cannot directly be compared to the RGPS data as also other factors than the model grid spacing will influence the deformation rate between the three model solutions.
We therefore changed the order of sections 4.4. and 4.5 (see last comment) and now start with the power law behavior of the probability density functions, which can be directly compared with the RGPS data. We also added some clarifications to the beginning of this sub-section.

**Minor Points - A**:

Page 5, line 2: define shear and divergence. They are defined but only much later in section 4.2.

Added reference to strain rate definitions

Page 6, line 18: It is not clear what the authors are referring to by "anisotropic behavior of sea ice". The authors are using the standard Hibler rheology which is isotropic. This should be clarified.

Yes, there is no sub-grid scale anisotropy and "anisotropic" probably was not the best word here. We were referring to the irregular distribution of ice stress causing e.g. LKFs and ice arches. Changed in manuscript.

corrected

Page 6, line 19: These are important sentences. They must be expanded. Describe the ice arching. Show example in a figure? "Leads to change in the sea ice circulation". This is vague. What kind of changes? How are they link with ice export? The paper is about P* and ice export. These must be documented.

We did not explore changes in ice circulation within the Arctic Basin in detail. This was already done by Steele et al. (1997) who find an acceleration of the Beaufort Gyre and a stronger piling up of ice at the coast of North America for reduced P*. We agree that this manuscript should focus on ice export. We added some more discussion about thermodynamic vs. dynamic ice thickening. However, the main part of the paper is about the model to RGPS data comparison. For a more detailed discussion about changes in the force balance, Arctic Basin thickness, and circulation we can only refer to Steele et al. (1997). They, however, do not consider changes in ice export, which we show to be one of the main contributors to the observed changes within the Arctic Basin (which we will not explore in detail here).

Page 6, line 20: Again vague statement. What fraction is due to arching, and what fraction is due to changes in the sea-ice circulation. This must be quantified.

See last answer.

corrected

Page 6, line 30: Is it really interesting to quote the total (sum over years) difference in ice export? I would prefer to see the new equilibrium numbers in km/yr.

We like to explain the change in ice volume of 6700/870km3 at the end of the integration period. This change has accumulated over the complete time period. We divide the causes for this volume change in (a) changes in ice production within the Arctic Basin and (b) in changes in ice export. Therefore, the integrated change (sum over years) in ice export is used here.

Page 7, line 2. No it should be discussed first. The fact that the change in export cant totally be discussed at this stage suggest that the order should be changed.

Order of sections 3.2 and 3.3 have been changed

Page 7, line 5: "...sea ice export (E^bar)..."

added

Page 7, line 15: I am guessing the export must increase since the ice strength is lower and that the ridging more than compensate for this in the first 5 years. You need to discuss the ice export variation in this part of the paper.

The ice export is not changing significantly between the three experiments during the first two years. Note that the export E is removed for the calculation of the ice production B. We will add the sentence: "This causes the ice production B to increase compared to the baseline. B is corrected for the influence of ice export E, which, however, does not change much from the baseline integration during the first two years (not shown).

Page 7, Line 23: This is counter-intuitive. I would have expected an increase in the ice volume export. Again, two opposing effects are at play: increase ice thickness and reduced ice velocity. A few additional words should be included to clarify this.

Yes, the reduction in ice export is not directly intuitive. Therefore, we describe the different effects leading to it in a separate section 3.2 and only reference to it here.

Page 8, line 5: Give many examples or kill "e.g."

Removed "e.g."

Page 8, line 9: The best value for P* is traditionally found minimizing the error between the simulated drift and the observed drift using models where the wind forcing is specified as observed. Of course biases in the thickness field will impact the optimal P*. But in principle, a model that assimilates sea ice concentrations, and ice thickness from satellite and forced with reanalysis data could be used to find an optimal value for P*.

We used a method similar to what you describe to find the optimal P* value for our baseline integration ("Greens Function approach"). That information was added to the text.

Page 8, line 12: give references.

added

Page 8, line 28: This should read "from the simulated ice motion dataset..."?

No, we are still talking about the observed RGPS SAR ice motion here.
Clarified in text.

Page 9, line 6:"...since November 1996 until 2008..."

done

Deformation rates higher than 1 are considered outliers. Clarified in text.

Page 11, line 29: Define the periods here as well (not just in the Table)

Maybe we misunderstand what you mean. Table 3 lists 20 periods. To include them all in the text would be hard to read.

done

Page 12, line 13: "…slightly differs from this general behavior…"

done

Page 12, line 12: This sentence is not English. "… shows a weak minimum in March in contrast with the RGPS data…"

done

Page 12, line 23: Is the model iterated to convergence? We see much better defined LKFs in a model that was iterated to convergence compared with one that was not, see for instance Lemieux Tremblay (JGR). I am curious if this has an impact on your simulation results.

We did not perform explicit tests regarding the convergence of the model. We, however, discussed the Lemieux Tremblay (JGR) paper and concluded that our iterations should be sufficient. In particular, please note that our integration time step is rather small (240 s) and that we use $10^{-4}$ convergence criterion for the iterative LSR solver.

done

Information added.

Page 14, line 2: missing word or one word too many. "…find an in magnitude…"

corrected

Page 14, line 18: When we do best linear fit in log-log scale the error for large D will be underestimated. I.e. you best fit will preferentially minimize the error for the small D. Can you comment on the impact of doing this?

We are not completely sure we understand the question. For large D the number of observations gets very small and therefore the scatter large. We therefore stop our fit at 0.8. We do not aim to minimize the error for larger D

If you are talking about doing the fit in log space and therefore having a non-linear distribution of D than you are right, the fit will preferentially minimize the error for small D. We cannot comment on the impact on that because that would depend on the question. One has to keep in mind that the probability is also scaled logarithmic and therefore there are many more observations with small deformation rates, which one could argue therefore should have a higher impact on the fit.

Page 14, line 22: typo. Missing dot in -0.54.

corrected

Page 14, line 18: You have already said above that there is a constant b value in the winter and a higher b value in the summer. I.e. we cannot just use a constant value. Why test the constant b case if this is so? Eliminate this part? Or say why you still want to look at it.

Yes, we agree one cannot use a single scaling exponent b to make detailed comparisons between strain rates from models with different grid spacing. As Fig 10 a and b demonstrates using a power law with constant b is still useful to compare mean (complete domain, yearly) strain rates of models with different resolution. While the reproduced details in sea ice deformation are very different between the three solutions, Fig 10b demonstrates that the mean deformation rate of all solutions is quite comparable if one takes the different grid scales into account. Added some more information about this to the text.

Page 14, line 20: "…approaches zero linearly…" instead? "…for 100% ice-covered … the deformation rate decreases exponentially". The part of the sentence "but in a more exponential way" is colloquial English.

Reformulated

Page 15, line 8: It is not clear why A=1 would prevent the power law to exist. The exponential dependence of P on A is a continuous function. Why are we loosing it only for A=1?

As said in the manuscript we do not have a clear answer to that. From theory one should expect the power law scaling to also exist for 100% ice cover. We are not saying that we are losing the power law scaling just for 100% ice the power law scaling exponent is converging to zero for high ice concentrations. We can only speculate that this has to do with the exponential dependence on the ice concentration in the model implementation. We remove this discussion from the manuscript as it is not conclusive at the moment.

Page 15, line 12: "geographic location" is not a physical parameter. I think you mean, that the power law exponent depends on the "mean internal ice stress" which is higher when we are in the proximity of continents.

Yes, reformulated

Page 17, paragraph starting at line 24: The authors need to discuss what works first and then discuss what does not work. It is the same content, just the order that needs to be changed.

Yes, agreed. The order was changed.

Page 18, line 5: Again the order should be reversed. The authors need to discuss the results using the same domain as the RGPS and then the one where they include the regions close to the coastlines.

Yes, agreed. The order was changed.

**Minor points - B**

Suggestion: "sea ice deformation" should read "sea-ice deformations" in most places in the text. "Sea ice" takes a hyphen when used as a compound adjective.

done

-- PAGE 1 --

Line 8-9 : Replace "All three model simulations can reproduce the large-scale ice deformation patterns but ..." with: "All three model simulations can reproduce the large-scale ice deformation patterns, but small scale sea-ice deformations and linear kinematic features are not adequately reproduced." Then go with "The overall sea ice..." followed by "A decrease in ...".

done

Line 10: Replace "The overall sea ice deformation" with "The mean sea-ice total deformation rate"

done

Line 16-17: "Either way, this study..." Delete sentence.

We prefer to keep this sentence.

-- PAGE 2 --
Line 4-5: Suggestion: Change "or if new sea ice rheologies like the one..." for "or if new sea-ice rheologies (Girard et al. 2011, Sulsky et al. 2007, etc.) have to be used."

Followed your suggestion

Line 6: "(2) brine rejection into the ocean, (3)..." Add "(2) brine rejection in the ocean due to freezing in open water areas, (3)..."

done

Line 13: "were" should be "are"

done

Line 13-15: Suggestion: Change to "The model sensitivity to the model ice strength parameterization is assessed by comparing the model solutions with different ice strength parameters to the RGPS satellite observations spatially and temporally. These comparisons also allow us to study the model uncertainties regarding the sea-ice deformation representation in the current formulation of VP models."

Followed your suggestion

Line 18: "into a mean and fluctuating field" change to "into mean and fluctuating fields"

done

Line 19: "to evaluate models with first order..." change to "to evaluate models on the basis of their first order mean velocity field and it can be correctly predicted even by simple sea ice models..."

Reformulated along the lines of your suggestion.

Line 20: "Second order sea ice deformation fields..." change to "The second order sea-ice velocity field, represented by the sea ice deformation fields (strain rates), has to be used for comparison to take into account the high frequency fluctuations of the sea-ice velocity field and to assess the quality of the sea-ice rheology formulation."

done

Line 24: "For RGPS deformation rates" should be "For RGPS total deformation rates"

done

Line 25: "a scale dependence" should be "a spatial scale dependence"

done

Line 34: Replace "for example they show" with "showing"

done

Line 35: Replace "Some improvement in modeling sea ice deformation" with "Improvements in the modeled sea-ice deformation"

done

-- PAGE 3 --
Line 4-6: "A recent example...." Delete sentence.

Why? The Tsamados et al. (2013) study should be mentioned. We kept the sentence.

Line 11: Replace "We reconstruct the observed sea ice deformation..." with "Using the VP model, we construct simulated deformation fields on the same spatial and temporal scales as in the RGPS observations."

done

Line 12: Replace "In addition we also compare..." with "We then compare the power law scaling properties of the modeled and observed deformation rates (section 4.4)

and we perform a sensitivity study of the deformation fields properties to the model ice strength parameter (section ??)"

Reformulated sentence

Line 13-14: Delete "sea" and "and thereby ice deformation"

done

Line 16: Delete "as a consequence also" and replace "can effect the Atlantic Ocean circulation" with "can also affect the modeled Atlantic Ocean circulation"

done

Line 16-18: "Ultimately, we would like..." Reformulate. Maybe write: "Ultimately, we would like to highlight why the sea-ice strength representation and the sea-ice rheology should receive more attention in models."

done

-- PAGE 4 --
Line 15: "fit to available satellite and in-situ data..." Data of what? Ice velocity? Ice thickness? Please specify.

Added "(e.g. ice drift, area, thickness)" and made a clearer reference in the next sentence that these data and approach is explained in Nguyen et al., 2011

Line 22: "As a consequence these higher-resolution simulations exhibit somewhat larger model drifts relative to observations than the 18-km simulation." Does that mean that therefore you would need to increase P* with increasing resolution to slow down the pack? Please state so if it is the case.

No we are talking about model to data differences here not ice drift. Exchanged "drift" with "deviations"

Line 27: Replace "thus the local ice thickness distribution" with "thus modifies the ice thickness distribution" and change "Furthermore, changes in the model ice strength alter the sea-ice drift speed..."

done

Line 28: Replace "changes in sea ice deformation therefore..." with "these changes can alter the equilibrium sea ice volume in the Arctic."

done

Line 29: Replace "a set of sensitivity experiments" with "a set of experiments" and replace "changes in sea ice deformation to motivate the importance of sea ice deformation" with "changes in ice strength parameter to highlight the importance of using accurate rheological models and sea-ice deformation fields"

Done

Done

Line 32: Replace "The sea ice deformation rate" with "The total sea-ice deformation rate"

done

-- PAGE 5 --
Line 1-3: Rewrite as: ", where nabla_dot is the divergence rate and tau_dot is the shear rate, is used as a measure for the overall sea-ice deformation occurring at a certain point in space (e.g. Stern and Lindsay 2009). The magnitude of both the divergence and shear rates are to some extent controlled by the strength of the sea ice. In our model configuration, we use the typical ice pressure formulation P (or strength) of Hibler 1979:"

done

Line 13: Maybe it would be worth noting that the differences in the values of P* that are used in different models come in part because of the need to calibrate the parameters of one's model depending on the forcing used (ocean + atm.) and drag formulations. There is also the need to recalibrate this P* parameter depending on the spatial resolution used in the model.

Added a sentence along these lines.

Line 13: What is the time step used for simulations?

For the 18 km simulations the time step is 20 minutes.

Line 18: Add "For any given month, the monthly deformation rate D_bar increases..."

done

Line 20: Replace "deformation rates" with "simulations"

done

Line 22: Replace "of these sea ice deformation" with "of changes in the deformation rates and ice velocity on..."

done

Line 25: Delete "will" and "for a discussion of geophysical sea ice volume change over time, see Nguyen et al. (2011)."

Deleted "will" but kept the reference to Nguyen et al.

Line 28: Replace "starts immediately to" with "rapidly" and delete sentence "A similar sensitivity...". Instead, add "Hence, after 8 years of integration, the sea ice volume has increased by 7%..." and continue with sentence from line 30-31.

Moved the sentence "A similar sensitivity ..." further done and followed the other suggestion.

Line 29: Maybe add a sentence here to clearly state that you do have thicker ice in agreement with Steele et al, but what controls the ice volume change in your simulations are the changes in ice export and ice production and melt.

done

Line 33: Replace "quickly diverges from the baseline. The divergence gets..." with "diverges from the baseline at a much faster rate than for the solution with 0.7P*_0. The rate of increase of the ice volume gets smaller after 1999, but the volume keeps increasing until 2005."

done

-- PAGE 6 --
Line 1: Why does the volume start decreasing after 2005 in both runs? And there seems to be much more variability in the case P*=0.3P*0. than with P* = 0.7P*0. Can you comment?

After about 2002 the ice production within the Arctic Basin gets smaller for the "weak" simulations than for the baseline ones. Together with the ice export this leads to a reduction in the ice volume difference, which is discussed in the following. We did not look in detail at the causes for the higher variability of the 0.3P* simulation. However, for 03P* the ice is in close to free drift for more cases and ice speeds can get higher. This should add more dynamic to the system causing more variability.

Line 4-5: Put this sentence in previous section, and maybe add something like "both these mechanisms are explored in the following sections".

Done

Line 5: Delete "also" and add it on line 6 between "experiments" and "diverges"

done

Line 8: Add "Even more pronounced is the change" Delete "however".

done

Line 11: Rewrite: "...(blue shaded area), and during winter, E_bar is lower than..."

done

Line 12: Delete "however" and "large" and replace "overall" with "the net annual"

done

Line 13: Add "nearly balance in the course of one year and this results in a net annual decrease in... "

Done

Line 13: Can the very enhanced seasonal cycle of run with P*=0.30P*0 explain the high variability seen in Fig. 1a of sea ice export compared to run with P* = 0.7P*0?

Yes, probably partly. Also see our comment above.

 If it is the case, then I would suggest moving this section before section 3.1 for clarity.

We moved section "Sea ice production and Melt" before this section now. We like to show the resulting change in sea ice volume first and afterwards explain the reasons. We can see that one also could do it the opposite way. We added a sentence that the explanations will follow at the end of section 3.1.

Line 15 : "Intuitively one might expect an increase of ice export for weaker ice since the ice speed increases." Add "Intuitively one might expect an increase of ice export for weaker ice even during winter since the ice speed increases."

done

Line 15-16: Change "The ice area export (not shown), however, is smaller for both "weak" experiments during the complete year." for "However, during both summer and winter, the ice area export (not shown) is smaller for both "weak" experiments."

Done

Line 17: "The increase in ice thickness..." This isn't shown in the paper. It would benefit the reader to see maps of mean thickness for your runs and could help you explain better the differences in ice volume, export and even later for your deformation fields.

We added maps of ice thickness for 15 November 1999, i.e., quite in the middle of the integration, for the baseline solution to Figure 1. To not extend this section too much we decided against showing thickness maps for the "weak" experiments. Figure 12a, however, shows how the ice thickness in the Arctic Basin increases for the "weak" experiments.

Line 18-20: I am confused here. You are using an isotropic VP model, yet you are talking about the anisotropic behavior of P. It is also not very clear why the export is less during the winter when the ice strength is weaker. Please expand this paragraph with further explanations.

"Anisotropic" was not the right word here. We slightly reworded this paragraph.

-- PAGE 7 --
Line 11: Please specify in text what a positive/negative delta_B means. Does a positive delta_B means that there is more ice production and negative delta_B means that there is more ice melting?

No, delta_B is the net production. Added: "A positive Delta_B means that more ice is produced (thermodynamically and dynamically) for the ``weak'' experiments, a negativ positive Delta_B the opposite. The smoothed Delta_B here represents the net ice production difference including both ice growth and melting and both processes can change Delta_B."

Line 25-26: Delete "and also small compared to the volume differences caused by the reduced sea ice export (Figure 3b)." In the run with P*=0.3P*0, it is approximately a third of the changes in the ice volume. It is not small.

Done

Line 27-28: "The results suggest that..." Maybe state that up front in section 3.1 when talking about the sea ice volume changes and say that you explain this in the next sections. Or again, move this section before section 3.1

Added a sentence along that line at the end of section 3.1

Line 29: Replace "deformation" with "strength"

done

-- Page 8 --
Line 28: Why not use the "Lagrangian ice deformation" product directly? Or even the Eulerian ice deformation product?

This is done to ensure highest possible consistency between the modeled and observed ice deformation. Added a sentence about that.

Line 19: Why using triangles and not a square grid? If I am not mistaken, RGPS uses a square grid to calculate these integrals. Also, the error associated with the estimates of deformation are greater when using triangles than with squares. See Thorndike, Kinematics of Sea Ice, Chapter 7 in The Geophysics of Sea Ice, NATO ASI Series, vol 146, 1986. In particular: section 5.4.5 - Errors in Estimating the Large Scale Deformation.

We agree. Using triangles mainly was done due to easier technical implementation. We remove acute triangles to reduce some of the uncertainties. Using triangles also has one advantage: we can resolve smaller areas for the deformation and the number of observations increases.

Equations (3) : Do you compute these integrals assuming u/v vary linearly between each corner? Please specify.

We are not sure we understand the question. We only have observations at the triangle corners. To calculate the line integral only these are used. No further assumptions are made for the velocities between the corners.
* * *
Line 4: In what sense do you associate a total deformation of 1 day^-1 to a deformation of 100%? What ratio are you taking to find a percentage?

We removed the percentage. Some authors express the deformation rate as percentage.

Line 17-18: Put this sentence before the last one? It is really referring to the fact that you are putting everything on the same grid, not that some runs are under-sampled or oversampled.

done

Last paragraph: Maybe differences in ice thickness could explain this? If the ice is too thick in the model, it will be stronger and you will have less deformations. It would be nice to see the thickness fields.

Examples of the thickness fields for 15 November 1999, i.e., quite in the middle of the integration, for all three solutions were added to Figure 1.

Line 30: Replace "...and model shear is worst." with "...and model shear is the worst."

done

Line 31: Replace "...and model is best." with "...and model is the best."

done

Line 3: Delete sentence "The picture changes when..."

Changed that sentence.

Line 5: Delete ": divergence, shear and vorticity."

Kept that part

Line 9: "...its deformation distribution is most consistent with RGPS observations." On what basis? PDFs? Spatial Patterns?

Added explanation that we do qualitative comparisons here. The quantitative comparisons follow.

Line 16-17: Delete sentence: "The representation of large-scale sea ice deformation..."

Changed sentence

Line 18: What is the black contour? How do you define seasonal ice? Please mention in your text.

The multiyear ice mask is based on QuikSCAT data. Added information to text when it fits is used in Fig 4.

Line 22-23: "The model sea ice strength P, as defined in Equation 2, depends linearly on ice thickness h. Clearly the linear relationship between P and h is not suitable to realistically model sea ice deformation." As mentioned earlier, the problem here could be instead that the model has too thick ice in the seasonal ice zone....

Examples of the thickness fields for 15 November 1999, i.e., quite in the middle of the integration, for all three solutions were added to Figure 1. The modeled ice thickness agrees well with observations (ICESat), see Nguyen et al.

Line 24: "Models with more ice thickness classes often use a P ~ h^(3/2) formulation (Rothrock, 1975; Lipscomb et al., 2007)" Doesn't this mean that you make ice more stiff? This will not fix the problem that you do not have enough deformations in the seasonal ice zone... it will in fact make it deform even less.

No, P ~ h^(3/2) will make ice weaker for thin ice and stronger for thick ice.

What I see is that the problem here is that your seasonal ice (supposed to be thinner) may be too and not deforming enough... Can you show a map of sea ice thickness? Increasing the dependence of P on h will not help this problem, since stronger ice deforms less and leads overall to an ice pack that is thinner (see Steele et al. 1997 for example).

Examples of the thickness fields for 15 November 1999, i.e., quite in the middle of the integration, for all three solutions were added to Figure 1.

Line 31: "for visual clarity the period means... " Not clear... Does this apply to figure 8a only? If so, then maybe write something like :
"Figure 8 shows (a) the period-averaged sea-ice deformation rate D_dot, and (b) the monthly-mean seasonal cycle of D_dot (both computed with all 20 RGPS periods available)."

Yes, only applies to Fig 8a. Followed your suggestion.

 -- PAGE 12 --
Line 1: Are these numbers the total mean? Please specify.

Yes, added in text

Line 5: Again, I would check the differences in the thickness field to see if it can explain the differences between your runs. Also, the fact that your model seems 50% too low in deformation could again be linked to the fact that the ice in your model is generally too thick, too strong...

Examples of the thickness fields for 15 November 1999, i.e., quite in the middle of

the integration, for all three solutions were added to Figure 1. The thickness fields differ slightly but not extensive between the three model simulations. Compared to observations the modeled ice thickness is not too strong. See Nguyen et al. (2011).

Line 11: March instead of May?

No, the 9 and 18 km model D in Fig. 8 is almost constant Jan to May. Excluded the 4.5 km solution from this statement.

Line 12: Replace "and shows a small but, compared to RGPS data, not very pronounced minimum during March." with "and shows a small but not very pronounced March minimum compared to RGPS data."

done

Line 13: Delete sentence "That is, the 4.5km solution..."

Line 17: Delete sentence "Again the 4.5km solution..."

Kept these sentences.

The discussion on Q could maybe be combined with section 4.3.1?

Hm, we think 4.3.1 shows an example and some qualitative discussion. 4.3.2 introduces the deformation time series and 4.3.2 then adds the time series of Q.

Line 12: Can you give more details about the implications of having an enhanced seasonal cycle of Q in the model?

We can only hypothesize and added: "This results in larger differences in Q between model simulations and RGPS during summer and hints towards a degraded performance of the model simulations to represent sea ice deformation during summer.

Line 27: Here do you compute the deformation rates D_dot from the triangulation of the RGPS positions? Or do you use the Eulerian grid of the model? Please clarify.

We use the Eulerian grid her. This is clarified a few lines below.

Line 3: Replace "find an in magnitude about 50% lower scaling exponent (i.e. b ~ −0.12 during winter) for the deformation rate." with "find the magnitude of the scaling exponent to be about 50% lower (ie, b approx -0.12 during winter) for the deformation rate."

done

Line 8: "...the mean sea ice deformation rate" Monthly means?

Kind of. These are daily means of the complete model domain smoothed with a 30-day running mean filter. Added this information to the legend.

Line 10-12: As you can see here with your mean deformation rates, you have much higher values than in figure 8 because you are considering regions of very high strain rates (probably near the coast and in the region of the transpolar drift)... If you are to compare those number with RGPS, you have to bring everything on the same domain covered by RGPS only.

Exactly, that is why we do not compare to RGPS in this section.

Line 13-14: "Some years, e.g., 1997–1999, have clearly reduced summer deformation rates in comparison to, e.g., the beginning of the 1990s or 2007 and 2008." This is not very clear to see on the figure... Maybe plot winter average and summer average on Fig 10 (a) and (b) instead of monthly means?

We kept this for the moment because we think the reduced maxima in 1997-1999 are clearly visible in the time series. If this is of serious concern for you, we could mark the respective years with boxes. We would like to keep the 30-day smoothed time series.

Line 14-15: Delete sentence "The deformation rate during 2008..."

Why? We kept the sentence.

Line 19: "daily mean", Maybe use a 3-day period to be as close as possible to RGPS?

We do not compare to RGPS data here.

Line 20: "the power law scaling exponent b is estimated to be −0.54." Maybe you should show the graph with all the daily mean deformation rates as a function of L and plot the regression line you find. It would make it more clear as to where that number comes from.

We only have three "L" here: 4.5, 9, and 18km. If you calculate the regression for the three graphs shown in a) on a daily basis and average these regression coefficients you get -0.54.

Line 20-21: "Figure 10b shows the deformation rate time series for the three model solutions normalized to a length scale of L = 10 km, using the estimated scaling exponent b = −0.54" How do you do this normalization to a different length scale?

Using eq. 8. Added in text.

Line 23-24: "If looked in detail, however, there remain some quite large differences." This is really not clear on figure. Maybe, as suggested earlier, if you present season means in the graph it would be more clear and we could see better the differences.

Could be. As the regression was done with the daily data we would like to keep it.

-- PAGE 15 --
Line 2-3: "The scaling exponent b gets more negative for weaker sea ice and approaches zero for very strong sea ice, i.e., thick ice and 100% ice concentration" Maybe you need to explain clearly what is the relation between b and Fig.10 b and c. It is the spacing between the curves, ie the larger the space, the larger the slope?

We do not fully understand the question. Are you talking about Fig. 10a and b? then your comment is correct. Fig. 10 c shows how the scaling exponent b between the three model solutions depends on ice concentration. In regions with low ice concentration the difference between the three model solution gets larger. This cannot be seen in Fig. 10b.

Line 6: Replace "even at 100% ice-cover a cell should show power-law scaling behavior." with "a cell should show power law scaling behavior even with a 100% concentration."

Removed that sentence.

Line 7-8: Why is that? So then, can we really expect to find a power-law scaling in winter, when concentration is almost 1 everywhere?

removed

Line 9: Replace "free ice drift" with "free-drift ice"

done

Line 15: Replace "the b values of" with "the values of b of" and replace "b values between" with "the values of b between"

Done

Line 17-18: Why not start the section with this? And then say that the value of b is dependent on the ice concentration and thickness, so that if you consider different regions in the Arctic you end up with different b's. And then present your results when considering the whole Arctic domain.

We added some more explanation about the purpose of this analysis at the beginning of the section: we would like to compare deformation rates obtained with model simulations with different grid spacing. For this we need to consider the whole model domain.

Line 30-31: "model output was bin-averaged to the same spatial scale, L = 12.5 km," What does that mean that the data is bin-averaged? Please explain method.

This section (now 3.2) was completely changed and the procedure used explained in more detail.

-- PAGE 16 --
Line 5: "A linear regression was applied to the PDFs in log-log space 5 for the deformation rate range 0.03–0.8 day−1, shown as dashed lines in Figure 11." Not very visible on the graph. Could be removed or offset.

Changed the color of the PDF lines, which should make the regression line better visible.

Line 25: Girard et al. 2009

corrected
* * *
Line 5: Replace "(ice growth equals ice export)" with "(ie, when ice growth equals ice export)"

Done

Line 10: Ocean sensitivity was never really mentioned in the paper... Delete this sentence?

Removed sentence

Line 11: Replace "more deformation" with "more deformations"

done

Line 11: "the ocean mixed layer depth increases during winter time." This was not shown.

removed

Line 14: Add "Deformations in Arctic ocean and sea ice simulations..."

done

Line 20-21: "The largest difference occurs for the magnitude of divergence, which is 67% to 79% too low (Table 4)." I do not recall seeing this clearly stated in the discussion. Please add.

added

Line 26-27: "This suggests a shortcoming of the ice rheology, for example, the linear dependence between ice strength and ice thickness." Not necessarily... Again, you have to check the ice thickness first. It could be due to the fact that your seasonal ice is too thick.

See answers above. We do not think that our modeled ice is much too thick in the seasonal ice zone because it agrees well with observations (see Nguyen et al., 2011).

---

## Referee Report (RR1)

Review of Sea-ice deformation in a coupled ocean-sea ice model and in satellite remote sensing data

Dear authors,

Thank you for your revised version of the paper. It is clear that you have put in substantial work to address the shortcomings pointed out by the reviewers and I appreciate this. I would also like to reiterate what both I and the other reviewers have said that this paper has the potential to be an important part of the literature and so it is equally important that it be correct and accurate. In line with that I have some major comments, which I would like to ask the authors should address before I can recommend publication.

Major comments:

I still don't think the two parts (sections 3 and 4 of the revised manuscript) are closely related enough to belong in the same paper. The link between the two parts is weak, as mentioned previously, and I believe that only through a substantial effort can this link be strengthened. An effort the authors don't seem to be willing to make, and neither would I, as a matter of fact. Section 4 is also weak scientifically. It gives little new information. The results are qualitatively already known or obvious for members of the community and the quantitative results are of little use as they are specific to the model and set-up. Besides, the model set-up itself is suspect since you don't initialize the ocean properly (see "other comments" for l.9 p.4). But most importantly the interesting science is in section 3 and you don't need a justification to do it! So from there on out section 4 becomes a distraction, or even an annoyance. I earnestly believe that removing section 4 would only strengthen your paper. As such it is the easiest thing you can do to improve your paper!

You still don't filter the results like Bouillon and Rampal suggest. I really think this is regrettable and encourage you to reconsider this choice. You should note also that the noise Bouillon and Rampal discuss is inherent to the method used to calculate the deformation quantities and not inherent to the RGPS data, per se. This means that using the same method on the model results, as you do, also introduces noise there (although it's probably less in your case, since you use triangles, but Kwok uses rectangles). You can therefore not say that there's noise in the observations and not the model and that this could explain some of the discrepencies between model and observations. If you don't filter, both are noisy. Even worse, since you don't filter then the real spatial scaling of the model is substantially weaker than the one reported in the paper.

I am also still not happy with section 3.1.1. Yes, I accept that the averages were properly calculated, although you still don't say it explicitly. Should be done in the first line of section 3.1.1. However, LKFs form on a much shorter time scale than one month. So what we're seeing in the figures is a superimposition of multiple events, something that is not discussed in the text. This is important, because we don't know if the LKFs that form in the model do so repeatedly in the same place or if we have few strong events. In reality they form in multiple

locations. This is, potentially (and in my opinion very probably) a fundamental difference between the model and observations that the current figures and discussion gloss over.

Also, w.r.t. section 3.1.1 I would really recommend considdering a month when the Arctic is full of ice, like March or April. If you do that you will avoid the problem of influences of the open boundary. You will have more pack ice, more LKFs and less MIZ areas. This will help you see the influence of ice thickness, rather than that of concentration, boundary conditions, and ice state (MIZ vs. pack).

In section 3.2.1 you calculate the scaling exponent, $b$, which is a most welcome addition. You do so, however, for the absolute divergence, which is surpriseing to me. We already know that the model has the most difficutlies in simulating the shear rate, so why not calculate the scaling for it? If this gives a poor result and the absoulte divergence is much better, then that is an interesting point for disucssion. In addition it is expected that Hibler's model will have more problems with shear rather than divergence and shear is also the dominant form of deformation in the Arctic. There is therefore plenty of reason to prefer shear over divergence. Even to the point of only showing divergence may look suspicious. The same holds for the PDF calculated in sectino 3.2.2. Also, why don't you do the multi-fractal analysis as well? You have all the ingredients right there and it would tell us more about the distribution of shear rate than the localisation does (section 3.1.3).

Finally, in section 3.2.3 you make a jump from $D$ to $b$, which leaves me completely behind. You don't show any dependence of $b$ on concentration or thickness, only of $D$. I'm not saying it's not there, but you don't show it. This leaves the conclusions to be drawn from this section unfounded. Also, you note that $b$ for the entire area is -0.54, but you previously showed it to be -0.1 for winter and -0.15 for summer. This difference shows you already that you can't compare models of different resolutions using a constant $b$.

I really do feel quite strongly that these points must be addressed adiquetly before this paper can be published.

Other comments:

l.18 p.1: There are no new quantitative metrics here. Everything you do has been done before, but not at such a high resolution VP model - that's what's new here.

l.23 p.2: A *linear* viscous rheology is not enough to reproduce the basic ice state

l.28 p.2: Should be 1,000 km, not 10,000 km

Table 1: You don't say it when introducing the table, but you are using the setup of Nguyen et al (2011), right? These are some interesting parameters you have there. The ice albedos are high (although not impossibly so), but the wet albedo is higher than the dry albedo. This cannot be right. The ocean albedo is also 0.16,

but I'd expect it to be more like 0.06 or maybe 0.10 at the highest in the Arctic. The air-ice drag coefficient is $1.1 \times 10^{-3}$, which is suitable for daily averaged geostrophic winds. I assume you're using 10 m winds every 6 hours. In this case the drag coefficent should probably be double what you have. I've never used JRA-25 myself, but it is very much off if you really need such a low drag coefficent. $P^*$ is also small and $H_0$ large. Not impossibly so, but suspiciously. I'm not sure what to make of all this, but it does not confer confidence in your setup.

l.9 p.4: Why don't you use ECCO2 as initial conditions for your model? If this was an ocean modelling paper then this would be grounds for rejection. You're giving your system a nasty shock by initializing with WOA, but forcing at the boundaries with ECCO2. Also, initializing with WOA requires you to spin up your system for a long (long) time. Fortunately for you then we're only interested in the ice and this probably won't affect the results of section 3 too much. Section 4 however is a different matter since the ocean state is important for the changes you're looking at there. At any rate you should re-run with propper initialisation.

l.19 p.4: The 4.5 km resolution solution does show more structure, but you can hardly call it leads when they are full with about 1 m thick ice.

l.19 p.4: You shouldn't refer to Menemenlis et al here but rather Nguyen et al (2011). Menemenlis et al doesn't even include the Arctic!

l.23 p4. Here's the Nguyen reference. It should come earlier.

l.29 p.4: "modest increase" instead of "modes increase"? Also you mean to compare the 4.5 and 9 km runs to the 18 km one, right?

l.7 p.5: You need to define "winter" and "summer". Are the seasons consistently used throught the paper?

l.21 p.5: "We are bilinearly interpolating" is not propper english. "We interpolate … using a bilinear interpolation" could be one way to go.

l.24 p.5: You use daily output for your calculations, but unfortunately that's not high enough temporal resolution for the 4.5 km grid, since the daily displacement is (often) larger than this. It means that you will transport your Lagrangian points over more than one Eulerian grid cell each time you move them and this causes errors. You should re-run with higher output frequency for the velocities (6 hourly should be ok, I guess), or at the very least discuss the issue.

l.2 p.7: Griard et al also use reconstructed Lagrangian trajectories, like you do. How is what you did different?

l.15 p.7: You look quite closely at November 1999, but there's no mention of any other months or years. Do they look similar? Can we see seasonal differences or inter-annual differences. It's fine to show just one month, but we need to know it's representative.

l.26 p.7: It is to be expected that the vorticity is the best because this is mostly inherited from the atmosphere (and ocean). This should be noted.

l.5 p.8: Ok, here you say it's the same for all 97 months - did you visually inspect them all?

l.17 p.8: The main difference between the seasonal and perennial sea ice is not neccessarily the ice thickness. This is the case for Hibler's model, but in reality floe size and the level of fragmentation and fracturing of the ice is probably more important, or at least as important. This should be noted.

l.22. p.8: Here you outline why it's a bad idea to choose November. You should pick a different month.

l.33 p.8: This part should contain a discussion on why the model is better in summer than winter. Or at least some ideas.

l.24 p.9: This was done first by Marsan et al in 2004, then by Girard et al and Stern and Lindsey in 2009. Marsan et al should be cited.

l.30 p.10: It's hardly a controversy. Controversy requires quite lively debate beween oposing viewpoints, but you're the first person to try to reproduce what Girard et al did (or at least to try and publish it), so there has been no debate in the literature.

l.3. p.11: We need to know the shear rate scaling, as mentioned above.

l.1 p.12: As noted above you need to be carefull when interpreting the noise. There is also noise in our model results.

l.13 p.12: Please show shear rate, not divergence. Also, why use L=20 km, why not 10?

l.16 p.12: What kind of linear regression - least mean squares?

l.18 p.12: Again, be careful with the noise

l.1 p.13: It's only the slopes of the tailes that are in good agreement

l.15 p.13: Now you use $D$ again, why are you switching back and forth?

l.19 p.14: For free drift $b$ gets less negative, not more.

l.24 p.14: This paragraph tells you well why November is a bad choice for section 3.1.1

l.35 p.14: Be sure to stress that it is only in the model that the scaling depends on ice strength, and it is only in the model that the strength depends on

concentration and thickness. In reality there more things that come in to play (floe size and level of fragmentation and fracturing for instance).

l.4 p.21: Again, "the slope of the tails of the PDF", not the PDF it self.

l.14 p.21: Ice conentration and thickenss is not the same as the ice internal stress!

---

## Referee Report (RR2)

**Title: Sea Ice Deformation in a Coupled Ocean-Sea Ice Model and in Satellite Remote Sensing Data**

**Authors: G. Spreen, R. Kwok, D. Menemenlis, A. Nguyen**

**General Comments**

This paper provides significant insights on the ability of viscous-plastic (VP) sea ice models to reproduce observed deformation fields. The confirmation of the power law scaling of VP deformation fields is a welcomed clarification to the previous results of Girard et al. 2009. The description of the dependence of the scaling exponent on sea ice thickness and concentration is also a novelty and is highly relevant for future sea ice rheology studies or comparison of deformations from different models.

I find the revised manuscript much easier to read, better structured and with improved grammar/writing. The key findings are now better highlighted, thanks to the improved organization of the paper. I really appreciate the precision on the methods and how the dataset was acquired for the power law scaling section, but I feel there still needs to be some clarification there (see specific comments).

Most of my previous interrogations have been eliminated given the new information that has been added in the manuscript. The authors indeed responded to all my previous comments and questions in a very complete manner. I therefore recommend that this paper be accepted subject to minor revisions (specific comments below).

Amélie Bouchat, PhD candidate

**Specific comments – Part 1**

- Section 2.3: Maybe rename the section to something like *"Common Reference Frame for Model and Observations"*, because it is more than just transforming your model output to correspond to simulate RGPS data. You have to make it clear that you are not using the original RGPS Lagrangian grid, but that you also manipulate the RGPS data to form a new dataset (resulting from a triangulation).

- Page 11, Line 10-16: The addition of how the strain rates are obtained for different spatial scales is really helpful to understand the following analysis, but I feel like those steps need to be clarified. Your first point ("Strain rates for the six nominal...") should be incorporated in the text before the steps are given, i.e. something like " Following the procedure described in Stern and Lindsay (2009) strain rates for six nominal length scales L* = 10, 20, 50, 100, 200, 500, and 1000 km are calculated. We obtain those strain rates as follows:" and then continue with the different steps. Also, from what I understood of Stern and Lindsay (2009), the first step should be something like : "- Starting from seeding positions, we aggregate all Lagrangian cells that are at a distance (L*/2) or less from the seeding points (center of the aggregate) within a 5-day window. " See also other specific comments below for more on how to rephrase/clarify the other steps.

- Section 3.2.2: The analysis of PDFs is done only for the absolute divergence. It would be interesting to also show the PDFs of shear rates for comparison with the results presented in Girard et al. (2009) and Girard et al. (2011). The PDF analysis is also done here only for the length scale of 20km. I am curious: Have you checked with other length scales? What do you get?

- Page 13, Line 30: You have many $D_i$ (daily mean deformation rates), but only three $L_i$ (grid spacing). Do you average the $D_i$'s for each grid spacing and then perform the least-square fit on the three points $(L_1, <D_1>)$, $(L_2, <D_2>)$, $(L_3, <D_3>)$? Please clarify this for the reader.

- Page 14, Line 24: The one $b$ you get from section 3.2.1 was obtained by "re-sampling" the same data set at different length scale, while in this section (3.2.3) you compare data sets from runs at different spatial resolutions. My first impression is to think that those two scaling exponent represent two different processes, and I am not sure how they can compare to each other, but maybe they are the same and I am wrong… Can you help me on this?

- Page 16, Line 12: The ice velocity is higher for weaker ice (at least initially before the ice gets stronger), and ice thickness is also higher for weaker ice, yet the export is reduced since the beginning of the experiments…? This is confusing and not intuitive. Added panels with time series of mean sea ice thickness and export would probably help understand this.

- Section 4.2/Figure 12: I find it interesting that ice production in both runs, 0.7P* and 0.3 P*, start decreasing at the same time and become negative at the same time. The argument here for the decrease is that because of the reduced ice strength, there is an initial increase in ice thickness, but after a while, the ice thickness increase will compensate for the low P* and ice will be stronger and deform less, which in turn will decrease the ice production, to the point that it becomes even less than in the control run (negative). I would have thought that the run with 0.3P* would reach that moment earlier than the run with 0.7P* because the drift is much faster and the ice is much weaker, so that the overall ice thickness would increase faster to reach that moment when the ice is strong enough to resist deformation again. Again, it would be interesting to see a time series of the mean ice thickness in figure 12 to understand why/how this happens.

- Figure 12: Panel (c) should be panel (b) and vice versa, since they are discussed in that order.

**Specific Comments – Part 2**
Here are some typos and other minor comments/suggestions that the authors may want to consider before final submission.

- Figures 1, 5, 8, 9: Please make labels bigger.

- Page 1, Line 2: "deformation strain rates" keep only "deformations" or "strain rates"

- Page 3, Line 13: "In comparison to…" → "As in…" ?

- Page 3, Line 25: "the model results depend on…" → "the modeled sea ice mass balance depends on…" ?

- Section 2.1: Please state somewhere in this section the period for which the model is run (e.g. January 1st 1992 – December 31st 2009) and also state if a spin up is done prior to starting the period analyzed here.

- Page 4, Line 19: "e.g., develops lead patterns." → "e.g. clear lead patterns." ?

- Page 4, Line 29-30: Replace "shows a mode increase…" with "is  higher for the 4.5 and 18-km simulations by 24 and 28 cm respectively, compared to the 9-km simulation."

- Page 6, Eqs 2-4: Why partial derivative on top and total derivative on the bottom?

- Page 6, Line 7: Put the comma at the end of Eq. (5) instead.

- Page 6, Line 10-11: Not clear why you have two different conditions on ice area. Why not just use the same condition for all analysis? Please clarify.

- Page 6, Line 21: Add "the" before "triangles" and replace "high deformation rates" with "the deformation rates higher than 1 day^-1".

- Page 6, Line 35: Typo: remove the double coma.

- Page 7, Line 31: "Next we…" → "We now…" ?

- Page 8, Line 12: Replace "the model therefore does depend less on" with "the model is therefore less dependent on"

- Page 8, Line 23: Remove "Anyway"

- Page 8, Line 25: "will become more severe" → "will have an important impact" ?

- Section 3.1.2: Rename to "Deformation Rate Time Series" ?

- Page 8, Line 27: "RGPS observations for 97 months from 20 RGPS …" why not just say "all 20 available periods of RGPS observations (i.e. 97 months, between Nov. 1996 and May 2008) are used (Table 2)."

- Page 8 , Line 29: Remove "(both compared with all 20 RGPS periods available)". This is already said in the previous sentence.

- Page 8, Line 29: Replace "Months" with "The months of"

- Page 8, Line 32: Replace "than the one of" with "than all of the"

- Page 8, Line 33: Replace "The same is the case" with "the same is true for'

- Page 8, Line 34: Replace "higher for" with "lower than"

- Page 9 , Line 19: "amount" → "magnitude" ?

- Page 9, Line 22: "we will look" → "we now look" ?

- Page 9, Line 30-31: "If the deformation…" → Move sentence to next paragraph?

- Page 10, Line 1: "As Q is normalized by the total deformation rate… " In the equation above, I see that Q is normalized by the total area… not the total deformation rate.

- Page 10, Line 11: "This is confirmed here by…" I don't understand how the strong localization for the 4.5 km run confirms that the strain rate distributions for the 18 and 9-km runs are similar… Rephrase?

- Page 10, Line 13: Replace "Disregarding" with "Despite"

- Page 10, Line 28: Replace "are, e.g., given in Weiss…" with "given in, e.g., Weiss…"

- Page 10, Line 30: Replace "suitable" with "able"

- Page 10, Line 31: Remove "will"

- Page 11, Line 8: Typo, "special" should be "spatial"

- Page 11,  Line 12: Replace "Lagrangian cells" with "individual Lagrangian cells" and "the area" with "their individual area". Also replace "The sum of all cell areas…" with "The total area of the remaining aggregated cells must be greater than 0.75L*^2.

- Page 11, Line 14: Replace sentence with "For each aggregate, mean strain rates (du/dx, du/dy, etc. - eq. 1) are computed from the individual strain rates in the aggregate by using the individual cell areas as weight. The deformation invariants ([...]) for the aggregates are then computed with those mean strain rates."

- Page 11, Line 15: Replace "length scale" with "actual length scale" and "sample" with "aggregate"

- Page 11, Line 29-30: "Our split in summer and winter…" It is not clear what this sentence is trying to say. Rephrase?

- Page 12, Line 13: "length scale L = 20 km" → "nominal length scale L* = 20 km" ?

- Page 12, Line 13: Add "are then calculated for all winters…" after "for absolute divergence" and delete "were then calculated".

- Page 12, Line 25,27: Do you have the errors on the slopes for the model as well?

- Page 12, Line 26: Remove "mostly"

- Page 12, Line 31: Replace "especially the 4.5 km solutions" with "the 4.5 km solutions especially"

- Page 12, Line 32: Replace "from the about 50% lower deformation rates" with "since the deformation rates are about 50% lower as discussed in section XX"

- Page 15, Line 8: Typo, "an sensitivity" should be "a sensitivity"

- Page 17, Line 25: Typo, "overlineB" should be in equation mode.

---

## Referee Report (RR3)

Third review of ``Sea-ice deformation in a coupled ocean-sea ice model and in satellite remote sensing data'' by G. Spreen, R. Kwok, D. Menemenlis, and A. Nguyen.

Dear authors,

I'm happy to see that the manuscript continues to improve. In my previous review I noted five major issues I asked the authors to address. Two of those have now been adequately addressed so that even if I don't necessarily agree with your approach I cannot say that it gives wrong or misleading results. Three of the five previously raised issues do, however, still need further work.

First it is the issue of filtering. Your response to my previous comments about filtering the data and model results are based on the misunderstanding that the noise in question originates in the observations themselves. It is of course correct that all observations contain some noise and/or uncertainty. But this is not the point. The point is that for any given velocity field, even artificial and perfectly noiseless ones, using the method you use to calculate the deformation rates gives noisy results. It is the method itself that causes this and that has nothing to do with the data. Indeed Bouillon and Rampal (2015) demonstrate this in their paper using an artificial and smooth velocity field. Crucially, this means that both the data and the model will appear to show stronger scaling due to this noise. Even more importantly you therefore cannot say that your failure to remove this noise can contribute in any way to the differences in deformation rates or scaling between data and model. You do this in a number of places in the manuscript and it is simply wrong.

I would argue that you should filter. It doesn't matter what people did before Bouillon and Rampal published their work; filtering gives more accurate results and should be used. Also, if you actually did the multi-fractal analysis (and I still don't understand why you don't – it's not much extra work) you would see immediately why filtering is unavoidable. But I cannot argue that filtering will change your results in a fundamental way (at least as long as you don't do the multi-fractal analysis) and seeing you seem very reluctant to implement the filter I cannot force you. I do however insist that you not use the noise to explain some of the difference between model and observations - that is simply wrong. I also insist that you should note that the noise inherently produced by the method you use causes an overestimation of the scaling exponent $b$, both for the observations and the model.

Secondly I still don't know how the LKFs form in the model. Is it one strong event that produces it or are there repeated weaker events at the same location? Your monthly averages don't show this but this is a potentially important difference. Why don't you show snapshots (3-day means)? Snapshots, or possibly the combination of snapshots and monthly means would be more useful in determining how the LKFs form.

Finally, in section 3.2.3 you discuss the regional differences in $D$ and then apply these differences to $b$ without showing or discussing the regional differences in

*b*. This is the jump from *D* to *b*. The implicit assumption seems to be that low concentration and thin ice gives high deformation rates and thus more negative *b*. But this is not really the case. We get more negative *b* because the mean deformation rate over a large area is smaller than that averaged over a small area. The deformation rate in the MIZ is indeed larger than in the pack, but that fact alone tells us nothing about *b*. It is therefore a prior not clear (to me at least) that large *D* will give large *b*, nor do you provide references to back such a claim up. Rather, to deduce the regional differences in *b* you should follow a partitioning scheme like Stern and Lindsay (2009, their section 7) use to calculate different values of *b* for different regions (their figures 9 and 10), and *b* as a function of multi-year fraction (their figure 11).

In fact all of section 3.2.3, as it stands seems a bit pointless to me. You do give us the dependence of deformation rate on thickness and concentration, but the link to scaling is not there. You also try to compare results from different resolutions using your equation (7) with a constant *b*, but we know *b* is not constant. You show that it changes seasonally and Stern and Lindsay (2009) show that it also changes depending on ice type, so of course you can't use constant *b* for all seasons and all areas, that much is obvious. I would recommend removing this section; I'm not sure how best to save it.

Now, on re-reading section 3.2.3 I realised that I may have overlooked a much more serious problem with it. Do I understand you correctly that you use the same method to calculate *D* for all the three model incarnations, i.e. for each model resolution you calculate the Lagrangian tracks starting from the RGPS positions etc.? I can't really tell from the text, but if this is the case then $L_i$ is 10 km *for all three model runs*. You can use equation (7) to compare different model resolutions if and only if you also calculate *D* at the model resolution. If *D* is calculated at the same resolution for different resolution models the results should be the same! What you show in figure 10a is then a very nice result, but you completely misinterpret it. It shows that there is in fact a fundamental problem with the VP model in that the deformation doesn't scale when you change the model scale. So if my second thoughts on this section are right then it is based on a misunderstanding of the scaling concepts and is completely wrong.

Aside from those major issues there are some minor ones I noted on my previous review but I don't feel are satisfactorily addressed yet (line numbers refer to the revised paper)

l.13 p.1: You removed the word new from the last sentence, but that doesn't really change the meaning of the sentence much. ``New'' is still implied. This last sentence is anyway a poor way to finish the abstract since it does not really summarise what the paper does. Try something along the lines of ``... this study provides an evaluation of high and coarse resolution VP simulations using existing metrics.''

l.17 p2.: You replaced ``viscous rheology'' with ``any nonlinear rheology''. This doesn't really help and is clearly wrong since *any* nonlinear rheology includes an infinite number of rheologies not suitable for modelling sea at all. You also say

that the ``first order mean velocity field as these can be correctly predicted even by simple sea ice models'', but this is also not correct - at least not for my understanding of what constitutes a ``simple sea ice model''. To remove such ambiguity but still motivate considering more advanced metrics you should say that we already know that current sea ice models (i.e. (E)VP class models) are capable of reproducing the first order mean velocity field reasonably well, so now it's time to look at something more difficult.

l.7 p.4 and/or table 1: Please include a reference to the relevant papers by Nguyen et al. for the parameters in table 1. I still think the parameters look strange, but all the more reason to make sure the reader has a proper reference for them. It is true that you cite Nguyen et al (2011) in the following paragraph, but this should be done earlier (and it should be made clearer that you are indeed using their setup, warts and all).

l.13 p.4: Please note that the choice of initial and boundary conditions for the ocean follows Nguyen et al. (2011). Again, this looks jarring to me so it's important to have the reference clear.

l.19 p.5: You use daily outputs to calculate drifter trajectories on a 4.5 km grid. This is insufficient because during one day the drifter will in many cases have drifted out of the grid cell it started in. So if your drifter algorithm looks something like:
   1) Calculate (u,v) at (x,y)
   2) Move drifter by dx = u*(1 day), dy = v*(1 day)
   3) Advance one day and goto 1),
you will incur some error due to the low temporal resolution when dx and dy is large enough for u and v to be substantially different at (x+dx,y+dy) compared to (x,y). I don't pretend this is a major issue (this is the minor issues section after all), but it should be addressed. If re-running at 6 hourly output is not an option say why and say that you don't think this will be a big issue – which is fine.

l.9 p. 9: My previous comment here was perhaps not clear enough. I wonder why the deformation rate is essentially independent of resolution in winter and not in summer – but you surmised as much. There is still no real discussion of this (neither here nor in the discussion section), which would be nice to have. This is a perplexing behaviour because we've seen that the high resolution model appears to give better spatial patterns in the pack ice and so you would expect better mean deformation rate in winter in the high resolution model. But instead it's the summer rates that are better. Does this hold for shear, divergence, and vorticity as well?

l.9 p.13: It's not because RGPS is noisy. More importantly you cannot say that the lack of filtering causes differences between model and observations, as discussed above.

Finally I have some minor remarks following my reading of this latest revision.

l. 2 p.3: Why do you single out Tsamados et al. (2013)? That seems unwarranted by the context.

l.8 p.11: There is no space between the word "section" and the section number.

l.12 p.11: "... we exemplarly use ..." is not proper English. Please rephrase.

l.3 p.12: You use averages at 1,000 km in your scaling calculation. This leaves you pushing against the finite size limit. I know Marsan et al and Stern and Lindsay use 1,000 km, but if you look at Bouillon and Rampal you see that they only go up to 700 km. The reason is that once you get too close to the size of the domain the calculated scaling is poorly affected. You can tell that this is happening because the last point on the log-log plot dips below the straight line. This is the case for figure 2 from Marsan et al, for figure 3 from Stern and Lindsay and for your figure 8. So this last point is suspect and should not be used. This will reduce your *b* value in all cases. You can also see that you have a problem if you compare the slope you get between all pairs of points. This should be more or less the same for all pairs, but it will be radically different for the last pair. It will also be different for the first pairs because of the noise I discussed above.

l.19 p.16: This paragraph starts by talking about seasonal vs. perennial ice, but then changes direction to talk about how the 4.5 km model gives more LKFs and then over to the localisation. This is very confusing and should be fixed. You need to split up the paragraph and rewrite.

l.3 p.17: Sentence should say "... scaling exponents ... is ..." not "are"

l.7 p.17: "... but, however, ..." is not proper English. Please rephrase.

l.13 p.17: You never show how the scaling exponent depends on concentration and thickness.

l.18 p.17: You cannot use the results of Bouillon and Rampal (2015) to explain the difference between model and observations.

l.20 p.17: Missing space between the word divergence and an opening bracket

l.23 p.17: The word reasonable is very subjective, but the number of LKFs produced by the model doesn't seem reasonable to me, even at 4.5 km resolution.

---

## Author Response (AR3)

Dear editor,

Reviewer 1 had still three main concerns with our manuscript:

I)      Filtering of the datasets using the filter presented in Bouillon & Rampal (2015)

We applied the suggested filter to all our RGPS observations and model simulation datasets. All results in the manuscript are now based on that filtered data. Results qualitatively stayed the same.

We included (i) the filter description in section 2.4, (ii) added a section 3.3 comparing filtered and unfiltered results, and (iii) included all results based on the unfiltered data as reference in a Supplementary Material document.

Based on this most of the figures and numbers in the manuscript changed and some text had to be rewritten. The attached manuscript difference file shows all the changes.

II)     Presenting the deformation field maps as 3-day means instead of monthly composites

We added a lot of more examples of the deformation fields to the supplementary material already for the last round of reviews as per request of the reviewer. We think that the monthly better show and support the points we want to make. We therefore stick to the monthly maps and do not exchange them with 3-daily ones. There would be no change in our conclusions based on the 3-daily data. Studying the temporal development of the LKFs in more detail is not the scope of our study.

III)    Concerns with section 3.2.3

The reviewer is concerned that we did not average the model solutions for different spatial scales. We can confirm here that this is not the case and that all calculations are correct. We clarified a sentence in that regards. We do not agree with the reviewers view that the results in this section are not relevant and therefore we kept it.

We included most of the minor comments suggested by the reviewer.

[revised manuscript text omitted]